# How does Gradient Descent Learn Features – A Local Analysis for Regularized Two-Layer Neural Networks

**Mo Zhou**[*]
University of Washington
mozhou17@cs.washington.edu

**Rong Ge**
Duke University
rongge@cs.duke.edu

## Abstract

The ability of learning useful features is one of the major advantages of neural networks. Although recent works show that neural network can operate in a neural tangent kernel (NTK) regime that does not allow feature learning, many works also demonstrate the potential for neural networks to go beyond NTK regime and perform feature learning. Recently, a line of work highlighted the feature learning capabilities of the early stages of gradient-based training. In this paper we consider another mechanism for feature learning via gradient descent through a local convergence analysis. We show that once the loss is below a certain threshold, gradient descent with a carefully regularized objective will capture ground-truth directions. We further strengthen this local convergence analysis by incorporating early-stage feature learning analysis. Our results demonstrate that feature learning not only happens at the initial gradient steps, but can also occur towards the end of training.

## 1 Introduction

Feature learning has long been considered to be a major advantage of neural networks. However, how gradient-based training algorithms can learn useful features is not well-understood. In particular, the most widely applied analysis for overparametrized neural networks is the neural tangent kernel (NTK) (Jacot et al., 2018; Du et al., 2019; Allen-Zhu et al., 2019b). In this setting, the neurons don't move far from their initialization and the features are determined by the network architecture and random initialization (Chizat et al., 2019).

While there are empirical and theoretical evidence on the limitation of NTK regime (Chizat et al., 2019; Arora et al., 2019), extending the analysis beyond the NTK regime has been challenging. For 2-layer networks, an alternative framework for analyzing overparametrized neural networks called mean-field analysis was introduced. Earlier mean-field analysis (e.g., Chizat and Bach, 2018; Mei et al., 2018) require either infinite or exponentially many neurons. Later works (e.g., Li et al., 2020; Ge et al., 2021; Bietti et al., 2022; Mahankali et al., 2024) can analyze the training dynamics of *mildly overparametrized networks* with polynomially many neurons with stronger assumptions on the ground-truth function.

Recently, a growing line of works (Daniely and Malach, 2020; Damian et al., 2022; Abbe et al., 2021, 2022, 2023; Yehudai and Shamir, 2019; Shi et al., 2022; Ba et al., 2022; Mousavi-Hosseini et al., 2023; Barak et al., 2022; Dandi et al., 2023; Wang et al., 2024; Nichani et al., 2024a,b) showed that early stages of gradient training (either one/a few steps of gradient descent or a small amount of time of gradient flow) can be useful in feature learning. These works show that after the early stages of gradient training, the first layer in a 2-layer neural network already captures useful features (usually in the form of a low dimensional subspace), and continuing training the second layer weights will

---

[*]Work done at Duke University.

38th Conference on Neural Information Processing Systems (NeurIPS 2024).

give performance guarantees that are stronger than any kernel or random feature based models. In this work, we consider the natural follow-up question:

*Does feature learning only happen in the early stages of gradient training?*

We show that this is not the case by demonstrating feature learning capability for the final stage of gradient training – local convergence. In particular, we prove the following result:

**Theorem 1** (Informal). *If the data is generated by a 2-layer teacher network $f_*$, as long as the width of student network $m$ is at least some quantity $m_0$ that only depends on $f_*$, a variant of gradient descent algorithm (Algorithm 1, roughly gradient descent with decreasing weight decay) can recover the target network within polynomial time. Moreover, the student neurons align with the teacher neurons at the end.*

Our result highlights the different mechanisms of feature learning: previous works show that in the early stages of gradient descent, the network learns the *subspace* spanned by the neurons in the teacher network. Our local convergence result shows that at later stages, gradient descent is able to learn the *exact directions* of the teacher neurons, which are much more informative compared to the subspace and lead to stronger guarantees.

Analyzing the entire training dynamics is still challenging so in our algorithm (see Algorithm 1) we use a convex second stage to "fast-forward" to the local analysis. Our technique for local convergence is similar to the earlier work (Zhou et al., 2021), however we consider a more complicated setting with ReLU activation and allow second-layer weights to be both positive or negative. This change requires additional regularization in the form of standard weight decay and new dual certificate analysis.

## 1.1 Related works

**Neural Tangent Kernel** Early works often studied neural network optimization using NTK theory (Jacot et al., 2018; Allen-Zhu et al., 2019b; Du et al., 2019). It is shown that highly-overparametrized neural nets are essentially kernel methods under certain initialization scale. However, NTK theory cannot explain the performance of neural nets in practice (Arora et al., 2019) and leads to lazy training dynamics that neurons stay close to their initialization (Chizat et al., 2019). Hence, later research efforts (e.g., Allen-Zhu et al., 2019a; Bai and Lee, 2020; Li et al., 2020), as well as current paper, focus on feature learning regime where neural nets can learn features and outperform kernel methods.

**Early stage feature learning** Researchers have recently tried to understand how neural networks trained with gradient descent (GD) can learn features, going beyond the kernel/lazy regime (Jacot et al., 2018; Chizat et al., 2019). A typical setup is to use 2-layer neural networks to learn certain target function, often equipped with low-dimensional structure. Examples include learning polynomials (Yehudai and Shamir, 2019; Damian et al., 2022), single-index models (Ba et al., 2022; Mousavi-Hosseini et al., 2023; Moniri et al., 2024; Cui et al., 2024), multi-index models (Dandi et al., 2023), sparse boolean functions (Abbe et al., 2021, 2022, 2023), sparse parity functions (Daniely and Malach, 2020; Shi et al., 2022; Barak et al., 2022) and causal graph (Nichani et al., 2024b). Also, few works use 3-layer networks as learner model (Nichani et al., 2024a; Wang et al., 2024). These works essentially showed that feature learning happens in the early stage of training. Specifically, they often use 2-stage layer-wise training procedure: first-layer weights/features are only trained with one or few steps of gradient descent/flow and only update the second-layer afterwards. Our results give a complementary view that feature learning can also happen in the *final stage* training that leading student neurons eventually align with ground-truth directions. This cannot be achieved if first-layer weights are fixed after few steps.

**Learning single/multi-index models with neural networks** Single/Multi-index models are the functions that only depend on one or few directions of the high dimensional input. Many recent works have studied the problem of using 2-layer networks to learn single-index models (Soltanolkotabi, 2017; Yehudai and Ohad, 2020; Frei et al., 2020; Wu, 2022; Bietti et al., 2022; Xu and Du, 2023; Berthier et al., 2023; Mahankali et al., 2024) and multi-index models (Damian et al., 2022; Bietti et al., 2023; Suzuki et al., 2024; Glasgow, 2024). These works show the advantages of feature learning over fixed random features in various settings. In this paper, we consider target multi-index function that can be represented by a small 2-layer network, and show a variant of GD with weight decay can learn it and, moreover, recover the ground-truth directions.

**Local loss landscape**    Safran et al. (2021) showed that in the overparametrized case with orthogonal teacher neurons, even around the local region of global minima, the landscape neither is convex nor satisfies PL condition. Chizat (2022) considered square loss with $\ell_2$ regularization similar to our setup and showed the local loss landscape is strongly-convex under certain non-degenerate assumptions. However, it is not known when such assumptions actually hold and the proof cannot handle ReLU. Later Akiyama and Suzuki (2021) gives a result for ReLU, but the non-degeneracy assumption is still required (and also focus on effective $\ell_1$ regularization instead of $\ell_2$ regularization). Zhou et al. (2021) studies a similar local convergence setting but restricts second-layer weights to be positive and uses absolute activation. In this paper, we focus on a more natural but technically challenging case that second-layer can be positive and negative and using ReLU activation. We develop new techniques to overcome the above challenges (additional assumption, ReLU, standard second-layer, etc).

## 2    Preliminary

**Notation**    Let $[n]$ be set $\{1, \ldots, n\}$. For vector $\boldsymbol{w}$, we use $\|\boldsymbol{w}\|_2$ for its 2-norm and $\overline{\boldsymbol{w}} = \boldsymbol{w}/\|\boldsymbol{w}\|_2$ as its normalized version. For two vectors $\boldsymbol{w}, \boldsymbol{v}$ we use $\angle(\boldsymbol{w}, \boldsymbol{v}) = \arccos(|\boldsymbol{w}^\top \boldsymbol{v}|/(\|\boldsymbol{w}\|_2 \|\boldsymbol{v}\|_2)) \in [0, \pi/2]$ as the angle between them (up to a sign).For matrix $\boldsymbol{A}$ let $\|\boldsymbol{A}\|_F$ be its Frobenius norm. We use standard $O, \Omega, \Theta$ to hide constants and $\widetilde{O}, \widetilde{\Omega}, \widetilde{\Theta}$ to hide polylog factors. We use $O_*, \Omega_*, \Theta_*$ to hide problem dependent parameters that only depend on the target network (see paragraph above (1)).

**Teacher-student setup**    We will consider the teacher-student setup for two-layer neural networks with Gaussian input $\boldsymbol{x} \sim N(\boldsymbol{0}, \boldsymbol{I}_d)$. The goal is to learn the teacher network of size $m_*$

$$f_*(\boldsymbol{x}) = \sum_{i=1}^{m^*} a_i^* \sigma(\boldsymbol{w}_i^{*\top} \boldsymbol{x}) + \boldsymbol{w}_0^{*\top} \boldsymbol{x} + b_0^*,$$

where $\sigma(x) := \max\{0, x\}$ is ReLU activation, $S_* := \mathrm{span}\{\boldsymbol{w}_1^*, \ldots, \boldsymbol{w}_{m^*}^*\}$ is the target subspace. Without loss of generality, we will assume $\|\boldsymbol{w}_i^*\|_2 = 1$ due to the homogeneity of ReLU.

Following the recent line of works in learning single/multi-index models (Ba et al., 2022; Damian et al., 2022), we assume the target network has low dimensional structure.

**Assumption 1.** *Teacher neurons form a low dimensional subspace in $\mathbb{R}^d$, that is*

$$\dim(S_*) = \dim(\mathrm{span}\{\boldsymbol{w}_1^*, \ldots, \boldsymbol{w}_{m^*}^*\}) = r \ll d.$$

We will also assume the teacher neurons are non-degenerate in the following sense:

**Assumption 2.** *Teacher neurons are $\Delta$-separated, that is angle $\angle(\boldsymbol{w}_i^*, \boldsymbol{w}_j^*) \geq \Delta$ for all $i \neq j$.*

**Assumption 3.** $\boldsymbol{H} := \sum_{i=1}^{m^*} a_i^* \boldsymbol{w}_i^* \boldsymbol{w}_i^{*\top}$ *is non-degenerate in target subspace $S_*$, i.e., $\mathrm{rank}(H) = r$. Denote $\kappa := |\lambda_r(\boldsymbol{H})|$.*

Assumption 2 simply requires all teacher neurons pointing to different directions, which is crucial for identifiability (Zhou et al., 2021).

Assumption 3 says the target network contains low-order (second-order) information, which is related with the notion of information exponent (Arous et al., 2021). In our setting, the information exponent is at most 2 due to Assumption 3. Indeed, one can show $\mathbb{E}_{\boldsymbol{x}}[f_*(\boldsymbol{x})h_2(\boldsymbol{v}^\top \boldsymbol{x})] = \hat{\sigma}_2 \boldsymbol{v}^\top \boldsymbol{H} \boldsymbol{v}$, where $h_2(x)$ is the 2nd-order normalized Hermite polynomial and $\hat{\sigma}_2$ is the 2nd Hermite coefficient of ReLU. See Appendix A for more details. Many previous works also rely on same or similar assumption to show neural networks can learn features to perform better than kernels (Damian et al., 2022; Abbe et al., 2022; Ba et al., 2022).

In this paper, we are interested in the case where the complexity of target network is small. Therefore, we will use $O_*, \Omega_*, \Theta_*$ to hide $\mathrm{poly}(r, m_*, \Delta, |a_1|, \ldots, |a_{m_*}|, \kappa)$, which is the polynomial dependency on relevant parameters of target $f_*$ (does not depend on student network).

We will use the following overparametrized student network:

$$f(\boldsymbol{x}; \boldsymbol{\theta}) = \sum_{i=1}^{m} a_i \sigma(\boldsymbol{w}_i^\top \boldsymbol{x}) + \alpha + \boldsymbol{\beta}^\top \boldsymbol{x}, \tag{1}$$

where $\boldsymbol{a} = (a_1, \ldots, a_m)^\top \in \mathbb{R}^m$, $\boldsymbol{W} = (\boldsymbol{w}_1 \cdots \boldsymbol{w}_m)^\top \in \mathbb{R}^{m \times d}$ and $\boldsymbol{\theta} = (\boldsymbol{a}, \boldsymbol{W}, \alpha, \boldsymbol{\beta})$.

**Loss and algorithm** Consider the square loss function with $\ell_2$ regularization under Gaussian input

$$L_\lambda(\boldsymbol{\theta}) = \mathbb{E}_{\boldsymbol{x} \sim N(0, \boldsymbol{I}_d)}[(f(\boldsymbol{x}; \boldsymbol{\theta}) - \widetilde{y})^2] + \frac{\lambda}{2} \|\boldsymbol{a}\|_2^2 + \frac{\lambda}{2} \|\boldsymbol{W}\|_2^2. \tag{2}$$

We will use $L$ to denote the square loss for simplicity. The $\ell_2$ regularization is the same as the commonly used weight decay in practice. Our goal is to find the minima of unregularized problem ($\lambda = 0$) to recover teacher network $f_*$. However, directly analyzing the unregularized problem is challenging so instead we choose to analyze the regularized problem and will gradually let $\lambda \to 0$.

In above, we use preprocessed data $(x, \widetilde{y})$ in the loss function as in Damian et al. (2022). Specifically, given any $(\boldsymbol{x}, y)$ with $y = f_*(\boldsymbol{x})$, denote $\alpha_* = \mathbb{E}_{\boldsymbol{x}}[y]$ and $\boldsymbol{\beta}_* = \mathbb{E}_{\boldsymbol{x}}[y\boldsymbol{x}]$, we get

$$\widetilde{f}_*(\boldsymbol{x}) = \widetilde{y} = y - \alpha_* - \boldsymbol{\beta}_*^\top \boldsymbol{x}. \tag{3}$$

This preprocessing process essentially removes the 0-th and 1-st order term in the Hermite expansion of $\sigma$. See Appendix A for a brief introduction of Hermite polynomials and Claim B.1.

Our algorithm is shown in Algorithm 1. It is roughly the standard GD following a given schedule of weight decay $\lambda_t$ that goes to 0. Due to the difficulty in analyzing gradient descent training beyond early and final stage, we choose to only train the norms in Stage 2 as a tractable way to reach the local convergence regime.

We will use symmetric initialization that $a_i = -a_{i+m/2}$, $\boldsymbol{w}_i = \boldsymbol{w}_{i+m/2}$ with $a_i \sim \mathrm{Unif}\{\pm\sqrt{d}\}$, $\boldsymbol{w}_i \sim \mathrm{Unif}((1/\sqrt{m})\mathbb{S}^{d-1})$, $\alpha = 0$, $\boldsymbol{\beta} = \boldsymbol{0}$. Our analysis is not sensitive to the initialization scale we choose here. The choice is just for the simplicity of the proof.

---

**Algorithm 1:** Learning 2-layer neural networks

---

**Input:** initialization $\boldsymbol{\theta}^{(0)}$, weight decay $\lambda_t$ and stepsize $\eta_t$
Data preprocess: get $(\boldsymbol{x}, \widetilde{y})$ according to (3)
Stage 1: one step gradient update
$\quad \boldsymbol{\theta}^{(1)} \leftarrow \boldsymbol{\theta}^{(0)} - \eta_0 \nabla_{\boldsymbol{\theta}} L_{\lambda_0}(\boldsymbol{\theta}^{(0)})$
Stage 2: norm adjustment by convex program
$\quad \boldsymbol{a}^{(T_2)}, \alpha^{(T_2)}, \boldsymbol{\beta}^{(T_2)} \leftarrow \min_{\boldsymbol{a}, \alpha, \boldsymbol{\beta}} L(\boldsymbol{a}, \boldsymbol{W}^{(1)}, \alpha, \boldsymbol{\beta}) + \lambda \sum_i \|\boldsymbol{w}_i\|_2 |a_i|$
$\quad$ Balancing norm between two layers s.t. $|a_i| = \|\boldsymbol{w}_i\|_2$ for all $i$
Stage 3: local convergence
$\quad$ **for** $k \le K$ **do** $\qquad\qquad$ // for each epoch, run GD until convergence
$\quad\quad$ **for** $T_{3,k-1} \le t \le T_{3,k}$ **do**
$\quad\quad\quad \boldsymbol{\theta}^{(t+1)} \leftarrow \boldsymbol{\theta}^{(t)} - \eta \nabla_{\boldsymbol{\theta}} L_{\lambda_{3,k}}(\boldsymbol{\theta}^{(t)})$
**Output:** $\boldsymbol{\theta}^{(T_{3,K})} = (\boldsymbol{a}^{(T_{3,K})}, \boldsymbol{W}^{(T_{3,K})}, \alpha^{(T_{3,K})}, \boldsymbol{\beta}^{(T_{3,K})})$

---

## 3 Main results

In this section, we give our main result that shows training student network using Algorithm 1 can recover the target network within polynomial time. We will focus on the case that $d \ge \Omega_*(1)$ when the complexity of target function is small.

**Theorem 2** (Main result). *Under Assumption 1, 2, 3, consider Algorithm 1 on loss (2). There exists a schedule of weight decay $\lambda_t$ and step size $\eta_t$ such that given $m \ge m_0 = \widetilde{O}_*(1) \cdot (1/\varepsilon_0)^{O(r)}$ neurons with small enough $\varepsilon_0 = \Theta_*(1)$, with high probability we will recover the target network $L(\boldsymbol{\theta}) \le \varepsilon$ within time $T = O_*(1/\eta\varepsilon^2)$ where $\eta = \mathrm{poly}(\varepsilon, 1/d, 1/m)$.*

*Moreover, when $\varepsilon \to 0$ every student neuron $\boldsymbol{w}_i$ either aligns with one of teacher neuron $\boldsymbol{w}_j^*$ as $\angle(\boldsymbol{w}_i, \boldsymbol{w}_j^*) = 0$ or vanishes as $|a_i| = \|\boldsymbol{w}_i\| = 0$.*

Note that our results can be extended to only have access to polynomial number of samples by using standard concentration tools. We omit the sample complexity for simplicity. See more discussion in Appendix J. We emphasize that the required width $m_0$ only depends on the complexity of target function $f_*$ (only quantities that are related to $f_*$, not student network $f$ or error $\varepsilon$), so any mildly overparametrized networks can learn $f_*$ efficiently to arbitrary small error.

The analysis consists of three stages: early-stage feature learning (Stage 1 and 2) and final-stage feature learning/local convergence (Stage 3). It will be clear in the later section that $\varepsilon_0$ is in fact the threshold to enter the local convergence regime. See Section 4 for more details.

Our result improves the previous works that only train the first layer weight with small number of gradient steps at the beginning (Damian et al., 2022; Ba et al., 2022; Abbe et al., 2021, 2022, 2023). In these works, neural networks only learn the target subspace and do random features within it (see Section 4.1 for more details). Intuitively, these random features need to span the whole space of the target function class to perform well, which means its number (the width) should be on the order of the dimension of target function class. For 2-layer networks, random features in the target subspace need $(1/\varepsilon)^{O(r)}$ neurons to achieve desired accuracy $\varepsilon$. In contrast, continue training both layer at the last phase of training allows us to learn not only subspace but also exactly the ground-truth directions. Moreover, we only use $(1/\varepsilon_0)^{O(r)}$ neurons that only depends on the complexity of target network. This highlights the benefit of continue training first layer weights instead of fixing them after first step.

## 4 Proof overview

In this section, we give the proof overview of these three stages separately.

Denote the optimality gap $\zeta$ at time $t$ as the difference between current loss and the best loss one could achieve with networks of any size (including infinite-width networks)

$$\zeta_t = L_{\lambda_t}(\boldsymbol{\theta}^{(t)}) - \min_{\mu \in \mathcal{M}(\mathbb{S}^{d-1})} L_{\lambda_t}(\mu), \tag{4}$$

where $\mathcal{M}(\mathbb{S}^{d-1})$ is the set of measures on the sphere $\mathbb{S}^{d-1}$. As an example, if $\mu = \sum_i a_i \|\boldsymbol{w}_i\| \delta_{\overline{\boldsymbol{w}}_i}$, then $L_\lambda(\mu)$ recovers $L_\lambda(\boldsymbol{\theta})$ when linear term $\alpha, \beta$ are perfectly fitted and norms are balanced $|a_i| = \|\boldsymbol{w}_i\|$. We defer the precise definition of $L_\lambda(\mu)$ to (6) in appendix.

### 4.1 Stage 1

For Stage 1, we show in the lemma below that the first step of gradient descent identifies the target subspace and ensures there always exists student neuron that is close to every teacher neuron.

**Lemma 3** (Stage 1). *Under Assumption 1,2,3, consider Algorithm 1 with $\lambda_0 = \eta_0 = 1$ and $m \geq m_0 = \widetilde{O}_*(1) \cdot (1/\varepsilon_0)^{O(r)}$ with any $\varepsilon_0 = \Theta_*(1)$. After first step, with probability $1 - \delta$ we have*

*(i) for every teacher neuron $\boldsymbol{w}_i^*$, there exists at least one student neuron $\boldsymbol{w}_j$ s.t. $\angle(\boldsymbol{w}_i^*, \boldsymbol{w}_j) \leq \varepsilon_0$.*

*(ii) $\left\|\boldsymbol{w}_i^{(1)}\right\|_2 = \Theta_*(1)$, $|a_i^{(1)}| \leq O_*(1/\sqrt{m})$ for all $i \in [m_*]$, $\alpha_1 = 0$ and $\boldsymbol{\beta}_1 = \boldsymbol{0}$.*

The key observation here is similar to Damian et al. (2022) that $\boldsymbol{w}_i^{(1)} \approx -2\eta_0 a_i^{(0)} \left(\hat{\sigma}_2^2 \boldsymbol{H} \overline{\boldsymbol{w}}_i\right)$ so that given $\boldsymbol{H}$ is non-degenerate in target subspace $S_*$ we essentially sample $\boldsymbol{w}_i^{(1)}$ from the target subspace. It is then natural to expect that the neurons form an $\varepsilon_0$-net in the target subspace given $m_0$ neurons.

### 4.2 Stage 2

Given the learned features (first-layer weights) in Stage 1, we now perform least squares to adjust the norms and reach a low loss solution in Stage 2.

**Lemma 4** (Stage 2). *Under Assumption 1,2,3, consider Algorithm 1 with $\lambda_t = \sqrt{\varepsilon_0}$. Given Stage 1 in Lemma 3, we have Stage 2 ends within time $T_2 = \widetilde{O}_*(1/\eta\varepsilon_0)$ such that optimality gap $\zeta_{T_2} = O_*(\varepsilon_0)$.*

It remains an open problem to prove the convergence when training both layers simultaneously beyond early and final stage. To overcome this technical challenge, we choose to use a simple least square for Stage 2. We use the simple (sub)gradient descent to optimize this loss. There exist many other algorithms that can solve this Lasso-type problem, but we omit it for simplicity as this is not the main focus of this paper.

Note that the regularization in Algorithm 1 is the same as standard weight decay when we train both layers. This regularization leads to several desired properties at the end of Stage 2: (1) prevent norm cancellation between neurons: neurons with similar direction but different sign of second layer weights cancel with each other; (2) neurons mostly concentrate around ground-truth directions. As we will see later, these nice properties continue to hold in Stage 3, thanks to the regularization.

### 4.3 Stage 3

After Stage 2 we are in the local convergence regime. The following lemma shows that we could recover the target network within polynomial time using a multi-epoch gradient descent with decreasing weight decay $\lambda$ at every epoch. Note that this result only requires the initial optimality gap is small and width $m \geq m_*$ (target network width, not $m_0$).

**Lemma 5** (Stage 3). *Under Assumption 1,2,3, consider Algorithm 1 on loss (2). Given Stage 2 in Lemma 4, if the initial optimality gap $\zeta_{3,0} \leq O_*(\lambda_{3,0}^{9/5})$, weight decay $\lambda$ follows the schedule of initial value $\lambda_{3,0} = O_*(1)$, and $k$-th epoch $\lambda_{3,k} = \lambda_{3,k-1}/2$ and stepsize $\eta_{3k} = \eta \leq O_*(\lambda_{3,k}^{12} d^{-3})$ for all $T_{3,k} \leq t \leq T_{3,k+1}$ in epoch $k$, then within $K = O_*(\log(1/\varepsilon))$ epochs and total $T_3 - T_2 = O_*(\lambda_{3,0}^{-4} \eta^{-1} \varepsilon^{-2})$ time we recover the ground-truth network $L(\boldsymbol{\theta}) \leq \varepsilon$.*

The lemma above relies on the following result that shows the local landscape is benign in the sense that it satisfies a special case of Łojasiewicz property (Lojasiewicz, 1963). This means GD can always make progress until the optimality gap $\zeta$ is small.

**Lemma 6** (Gradient lower bound). *When $\Omega_*(\lambda^2) \leq \zeta \leq O_*(\lambda^{9/5})$ and $\lambda \leq O_*(1)$, we have*

$$\|\nabla_{\boldsymbol{\theta}} L_\lambda\|_F^2 \geq \Omega_*(\zeta^4/\lambda^2).$$

Note that this generalizes previous result in Zhou et al. (2021) that only focuses on 2-layer networks with positive second layer weights. This turns out to be technically challenging as two neurons with different signs can cancel each other. We discuss how to deal with this challenge in the next section.

## 5 Descent direction in local convergence (Stage 3): the benefit of weight decay

In this section, we give the high-level proof ideas for the most technical challenging part of our results — characterize the local landscape in Stage 3 (Lemma 6).

The key idea is to construct descent direction — a direction that has positive correlation with the gradient direction. The gradient lower bound follows from the existence of such descent direction.

It turns out that the existence of both positive and negative second-layer weights introduces significant challenge for the analysis: there might exist neurons with similar directions (e.g., $(a, \boldsymbol{w})$ and $(-a, \boldsymbol{w})$) that can cancel with each other to have no effect on the output of network. Intuitively, we would hope all of them to move towards 0, but they have no incentive to do so. Moreover, if they are not exactly symmetric it's hard to characterize which directions these neurons will move.

We use standard weight decay to address the above challenge. Specifically, weight decay helps us to

- *Balance norm between neurons.* When norm between two layers are balanced, the $\ell_2$ regularization $\sum_i |a_i|^2 + \|\boldsymbol{w}_i\|^2$ would become the effective $\ell_1$ regularization $2\sum_i |a_i| \|\boldsymbol{w}_i\|$ over the distribution of neurons. Such sparsity penalty ensures most neurons concentrate around the ground-truth directions, especially preventing norm cancellation between far-away neurons.
- *Reduce cancellation between close-by neurons.* For close-by neurons, weight decay helps to reduce the norm of neurons with the 'incorrect' sign (different sign with the ground-truth neuron). This is because weight decay prefers low norm solutions, and reducing cancellations between neurons can reduce total norm (regularization term) while keeping the square loss same.

We will group the neurons (i.e., partitioning $\mathbb{S}^{d-1}$) based on their distance to the closest teacher neurons: denote $\mathcal{T}_i = \{\boldsymbol{w} : \angle(\boldsymbol{w}, \boldsymbol{w}_i^*) \leq \angle(\boldsymbol{w}, \boldsymbol{w}_j^*)$ for any $j \neq i\}$ (break the tie arbitrarily) so that $\cup_i \mathcal{T}_i = \mathbb{S}^{d-1}$. We will also use $\delta_j$ to denote $\angle(\boldsymbol{w}_j, \boldsymbol{w}_i^*)$ for $j \in \mathcal{T}_i$.

As described above, weight decay can always lead to descent direction when norms are not balanced or norm cancellation happens (see Lemma F.15 and Lemma F.16). The following lemma shows that in other scenarios we can always improve features towards the ground-truth directions.

**Lemma 7** (Feature improvement descent direction, informal). *When norms are balanced and no norm cancellation happens, there exists properly chosen $q_{ij} \geq 0$ and $\sum_{j \in \mathcal{T}_i} a_j q_{ij} = a_i^*$ such that*

$$\sum_{i \in [m_*]} \sum_{j \in \mathcal{T}_i} \langle \nabla_{\boldsymbol{w}_i} L_\lambda, \boldsymbol{w}_j - q_{ij} \boldsymbol{w}_i^* \rangle = \Omega(\zeta).$$

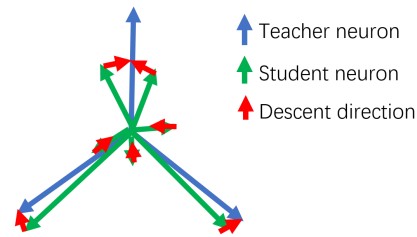

Figure 1: Illustration of descent direction

In words, this descent direction is the following: we move neuron $\boldsymbol{w}_j \in \mathcal{T}_i$ toward either ground-truth direction $\boldsymbol{w}_i^*$ or 0 depending on whether it is in the neighborhood of teacher neuron $\boldsymbol{w}_i^*$. Specifically, we move far-away neurons towards 0 (and thus setting $q_{ij} = 0$) and move close-by neurons towards its 'closest' minima $q_{ij}\boldsymbol{w}_i^*$ (the fraction of $\boldsymbol{w}_i^*$ that neuron $\boldsymbol{w}_j$ should target to approximate). See Figure 1 for an illustration.

The proof of the above lemma requires a dedicated characterization of the low loss solution's structure, which we describe in Section 6.

## 6 Structure of (approximated) minima

In this section, we first highlight the importance of understanding local geometry by showing the challenges in proving the existence of descent direction (Lemma 7). Then after presenting the main result of this section to show the structure of (approximated) minima (Lemma 8), we discuss several proof ideas such as dual certificate analysis in the remaining part.

### 6.1 Constructing descent direction requires better understanding of local geometry

To show the existence of descent direction in Lemma 7, we compute the inner product between gradient and constructed descent direction. We can lower bound it by (assuming norms are balanced)

$$\zeta + 2 \sum_{i \in [m_*]} \sum_{j \in \mathcal{T}_i} \mathbb{E}_{\boldsymbol{x}}[R(\boldsymbol{x}) a_j q_{ij} \boldsymbol{w}_i^{*\top} \boldsymbol{x} (\sigma'(\boldsymbol{w}_i^{*\top}\boldsymbol{x}) - \sigma'(\boldsymbol{w}_j^\top \boldsymbol{x}))],$$

where $R(\boldsymbol{x}) = f(\boldsymbol{x}) - \widetilde{f}_*(\boldsymbol{x})$ is the residual. Thus, in order to get a lower bound, the goal is to show second term above is small than $\zeta$. As we can see, this term is quite complicated and can be viewed as the inner product between $R(\boldsymbol{x})$ and $h(\boldsymbol{x}) = \sum_{i \in [m_*]} \sum_{j \in \mathcal{T}_i} a_j q_{ij} \boldsymbol{w}_i^{*\top} \boldsymbol{x} (\sigma'(\boldsymbol{w}_i^{*\top}\boldsymbol{x}) - \sigma'(\boldsymbol{w}_j^\top \boldsymbol{x}))$.

**Average neuron and residual decomposition** To deal with above challenge, we use the idea of average neuron and residual decomposition. For each teacher neuron $\boldsymbol{w}_i^*$, denote $\boldsymbol{v}_i = \sum_{j \in \mathcal{T}_i} a_j \boldsymbol{w}_j$ as the average neuron. Intuitively, this average neuron $\boldsymbol{v}_i$ stands for an idealize case where all neurons belong to $\mathcal{T}_i$ (closer to $\boldsymbol{w}_i^*$ than other $\boldsymbol{w}_j^*$) collapse into a single neuron.

We decompose the residual $R(\boldsymbol{x}) = f(\boldsymbol{x}) - \widetilde{f}_*(\boldsymbol{x})$ into the 3 terms below: denote $\hat{\boldsymbol{v}}_i = \boldsymbol{v}_i - \boldsymbol{w}_i^*$

$$R_1(\boldsymbol{x}) = \frac{1}{2} \sum_{i \in [m_*]} \hat{\boldsymbol{v}}_i^\top \boldsymbol{x}\, \text{sign}(\boldsymbol{w}_i^{*\top}\boldsymbol{x}), R_2(\boldsymbol{x}) = \frac{1}{2} \sum_{i \in [m_*], j \in \mathcal{T}_i} a_j \boldsymbol{w}_j^\top \boldsymbol{x}(\text{sign}(\boldsymbol{w}_j^\top \boldsymbol{x}) - \text{sign}(\boldsymbol{w}_i^{*\top}\boldsymbol{x})),$$

$$R_3(\boldsymbol{x}) = \frac{1}{\sqrt{2\pi}} \left( \sum_{i \in [m_*]} a_i^* \|\boldsymbol{w}_i^*\|_2 - \sum_{i \in [m]} a_i \|\boldsymbol{w}_i\|_2 \right) + \alpha - \hat{\alpha} + (\boldsymbol{\beta} - \hat{\boldsymbol{\beta}})^\top \boldsymbol{x}.$$

$R_1$ can be thought as the exact-parametrization setting (use $m_*$ neurons to learn $m_*$ neurons), where the average neurons $\{\boldsymbol{v}_i\}_{i=1}^{m_*}$ are the effective neurons. The difference between this exact-parametrization and overparametrization setting is then characterized by the term $R_2$, which captures the difference in nonlinear activation pattern. This term in fact suggests the loss landscape is degenerate in overparametrized case and slows down the convergence (Zhou et al., 2021; Xu and Du, 2023). Overall, this residual decomposition is similar to Zhou et al. (2021), with additional modification of $R_3$ to deal with ReLU activation and linear term $\alpha, \boldsymbol{\beta}$.

To some extent, our residual decomposition can be viewed as a kind of 'bias-variance' decomposition in the sense that the 'bias' term $R_1$ captures the overall average contribution of all neurons, and the 'variance' term $R_2$ captures the individual contributions of each neuron that are not reflected in $R_1$.

**High-level proof plan of Lemma 7** We now are ready to give a proof plan for Lemma 7. The key is to show properties of minima that can help us to bound $\langle R, h \rangle$.

1. Show that neurons mostly concentrate around ground-truth directions.

2. Show that average neuron $\boldsymbol{v}_i$ is close to teacher neuron $\boldsymbol{w}_i^*$ for all $i \in [m]$.

3. Use above structure to bound $\langle R_i, h \rangle$. Specifically, bounding $\langle R_1, h \rangle$ relies on the fact that average neuron is close to teacher neuron (step 2); a bound on $\langle R_2, h \rangle$ follows from far-away neurons are small (step 1); third term $\langle R_3, h \rangle$ can be directly bounded using the loss. Detailed calculations are deferred into Appendix H.3.

We give main result of this section that shows the desired local geometry properties more precisely ((i)(ii) corresponding to step 1 and (iii) corresponding to step 2 above).

**Lemma 8** (Informal). *Suppose the optimality gap is $\zeta$, we have*

*(i) Total norm of far-away neurons is small:* $\sum_{i \in [m_*]} \sum_{j \in \mathcal{T}_i} |a_j| \, \|\boldsymbol{w}_j\|_2 \, \delta_j^2 = O_*(\zeta/\lambda)$, *where angle* $\delta_j = \angle(\boldsymbol{w}_j, \boldsymbol{w}_i^*)$ *for* $\boldsymbol{w}_j$ *that* $j \in \mathcal{T}_i$.

*(ii) For every* $\boldsymbol{w}_i^*$, *there exists at least one close-by neuron* $\boldsymbol{w}$ *s.t.* $\angle(\boldsymbol{w}, \boldsymbol{w}_i^*) \leq \delta_{close} = O_*(\zeta^{1/3})$.

*(iii) Average neuron is close to teach neurons: we have* $\|\boldsymbol{v}_i - \boldsymbol{w}_i^*\|_2 \leq O_*((\zeta/\lambda)^{3/4})$.

These properties give us a sense of what the network should look like when loss is small: neurons have large norm only if they are around the ground-truth directions. Moreover, when $\zeta/\lambda \to 0$, student neuron must align with one of teacher neurons ($\delta_j = 0$) or norm becomes 0 ($|a_j| \, \|\boldsymbol{w}_j\| = 0$). This can be understood from the $\ell_1$ regularized loss (equivalent to $\ell_2$ regularization on both layers) that promotes the sparsity over the distribution of neurons. In the rest of this section, we discuss new techniques such as dual certificate that we develop for the proof.

### 6.2 Neurons concentrate around teacher neurons: dual certificate analysis and test function

We focus on Lemma 8(i)(ii) here. We will use a dual certificate technique similar to Poon et al. (2023) to prove Lemma 8(i), and a more general construction of test function to prove Lemma 8(ii). In below, we consider a relaxed version of original optimization problem (2) by allowing infinite number of neurons, i.e., distribution of neurons, with $\sigma_{\geq 2}(x) = \text{ReLU}(x) - 1/\sqrt{2\pi} - x/2$ instead of ReLU:

$$\min_{\mu \in \mathcal{M}(\mathbb{S}^{d-1})} L_\lambda(\mu) := L(\mu; \sigma_{\geq 2}) + \lambda |\mu|_1, \tag{5}$$

where $\mu_\lambda^*$ is the minimizer. We use $\sigma_{\geq 2}$ activation because this is the effective activation when linear terms $\alpha, \beta$ are perfectly fitted (remove 0th and 1st order Hermite expansion of ReLU, see Claim B.1 and (6) in appendix).

This is the loss function we would have in the idealized setting: (1) linear term $\alpha, \beta$ reach their global minima (this is easy to achieve as loss is convex in them); (2) use $\ell_1$ regularization instead of $\ell_2$ regularization, since this is the case when the first and second layer norm are balanced (weight decay encourages this to happen). Note that the results in this part can handle almost all activation as long as its Hermite expansion is well-defined, generalizing Zhou et al. (2021) that can only handle absolute/ReLU activation. In below we will focus on the activation $\sigma_{\geq 2}$ for simplicity.

**Dual certificate** This optimization problem (5) can be viewed as a natural extension of the classical compressed sensing problem (Donoho, 2006; Candès et al., 2006) and Lasso-type problem (Tibshirani, 1996) in the infinite dimensional space, which has been studied in recent years (Bach, 2017; Poon et al., 2023). One common way is to study its dual problem. The dual solution $p_0(\boldsymbol{x})$ (maps $\mathbb{R}^d$ to $\mathbb{R}$) of (5) when $\lambda = 0$ satisfies $\mathbb{E}_{\boldsymbol{x}}[p_0(\boldsymbol{x})\sigma_{\geq 2}(\boldsymbol{w}^\top \boldsymbol{x})] \in \partial |\mu_*|(\mathbb{S}^{d-1})$ (more detailed discussions on this dual problem can be found in e.g., Poon et al. (2023)). Here $\eta(\boldsymbol{w}) = \mathbb{E}_{\boldsymbol{x}}[p(\boldsymbol{x})\sigma_{\geq 2}(\boldsymbol{w}^\top \boldsymbol{x})]$ is often called dual certificate, as it serves as a certificate of whether a solution $\mu$ is optimal. Its meaning will be clear in the discussions below.

We now introduce the notion of non-degenerate dual certificate, motivated by Poon et al. (2023). Note that the condition $\eta(\boldsymbol{w}) \in \partial|\mu_*|(\mathbb{S}^{d-1})$ implies that $\eta(\boldsymbol{w}_i^*) = \text{sign}(a_i^*)$ and $\|\eta\|_\infty \leq 1$. The following definition is a slightly stronger version of the above implications as it requires $\eta$ to decay at least quadratic when moves away from $\boldsymbol{w}_i^*$.

**Definition 1** (Non-degenerate dual certificate). *$\eta(\boldsymbol{w})$ is called a non-degenerate dual certificate if there exists $p(\boldsymbol{x})$ such that $\eta(\boldsymbol{w}) = \mathbb{E}_{\boldsymbol{x}}[p(\boldsymbol{x})\sigma_{\geq 2}(\boldsymbol{w}^\top \boldsymbol{x})]$ for $\boldsymbol{w} \in \mathbb{S}^{d-1}$ and*

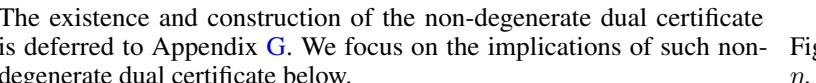

*(i) $\eta(\boldsymbol{w}_i^*) = \text{sign}(a_i^*)$ for $i = 1, \ldots, m_*$.*

*(ii) $|\eta(\boldsymbol{w})| \leq 1 - \rho_\eta \delta(\boldsymbol{w}, \boldsymbol{w}_i^*)^2$ if $\boldsymbol{w} \in \mathcal{T}_i$, where $\delta(\boldsymbol{w}, \boldsymbol{w}_i^*) = \angle(\boldsymbol{w}, \boldsymbol{w}_i^*)$.*

The existence and construction of the non-degenerate dual certificate is deferred to Appendix G. We focus on the implications of such non-degenerate dual certificate below.

Figure 2: Dual certificate $\eta$.

Roughly speaking, the dual certificate only focuses on the position of ground-truth directions $\boldsymbol{w}_i^*$ as it decays fast when moving away from these directions (Figure 2). Thus, if $\mu$ exactly recovers ground-truth $\mu_*$, then we have $\langle \eta, \mu_* \rangle = |\mu_*|_1$. The gap between $\langle \eta, \mu \rangle$ and $|\mu|_1$ is large when $\mu$ is away from $\mu_*$. Therefore, $\eta$ can be viewed as a certificate to test the optimality of $\mu$. The lemma below makes it more precise.

**Lemma 9.** *Given a non-degenerate dual certificate $\eta$, then*

*(i) $\langle \eta, \mu^* \rangle = |\mu^*|_1$ and $|\langle \eta, \mu - \mu^* \rangle| \leq \|p\|_2 \sqrt{L(\mu)}$.*

*(ii) For any measure $\mu \in \mathcal{M}(\mathbb{S}^{d-1})$, $|\langle \eta, \mu \rangle| \leq |\mu|_1 - \rho_\eta \sum_{i \in [m_*]} \int_{\mathcal{T}_i} \delta(\boldsymbol{w}, \boldsymbol{w}_i^*)^2 \, \mathrm{d}|\mu|(\boldsymbol{w})$.*

In the finite width case, we have $\sum_{i \in [m_*]} \int_{\mathcal{T}_i} \delta(\boldsymbol{w}, \boldsymbol{w}_i^*)^2 \, \mathrm{d}|\mu|(\boldsymbol{w}) = \sum_i |a_i| \, \|\boldsymbol{w}_i\| \, \delta_i^2$. This is exactly the quantity that we are interested in Lemma 8.

To see the usefulness of Lemma 9, we show a proof for total norm bound of the optimal solution $\mu_\lambda^*$. The proof for general $\mu$ with optimality gap $\zeta$ is similar (Lemma F.5).

**Claim 1** (Lemma 8(i) for $\mu_\lambda^*$). $\sum_{i \in [m_*]} \int_{\mathcal{T}_i} \delta(\boldsymbol{w}, \boldsymbol{w}_i^*)^2 \, \mathrm{d}|\mu_\lambda^*|(\boldsymbol{w}) \leq O_*(\lambda)$

*Proof.* It is not hard to show $|\mu_\lambda^*|_1 \leq |\mu^*|_1$ (Lemma F.3) so we have

$$|\mu_\lambda^*|_1 - |\mu^*|_1 - \langle \eta, \mu_\lambda^* - \mu^* \rangle \leq -\langle \eta, \mu_\lambda^* - \mu^* \rangle.$$

Using Lemma 9 and the fact $L(\mu_\lambda^*) = O_*(\lambda^2)$ from Lemma F.3,

$$\text{LHS} = |\mu_\lambda^*|_1 - \langle \eta, \mu_\lambda^* \rangle \geq \rho_\eta \sum_{i \in [m_*]} \int_{\mathcal{T}_i} \delta(\boldsymbol{w}, \boldsymbol{w}_i^*)^2 \, \mathrm{d}|\mu_\lambda^*|(\boldsymbol{w}), \quad \text{RHS} \leq \|p\|_2 \sqrt{L(\mu_\lambda^*)} = O_*(\lambda).$$

We have $\sum_{i \in [m_*]} \int_{\mathcal{T}_i} \delta(\boldsymbol{w}, \boldsymbol{w}_i^*)^2 \, \mathrm{d}|\mu_\lambda^*|(\boldsymbol{w}) = O_*(\lambda)$. $\square$

**Test function** The idea of using test function is to identify certain properties of the target function/distribution that we are interested in. Specifically, we construct test function so that it only correlates well with the target function that has the desired property. Generally speaking, the dual certificate above can be consider as a specific case of a test function: the correlation between dual certificate $\eta$ and distribution of neurons $\mu$ is large (reach $|\mu|_1$) only when $\mu \approx \mu_*$.

In below, we use this test function idea to show that every ground-truth direction has close-by neuron (Lemma 8(ii)). Denote $\mathcal{T}_i(\delta) := \{j : \angle(\boldsymbol{w}_j, \boldsymbol{w}_i) \leq \delta\} \cap \mathcal{T}_i$ as the neurons that are $\delta$-close to $\boldsymbol{w}_i^*$.

**Lemma 10** (Lemma 8(ii), informal). *Given the optimality gap $\zeta$, we have the total mass near each target direction is large, i.e., $\mu(\mathcal{T}_i(\delta)) \, \text{sign}(a_i^*) \geq |a_i^*|/2$ for all $i \in [m_*]$ and any $\delta \geq \Theta_* \left( \zeta^{1/3} \right)$.*

Note that although the results in the dual certificate part (Lemma 9(ii)) can imply that there are neurons close to teacher neurons, the bound we get here using carefully designed test function are sharper ($\zeta^{1/3}$ vs. $\zeta^{1/4}$). This is in fact important to the descent direction construction (Lemma 7).

In the proof, we view the residual $R(\boldsymbol{x}) = f_\mu(\boldsymbol{x}) - f_*(\boldsymbol{x})$ as the target function and construct test function that will only have large correlation if there is a teacher neuron that have no close student neurons. Specifically, the test function $g$ only consists of high-order Hermite polynomial such that it is large around the ground-truth direction and decays fast when moving away (Figure 3). It looks like a single spike in dual certificate $\eta$, but in fact decays much faster than $\eta$ when moving away. It is more flexible to choose test function than dual certificate, so test function $g$ can focus only on a local region of one ground-truth direction and give a better guarantee than dual certificate analysis.

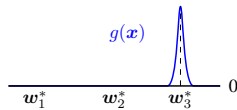

Figure 3: Test function $g$.

### 6.3  Average neuron is close to teacher neuron: residual decomposition and average neuron

We give the proof idea for Lemma 8(iii) that shows average neuron $\boldsymbol{v}_i$ is close to teacher neuron $\boldsymbol{w}_i^*$ using the residual decomposition $R = R_1 + R_2 + R_3$.

The key is to observe that $R_1$ is an analogue to exact-parametrization case where loss is often strongly-convex, so we have $\|R_1\|_2^2 = \Omega_*(1) \sum_i \|\boldsymbol{v}_i - \boldsymbol{w}_i^*\|_2^2$. Then the goal is to upper bound $\|R_1\|$. Given the decomposition $R = R_1 + R_2 + R_3$, it is easy to bound $\|R_1\| \le \|R\| + \|R_2\| + \|R_3\|$. We focus on $\|R_2\|$ as the other two are not hard to bound (loss is small in local regime). $R_2$ is in fact closely related with the total weighted norm bound in Lemma 8: we show $\|R_2\| = O_*(1) \left( \sum_{j \in \mathcal{T}_i} |a_j| \|\boldsymbol{w}_j\|_2 \delta_j^2 \right)^{3/2} = O_*((\zeta/\lambda)^{3/2})$. Thus, we get a bound for $\|\boldsymbol{v}_i - \boldsymbol{w}_i^*\|$. See Appendix F.1.4 for details.

## 7  Conclusion

In this paper we showed that gradient descent converges in a large local region depending on the complexity of the teacher network, and the local convergence allows 2-layer networks to perform a strong notion of feature learning (matching the directions of ground-truth teacher networks). We hope our result gives a better understanding of why gradient-based training is important for feature learning in neural networks. Our results rely on adding standard weight decay and new constructions of dual certificate and test functions, which can be helpful in understanding local optimization landscape in other problems. A natural but challenging next step is to understand whether the intermediate steps are also important for feature learning.

## Acknowledgement

Rong Ge and Mo Zhou are supported by NSF Award DMS-2031849 and CCF-1845171 (CAREER).

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

# A  Some properties of Hermite polynomials

In this section, we give several properties of Hermite polynomials that are useful in our analysis. See O'Donnell (2021) for a more complete discussion on Hermite polynomials. Let $H_k$ be the probabilists' Hermite polynomial where

$$H_k(x) = (-1)^k e^{x^2/2} \frac{\mathrm{d}^k}{\mathrm{d}x^k}(e^{-x^2/2})$$

and $h_k = \frac{1}{\sqrt{k!}} H_k$ be the normalized Hermite polynomials.

Hermite polynomials are classical orthogonal polynomials, which means $\mathbb{E}_{x\sim N(0,1)}[h_m(x)h_n(x)] = 1$ if $m = n$ and otherwise 0. Given a function $\sigma$, we call $\sigma(x) = \sum_{k=0}^{\infty} \hat{\sigma}_k h_k(x)$ as the Hermit expansion of $\sigma$ and $\hat{\sigma}_k = \mathbb{E}_{x\sim N(0,1)}[\sigma(x)h_k(x)]$ as the $k$-th Hermite coefficient of $\sigma$.

The following is a useful property of Hermite polynomial.

**Claim A.1** ((O'Donnell, 2021), Section 11.2)*. Let $(x, y)$ be $\rho$-correlated standard normal variables (that is, both $x, y$ have marginal distribution $N(0,1)$ and $\mathbb{E}[xy] = \rho$). Then, $\mathbb{E}[h_m(x)h_n(y)] = \rho^n \delta_{mn}$, where $\delta_{mn} = 1$ if $m = n$ and otherwise 0.*

The following lemma gives the Hermite coefficients for absolute value function and ReLU.

**Lemma A.1.** *Let $\hat{\sigma}_k = \mathbb{E}_{x\sim N(0,1)}[\sigma(x)h_k(x)]$ be the Hermite coefficient of $\sigma$. For $\sigma$ is ReLU or absolute function, we have $|\hat{\sigma}_k| = \Theta(k^{-5/4})$.*

*Proof.* From Goel et al. (2020); Zhou et al. (2021) we have

$$\hat{\sigma}_{abs,k} = \begin{cases} 0 & , k \text{ is odd} \\ \sqrt{2/\pi} & , k = 0 \\ (-1)^{\frac{k}{2}-1}\sqrt{\frac{2}{\pi}}\frac{(k-2)!}{\sqrt{k!}2^{k/2-1}(k/2-1)!} & , k \text{ is even and } k \geq 2 \end{cases}$$

$$\hat{\sigma}_{relu,k} = \begin{cases} 0 & , k \text{ is odd and } k \geq 3 \\ \sqrt{1/2\pi} & , k = 0 \\ 1/2 & , k = 1 \\ (-1)^{\frac{k}{2}-1}\sqrt{\frac{1}{2\pi}}\frac{(k-2)!}{\sqrt{k!}2^{k/2-1}(k/2-1)!} & , k \text{ is even and } k \geq 2 \end{cases}$$

Using Stirling's formula, we get $|\hat{\sigma}_{abs,k}|, |\hat{\sigma}_{relu,k}| = \Theta(k^{-5/4})$. $\qquad\square$

# B  Useful facts and proof of Theorem 2

In this section we provide several useful facts and present the proof of Theorem 2.

The following claim shows that the square loss can be decomposed into 3 terms, where $\alpha, \beta$ are corresponding to 0th and 1st order of Hermite expansion. The effective activation is in fact $\sigma_{\geq 2}$ as defined below.

**Claim B.1.** *Denote $\hat{\alpha} = -(1/\sqrt{2\pi})\sum_{i=1}^{m} a_i \|\boldsymbol{w}_i\|_2$, $\hat{\boldsymbol{\beta}} = -(1/2)\sum_{i=1}^{m} a_i \boldsymbol{w}_i$. We have square loss*

$$L(\boldsymbol{\theta}) = |\alpha - \hat{\alpha}|^2 + \left\|\boldsymbol{\beta} - \hat{\boldsymbol{\beta}}\right\|_2^2 + \mathbb{E}_{\boldsymbol{x}}[(f_{\geq 2}(\boldsymbol{x}) - \widetilde{f}_*(\boldsymbol{x}))^2]$$

*where $f_{\geq 2}(\boldsymbol{x}; \boldsymbol{\theta}) = \sum_{i\in[m]} a_i \sigma_{\geq 2}(\boldsymbol{w}_i^\top \boldsymbol{x})$ and $\sigma_{\geq 2}(x) = \sigma(x) - 1/\sqrt{2\pi} - x/2$ is the activation that after removing 0th and 1st order term in Hermite expansion.*

*As a result, when $\alpha, \beta$ are perfectly fitted and norms are balanced we have*

$$L_\lambda(\boldsymbol{\theta}) = \mathbb{E}_{\boldsymbol{x}}[(f_{\geq 2}(\boldsymbol{x}) - \widetilde{f}_*(\boldsymbol{x}))^2] + \lambda \sum_{i\in[m]} .|a_i| \|\boldsymbol{w}_i\|_2$$

*Proof.* Following Ge et al. (2018), we can write the loss $L(\boldsymbol{\theta})$ as a sum of tensor decomposition problem using Hermite expansion as in Section A (recall $\|\boldsymbol{w}_i^*\|_2 = 1$ and preprocessing procedure removes the 0-th and 1-st order term in the Hermite expansion of $\sigma$):

$$
\begin{aligned}
L(\boldsymbol{\theta}) =& \mathbb{E}_{\boldsymbol{x}} \left[ \left( \sum_{i \in [m]} a_i \|\boldsymbol{w}_i\|_2 \sum_{k \geq 0} \hat{\sigma}_k h_k(\overline{\boldsymbol{w}}_i^\top \boldsymbol{x}) + \alpha + h_1(\boldsymbol{\beta}^\top \boldsymbol{x}) - \sum_{i \in [m_*]} a_i^* \|\boldsymbol{w}_i^*\|_2 \sum_{k \geq 2} \hat{\sigma}_k h_k(\boldsymbol{w}_i^{*\top} \boldsymbol{x}) \right)^2 \right] \\
=& \left| \alpha + \hat{\sigma}_0 \sum_{i \in [m]} a_i \|\boldsymbol{w}_i\|_2 \right|^2 + \left\| \boldsymbol{\beta} + \hat{\sigma}_1 \sum_{i \in [m]} a_i \boldsymbol{w}_i \right\|_2^2 \\
& + \sum_{k \geq 2} \hat{\sigma}_k^2 \left\| \sum_{i \in [m]} a_i \|\boldsymbol{w}_i\|_2 \overline{\boldsymbol{w}}_i^{\otimes k} - \sum_{i \in [m_*]} a_i^* \|\boldsymbol{w}_i^*\|_2 \boldsymbol{w}_i^{*\otimes k} \right\|_F^2.
\end{aligned}
$$

Note that $\hat{\sigma}_0 = 1/\sqrt{2\pi}$, $\hat{\sigma}_1 = 1/2$ as in Lemma A.1, we get the result. $\qquad\square$

The proof of main result Theorem 2 is simply a combination of few lemmas appear in other sections. We refer the detailed proof and discussion to their corresponding sections.

**Theorem 2** (Main result). *Under Assumption 1, 2, 3, consider Algorithm 1 on loss (2). There exists a schedule of weight decay $\lambda_t$ and step size $\eta_t$ such that given $m \geq m_0 = \widetilde{O}_*(1) \cdot (1/\varepsilon_0)^{O(r)}$ neurons with small enough $\varepsilon_0 = \Theta_*(1)$, with high probability we will recover the target network $L(\boldsymbol{\theta}) \leq \varepsilon$ within time $T = O_*(1/\eta\varepsilon^2)$ where $\eta = \text{poly}(\varepsilon, 1/d, 1/m)$.*

*Moreover, when $\varepsilon \to 0$ every student neuron $\boldsymbol{w}_i$ either aligns with one of teacher neuron $\boldsymbol{w}_j^*$ as $\angle(\boldsymbol{w}_i, \boldsymbol{w}_j^*) = 0$ or vanishes as $|a_i| = \|\boldsymbol{w}_i\| = 0$.*

*Proof.* Combine Lemma 3 (Stage 1), Lemma 4 (Stage 2) and Lemma 5 (Stage 3) together and follow the choice of $\lambda_t$ and $\eta_t$ we get the result.

For the student neurons' alignment, it is a direct corollary from Lemma F.6 and Lemma F.5. $\qquad\square$

## C    Stage 1: first gradient step

In this section, we show that after the first gradient update the first layer weights $\boldsymbol{w}_1, \ldots, \boldsymbol{w}_m$ form a $\varepsilon_0$-net of the target subspace $S_*$, given $m = (1/\varepsilon_0)^{O(r)}$ neurons. The proof is deferred to Section C.1.

**Lemma 3** (Stage 1). *Under Assumption 1,2,3, consider Algorithm 1 with $\lambda_0 = \eta_0 = 1$ and $m \geq m_0 = \widetilde{O}_*(1) \cdot (1/\varepsilon_0)^{O(r)}$ with any $\varepsilon_0 = \Theta_*(1)$. After first step, with probability $1 - \delta$ we have*

*(i) for every teacher neuron $\boldsymbol{w}_i^*$, there exists at least one student neuron $\boldsymbol{w}_j$ s.t. $\angle(\boldsymbol{w}_i^*, \boldsymbol{w}_j) \leq \varepsilon_0$.*

*(ii) $\left\| \boldsymbol{w}_i^{(1)} \right\|_2 = \Theta_*(1)$, $|a_i^{(1)}| \leq O_*(1/\sqrt{m})$ for all $i \in [m_*]$, $\alpha_1 = 0$ and $\boldsymbol{\beta}_1 = \boldsymbol{0}$.*

The proof relies on the following lemma from Damian et al. (2022) that shows after the first step update $\boldsymbol{w}_i$'s are located at positions as if they are sampled within the target subspace $S_*$.

**Lemma C.1** (Lemma 4, Damian et al. (2022)). *Under Assumption 3, we have with high probability in the $\ell_2$ norm sense*

$$
\boldsymbol{w}_i^{(1)} = -\eta_0 \nabla_{\boldsymbol{w}_i} L(\boldsymbol{a}^{(0)}, \boldsymbol{W}^{(0)}) = -2\eta_0 a_i^{(0)} \left( \hat{\sigma}_2^2 \boldsymbol{H} \overline{\boldsymbol{w}}_i \pm \widetilde{O}(\frac{\sqrt{r}}{d}) \right),
$$

*where $\hat{\sigma}_k := \mathbb{E}_{\boldsymbol{x}}[\sigma(\boldsymbol{x}) h_k(\boldsymbol{x})]$ is the k-th Hermite polynomial coefficient.*

### C.1    Proofs in Section C

We now are ready to give the proof of Lemma 3.

**Lemma 3** (Stage 1). *Under Assumption 1,2,3, consider Algorithm 1 with $\lambda_0 = \eta_0 = 1$ and $m \geq m_0 = \widetilde{O}_*(1) \cdot (1/\varepsilon_0)^{O(r)}$ with any $\varepsilon_0 = \Theta_*(1)$. After first step, with probability $1 - \delta$ we have*

*(i) for every teacher neuron $\boldsymbol{w}_i^*$, there exists at least one student neuron $\boldsymbol{w}_j$ s.t. $\angle(\boldsymbol{w}_i^*, \boldsymbol{w}_j) \leq \varepsilon_0$.*

*(ii) $\left\| \boldsymbol{w}_i^{(1)} \right\|_2 = \Theta_*(1)$, $|a_i^{(1)}| \leq O_*(1/\sqrt{m})$ for all $i \in [m_*]$, $\alpha_1 = 0$ and $\boldsymbol{\beta}_1 = \boldsymbol{0}$.*

*Proof.* We show them one by one.

**Part (i)** From Lemma C.1 and the fact that $\overline{\boldsymbol{w}}_i^{(0)}$ samples uniformly from unit sphere, we know the probability of $\angle(\overline{\boldsymbol{w}}_i^{(1)}, \boldsymbol{w})$ for any given $\boldsymbol{w}$ is at least $\Omega_*(\varepsilon_0^r)$. Applying union bound we get the desired result.

**Part (ii)** We have

$$\boldsymbol{w}_i^{(1)} = -\eta_0 \nabla_{\boldsymbol{w}_i} L(\boldsymbol{a}^{(0)}, \boldsymbol{W}^{(0)}) = a_i^{(0)} \mathbb{E}_{\boldsymbol{x}}[\widetilde{f}_*(\boldsymbol{x})\sigma'(\boldsymbol{w}_i^\top \boldsymbol{x})\boldsymbol{x}]$$

For the norm bound, using Lemma C.1 we know

$$\sqrt{d} \left( \left\| \boldsymbol{H}\overline{\boldsymbol{w}}_i^{(0)} \right\|_2 - \widetilde{O}(\frac{\sqrt{r}}{d}) \right) \leq \left\| \boldsymbol{w}_i^{(1)} \right\|_2 \leq \sqrt{d} \left( \left\| \boldsymbol{H}\overline{\boldsymbol{w}}_i^{(0)} \right\|_2 + \widetilde{O}(\frac{\sqrt{r}}{d}) \right).$$

Since $\boldsymbol{w}_i^{(0)}$ initializes from Gaussian distribution, we know the desired bound hold. Similarly, one can bound $|a_i^{(1)}|$.

Since we use a symmetric initialization and have preprocessed the data, it is easy to see $\alpha, \boldsymbol{\beta}$ remains at 0.

$\square$

# D   Stage 2: reaching low loss

In Stage 2, we show that given the features learned in Stage 1 one can adjust the norms on top of it to reach low loss that enters the local convergence regime in Stage 3.

**Procedure** We first specify the procedure to solve $\min_{\boldsymbol{a}} \min_{\alpha, \boldsymbol{\beta}} L(\boldsymbol{\theta}) + \lambda \sum_i \|\boldsymbol{w}_i\|_2 |a_i|$. For $\boldsymbol{a}$ at current point, we first solve the inner optimization problem, which is a linear regression on $\alpha, \boldsymbol{\beta}$. From Claim B.1 we know the global minima is $(\hat{\alpha}, \hat{\boldsymbol{\beta}})$. For simplicity of the proof, we just directly set $(\alpha, \boldsymbol{\beta}) = (\hat{\alpha}, \hat{\boldsymbol{\beta}})$. Then given the $\alpha, \boldsymbol{\beta}$, the outer optimization is a convex optimization for $\boldsymbol{a}$, which can also be solved efficiently. Specifically, we perform 1 step of (sub)gradient on the loss function. We repeat the above 2 steps until convergence.

From Claim B.1 we know the actual objective that we optimize is

$$\widetilde{L}_{1,\lambda}(\boldsymbol{a}) = \mathbb{E}_{\boldsymbol{x}}[(\boldsymbol{a}^\top \sigma_{\geq 2}(\boldsymbol{W}\boldsymbol{x}) - \widetilde{y})^2] + \lambda \sum_i \|\boldsymbol{w}_i\|_2 |a_i|.$$

The following lemma shows that after Stage 2 we reach a low loss solution given the first layer features learned after first gradient step. The proof requires $\eta$ to be small enough that depends on $1/m$, mostly due to the large gradient norm. We believe using more advance algorithm for this type of problem can alleviate this issue. However, as this is not the focus of this paper, we omit it for simplicity.

**Lemma 4** (Stage 2). *Under Assumption 1,2,3, consider Algorithm 1 with $\lambda_t = \sqrt{\varepsilon_0}$. Given Stage 1 in Lemma 3, we have Stage 2 ends within time $T_2 = \widetilde{O}_*(1/\eta\varepsilon_0)$ such that optimality gap $\zeta_{T_2} = O_*(\varepsilon_0)$.*

*Proof.* Denote $\widetilde{\boldsymbol{a}}_*$ as the minima of $\widetilde{L}_{1,\lambda}$. Then, we have

$$
\begin{aligned}
\left\|\boldsymbol{a}^{(t+1)} - \widetilde{\boldsymbol{a}}_*\right\|_2^2 &= \left\|\boldsymbol{a}^{(t)} - \widetilde{\boldsymbol{a}}_*\right\|_2^2 - 2\eta\langle\nabla_{\boldsymbol{a}}\widetilde{L}_{1,\lambda}(\boldsymbol{a}^{(t)}), \boldsymbol{a}^{(t)} - \widetilde{\boldsymbol{a}}_*\rangle + \eta^2\left\|\nabla_{\boldsymbol{a}}\widetilde{L}_{1,\lambda}(\boldsymbol{a}^{(t)})\right\|_2^2 \\
&\overset{(a)}{\leq} \left\|\boldsymbol{a}^{(t)} - \widetilde{\boldsymbol{a}}_*\right\|_2^2 - 2\eta(\widetilde{L}_{1,\lambda}(\boldsymbol{a}^{(t)}) - \widetilde{L}_{1,\lambda}(\widetilde{\boldsymbol{a}}_*)) + \eta^2 O_*(m) \\
&= \left\|\boldsymbol{a}^{(t)} - \widetilde{\boldsymbol{a}}_*\right\|_2^2 - 2\eta(\widetilde{L}_{1,\lambda}(\boldsymbol{a}^{(t)}) - \widetilde{L}_{1,\lambda}(\widetilde{\boldsymbol{a}}_*)) + \eta\varepsilon_0/2,
\end{aligned}
$$

where (a) we use idea loss $\widetilde{L}_{1,\lambda}$ is convex in $\boldsymbol{a}$.

Iterating the above inequality over all $t$ we have

$$
\left\|\boldsymbol{a}^{(T)} - \widetilde{\boldsymbol{a}}_*\right\|_2^2 \leq \left\|\boldsymbol{a}^{(1)} - \widetilde{\boldsymbol{a}}_*\right\|_2^2 - 2\eta\sum_{t\leq T}(\widetilde{L}_{1,\lambda}(\boldsymbol{a}^{(t)}) - \widetilde{L}_{1,\lambda}(\widetilde{\boldsymbol{a}}_*)) + \eta T\varepsilon_0/2,
$$

which means

$$
\min_{t\leq T}\widetilde{L}_{1,\lambda}(\boldsymbol{a}^{(t)}) - \widetilde{L}_{1,\lambda}(\widetilde{\boldsymbol{a}}_*) \leq \frac{1}{T}\sum_{t\leq T}(\widetilde{L}_{1,\lambda}(\boldsymbol{a}^{(t)}) - \widetilde{L}_{1,\lambda}(\widetilde{\boldsymbol{a}}_*)) \leq \frac{\left\|\boldsymbol{a}^{(1)} - \widetilde{\boldsymbol{a}}_*\right\|_2^2}{\eta T} + \varepsilon_0/2.
$$

It is easy to see $\left\|\boldsymbol{a}^{(1)}\right\|_2, \|\widetilde{\boldsymbol{a}}_*\|_1 = O_*(1)$. Thus, when $T \geq O_*(1/\eta\varepsilon_0)$ we know $\widetilde{L}_{1,\lambda}(\boldsymbol{a}^{(T_2)}) - \widetilde{L}_{1,\lambda}(\widetilde{\boldsymbol{a}}_*) \leq 3\varepsilon_0/4$.

This suggests the optimality gap after balancing the norm (so that $L_\lambda(\boldsymbol{\theta}^{(T_2)}) = \widetilde{L}_{1,\lambda}(\boldsymbol{a}^{(T_2)})$)

$$
\begin{aligned}
\zeta_{T_2} &= L_\lambda(\boldsymbol{\theta}^{(T_2)}) - \min_{\mu\in\mathcal{M}(\mathbb{S}^{d-1})} L_\lambda(\mu) \\
&= \widetilde{L}_{1,\lambda}(\boldsymbol{a}^{(T_2)}) - \widetilde{L}_{1,\lambda}(\widetilde{\boldsymbol{a}}_*) + \widetilde{L}_{1,\lambda}(\widetilde{\boldsymbol{a}}_*) - \min_{\mu\in\mathcal{M}(\mathbb{S}^{d-1})} L_\lambda(\mu).
\end{aligned}
$$

For $\widetilde{L}_{1,\lambda}(\boldsymbol{a}^{(T_2)}) - \widetilde{L}_{1,\lambda}(\widetilde{\boldsymbol{a}}_*)$, we just show above that it is less than $3\varepsilon_0/4$.

For $\widetilde{L}_{1,\lambda}(\widetilde{\boldsymbol{a}}_*) - \min_{\mu\in\mathcal{M}(\mathbb{S}^{d-1})} L_\lambda(\mu)$, we have

$$
\begin{aligned}
\widetilde{L}_{1,\lambda}(\widetilde{\boldsymbol{a}}_*) - \min_{\mu\in\mathcal{M}(\mathbb{S}^{d-1})} L_\lambda(\mu) &\leq \widetilde{L}_{1,\lambda}(\hat{\boldsymbol{a}}_*) - \min_{\mu\in\mathcal{M}(\mathbb{S}^{d-1})} L_\lambda(\mu) \\
&\leq O_*(\varepsilon_0^2) + \lambda\left\|\boldsymbol{a}_*\right\|_1 - \lambda|\mu_\lambda^*|_1 \leq O_*(\lambda^2),
\end{aligned}
$$

where in the last inequality we use Lemma F.3 and $\mu_\lambda^* = \arg\min_{\mu\in\mathcal{M}(\mathbb{S}^{d-1})} L_\lambda(\mu)$. Here $\hat{\boldsymbol{a}}_*$ is a rescaled version of $\boldsymbol{a}_*$ and is constructed as: for every teacher neuron $\boldsymbol{w}_i^*$ choose the closest neuron $\boldsymbol{w}_j$ s.t. $\angle(\boldsymbol{w}_j, \boldsymbol{w}_i^*) \leq \varepsilon_0$ and set $\hat{a}_{*,j} = a_i^*/\left\|\boldsymbol{w}_j\right\|_2$. Set all other $\hat{a}_{*,k} = 0$.

Together with above calculations, we have $\zeta_{T_2} \leq O_*(\varepsilon_0)$. $\qquad\square$

# E  Stage 3: local convergence for regularized 2-layer neural networks

In this section we show the local convergence that loss eventually goes to 0 within polynomial time and recovers teacher neurons' direction.

The results in this section only need the width $m \geq m_*$ as long as its initial loss is small.

**Lemma 5** (Stage 3). *Under Assumption 1,2,3, consider Algorithm 1 on loss (2). Given Stage 2 in Lemma 4, if the initial optimality gap $\zeta_{3,0} \leq O_*(\lambda_{3,0}^{9/5})$, weight decay $\lambda$ follows the schedule of initial value $\lambda_{3,0} = O_*(1)$, and $k$-th epoch $\lambda_{3,k} = \lambda_{3,k-1}/2$ and stepsize $\eta_{3k} = \eta \leq O_*(\lambda_{3,k}^{12}d^{-3})$ for all $T_{3,k} \leq t \leq T_{3,k+1}$ in epoch $k$, then within $K = O_*(\log(1/\varepsilon))$ epochs and total $T_3 - T_2 = O_*(\lambda_{3,0}^{-4}\eta^{-1}\varepsilon^{-2})$ time we recover the ground-truth network $L(\boldsymbol{\theta}) \leq \varepsilon$.*

The goal of each epoch is to minimize the loss $L_\lambda$ with a fix $\lambda$. The lemma below shows that as long as the initial optimality gap is $O_*(\lambda^{9/5})$, then at the end of each epoch, $L_\lambda$ could decrease to $O_*(\lambda^2)$. Therefore, using a slow decay of weight decay parameter $\lambda$ for each epoch we could stay in the local convergence regime for each epoch and eventually recovers the target network.

**Lemma E.1** (Loss improve within one epoch). *Suppose $|a_i^{(0)}| \leq \left\|\boldsymbol{w}_i^{(0)}\right\|_2$ for all $i \in [m]$. If $\zeta_0 \leq O_*(\lambda^{9/5})$ and $\lambda \leq O_*(1)$ and $\eta \leq O_*(\lambda^{12}d^{-3})$, then within $O_*(\lambda^{-4}\eta^{-1})$ time the optimality gap becomes $L_\lambda - L_\lambda(\mu_\lambda^*) = O_*(\lambda^2)$.*

The above result relies on the following characterization of local landscape of regularized loss. We show the gradient is large whenever the optimality gap is large. This is the main contribution of this paper, see Section F for detailed proofs.

**Lemma 6** (Gradient lower bound). *When $\Omega_*(\lambda^2) \leq \zeta \leq O_*(\lambda^{9/5})$ and $\lambda \leq O_*(1)$, we have*

$$\|\nabla_{\boldsymbol{\theta}} L_\lambda\|_F^2 \geq \Omega_*(\zeta^4/\lambda^2).$$

In order to use the above landscape result with standard descent lemma, we also need certain smoothness condition on the loss function. We show below that this regularized loss indeed satisfies certain smoothness condition (though weaker than standard smoothness condition) to allow the convergence analysis.

**Lemma E.2** (Smoothness). *Suppose $|a_i| \leq \|\boldsymbol{w}_i\|_2$ and $\left\|\mathbb{E}_{\boldsymbol{x}}[R(\boldsymbol{x})\sigma'(\overline{\boldsymbol{w}}_i^{(t)\top}\boldsymbol{x})\boldsymbol{x}]\right\|_2^2 = O_*(d)$ for all $i \in [m]$. If $\eta = O_*(1/d)$, then*

$$L_\lambda(\boldsymbol{\theta} - \eta\nabla_{\boldsymbol{\theta}} L_\lambda) \leq L_\lambda(\boldsymbol{\theta}) - \eta\|\nabla_{\boldsymbol{\theta}} L_\lambda\|_F^2 + O_*(\eta^{3/2}d^{3/2})$$

## E.1 Proofs in Section E

We now are ready to show the convergence of Stage 3 by using Lemma E.1 to show the loss makes progress every epoch.

**Lemma 5** (Stage 3). *Under Assumption 1,2,3, consider Algorithm 1 on loss (2). Given Stage 2 in Lemma 4, if the initial optimality gap $\zeta_{3,0} \leq O_*(\lambda_{3,0}^{9/5})$, weight decay $\lambda$ follows the schedule of initial value $\lambda_{3,0} = O_*(1)$, and $k$-th epoch $\lambda_{3,k} = \lambda_{3,k-1}/2$ and stepsize $\eta_{3k} = \eta \leq O_*(\lambda_{3,k}^{12}d^{-3})$ for all $T_{3,k} \leq t \leq T_{3,k+1}$ in epoch $k$, then within $K = O_*(\log(1/\varepsilon))$ epochs and total $T_3 - T_2 = O_*(\lambda_{3,0}^{-4}\eta^{-1}\varepsilon^{-2})$ time we recover the ground-truth network $L(\boldsymbol{\theta}) \leq \varepsilon$.*

*Proof.* Since $|a_i^{(0)}| \leq \left\|\boldsymbol{w}_i^{(0)}\right\|_2$ for all $i \in [m]$ at the beginning of Stage 3, from Lemma E.3 we know they will remain hold for all epoch and all time $t$.

From Lemma E.1 we know for epoch $k$ it finishes within $O_*(\lambda_k^{-4}\eta^{-1})$ time and achieves $L_{\lambda_k} - L_{\lambda_k}(\mu_{\lambda_k}^*) = O_*(\lambda_k^2)$. To proceed to next epoch $k+1$, we only need to show the solution at the end of epoch $k$ $\boldsymbol{\theta}^{(k)}$ gives the optimality gap $\zeta = O_*(\lambda_{k+1}^{9/5})$ for the next $\lambda_{k+1}$. We have

$$
\begin{aligned}
L_{\lambda_{k+1}}(\boldsymbol{\theta}^{(k)}) - L_{\lambda_{k+1}}(\mu_{\lambda_{k+1}}^*) =& L(\boldsymbol{\theta}^{(k)}) - L(\mu_{\lambda_{k+1}}^*) + \frac{\lambda_{k+1}}{2}\left\|\boldsymbol{a}^{(k)}\right\|_2^2 + \frac{\lambda_{k+1}}{2}\left\|\boldsymbol{W}^{(k)}\right\|_F^2 - \lambda_{k+1}|\mu_{\lambda_{k+1}}^*|_1 \\
\overset{(a)}{\leq}& O_*(\lambda_k^2) + \frac{\lambda_{k+1}}{\lambda_k}\left(\frac{\lambda_k}{2}\left\|\boldsymbol{a}^{(k)}\right\|_2^2 + \frac{\lambda_k}{2}\left\|\boldsymbol{W}^{(k)}\right\|_F^2 - \lambda_k|\mu_{\lambda_{k+1}}^*|_1\right) \\
\overset{(b)}{\leq}& O_*(\lambda_k^2) + \frac{\lambda_{k+1}}{\lambda_k}\left(O_*(\lambda_k^2) + L(\mu_{\lambda_k}^*) - L(\boldsymbol{\theta}^{(k)})\right) \\
\overset{(c)}{\leq}& O_*(\lambda_k^2) \leq O_*(\lambda_{k+1}^{9/5})
\end{aligned}
$$

where (a) due to Lemma F.4 that $L(\boldsymbol{\theta}^{(k)})$ is small; (b) the optimality gap at the end of epoch $k$ is $O_*(\lambda_k^2)$ and $|\mu_{\lambda_k}^*|_1 - |\mu_{\lambda_{k+1}}^*|_1 = O_*(\lambda_k)$ from Lemma F.3; (c) due to Lemma F.3 that $L(\mu_{\lambda_k}^*)$ is small. In this way, we can apply Lemma E.1 again for epoch $k+1$.

From Lemma F.4 we know at the end of epoch $k$ the square loss $L(\boldsymbol{\theta}^{(k)}) = O_*(\lambda_k^2)$. Thus, to reach $\varepsilon$ square loss, we need $\lambda_k = O_*(\varepsilon^{1/2})$, which means we need to take $O_*(\log(1/\varepsilon))$ epoch. Since epoch $k$ it finishes within $O_*(\lambda_k^{-4}\eta^{-1})$ time, we know the total time is at most $O_*(\lambda_0^{-4}\eta^{-1}\varepsilon^{-2})$ time. $\qquad\square$

To show the lemma below that loss makes progress within every epoch, we rely on the gradient lower bound (Lemma 6) and smoothness condition of loss function (Lemma E.2).

**Lemma E.1** (Loss improve within one epoch). *Suppose $|a_i^{(0)}| \leq \left\|\boldsymbol{w}_i^{(0)}\right\|_2$ for all $i \in [m]$. If $\zeta_0 \leq O_*(\lambda^{9/5})$ and $\lambda \leq O_*(1)$ and $\eta \leq O_*(\lambda^{12}d^{-3})$, then within $O_*(\lambda^{-4}\eta^{-1})$ time the optimality gap becomes $L_\lambda - L_\lambda(\mu_\lambda^*) = O_*(\lambda^2)$.*

*Proof.* Since $|a_i^{(0)}| \leq \left\|\boldsymbol{w}_i^{(0)}\right\|_2$ for all $i \in [m]$ at the beginning of current epoch, from Lemma E.3 we know they will remain hold for all time $t$. Then combine Lemma E.4 and Lemma E.2 we know

$$L_\lambda(\boldsymbol{\theta} - \eta\nabla_{\boldsymbol{\theta}}L_\lambda) \leq L_\lambda(\boldsymbol{\theta}) - \eta\left\|\nabla_{\boldsymbol{\theta}}L_\lambda\right\|_F^2 + O_*(\eta^{3/2}d^{3/2}).$$

Recall $\zeta_t = L_\lambda(\boldsymbol{\theta}^{(t)}) - L_\lambda(\mu_\lambda^*)$. Using gradient lower bound Lemma 6 and consider the time before $\zeta_t$ reach $O_*(\lambda^2)$ we have

$$\zeta_{t+1} \leq \zeta_t - \eta\Omega_*(\zeta_t^4/\lambda^2) + O_*(\eta^{3/2}d^{3/2}) \leq \zeta_t - \Omega_*(\eta\zeta_t^4/\lambda^2),$$

where we use $\eta = O_*(\lambda^{12}d^{-3})$ to be small enough.

The above recursion implies that

$$\zeta_t = O_*((t/\lambda^2 + \zeta_0^{-3})^{-1/3}).$$

Thus, within $O_*(1/\lambda^4)$ the optimality gap $\zeta_t$ reaches $O_*(\lambda^2)$. $\qquad\square$

The lemma below shows a regularity condition on the norm between two layers.

**Lemma E.3.** *If we start at $|a_i^{(0)}| \leq \left\|\boldsymbol{w}_i^{(0)}\right\|_2$ and $\eta = O_*(1)$, then we have $|a_i^{(t)}|^2 \leq \left\|\boldsymbol{w}_i^{(t)}\right\|_2^2$ for all $i \in [m_*]$ and all time $t$.*

*Proof.* Denote $R(\boldsymbol{x}) = f(\boldsymbol{x}) - f_*(\boldsymbol{x})$. Assume $|a_i^{(t)}|^2 - \left\|\boldsymbol{w}_i^{(t)}\right\|_2^2 \leq 0$ we show it remains at $t+1$. We have

$$|a_i^{(t+1)}|^2 - \left\|\boldsymbol{w}_i^{(t+1)}\right\|_2^2$$
$$=|a_i^{(t)} - \eta\nabla_{a_i}L_\lambda(\boldsymbol{\theta}^{(t)})|^2 - \left\|\boldsymbol{w}_i^{(t)} - \eta\nabla_{\boldsymbol{w}_i}L_\lambda(\boldsymbol{\theta}^{(t)})\right\|_2^2$$
$$=|a_i^{(t)}|^2 - \left\|\boldsymbol{w}_i^{(t)}\right\|_2^2 + \eta^2|\nabla_{a_i}L_\lambda(\boldsymbol{\theta}^{(t)})|^2 - \eta^2\left\|\nabla_{\boldsymbol{w}_i}L_\lambda(\boldsymbol{\theta}^{(t)})\right\|_2^2$$
$$=|a_i^{(t)}|^2 - \left\|\boldsymbol{w}_i^{(t)}\right\|_2^2 + \eta^2|2\mathbb{E}_{\boldsymbol{x}}[R(\boldsymbol{x})\sigma(\boldsymbol{w}_i^{(t)\top}\boldsymbol{x})] + \lambda a_i^{(t)}|^2 - \eta^2\left\|2\mathbb{E}_{\boldsymbol{x}}[R(\boldsymbol{x})a_i^{(t)}\sigma'(\overline{\boldsymbol{w}}_i^{(t)\top}\boldsymbol{x})\boldsymbol{x}] + \lambda\boldsymbol{w}_i^{(t)}\right\|_2^2$$

We first focus on the last 2 terms. We have

$$|2\mathbb{E}_{\boldsymbol{x}}[R(\boldsymbol{x})\sigma(\boldsymbol{w}_i^{(t)\top}\boldsymbol{x})] + \lambda a_i^{(t)}|^2 - \left\|2\mathbb{E}_{\boldsymbol{x}}[R(\boldsymbol{x})a_i^{(t)}\sigma'(\overline{\boldsymbol{w}}_i^{(t)\top}\boldsymbol{x})\boldsymbol{x}] + \lambda\boldsymbol{w}_i^{(t)}\right\|_2^2$$
$$= \left\|\boldsymbol{w}_i^{(t)}\right\|_2^2 |2\mathbb{E}_{\boldsymbol{x}}[R(\boldsymbol{x})\sigma(\overline{\boldsymbol{w}}_i^{(t)\top}\boldsymbol{x})]|^2 + \lambda^2|a_i^{(t)}|^2 - |a_i^{(t)}|^2\left\|2\mathbb{E}_{\boldsymbol{x}}[R(\boldsymbol{x})\sigma'(\overline{\boldsymbol{w}}_i^{(t)\top}\boldsymbol{x})\boldsymbol{x}]\right\|_2^2 - \lambda^2\left\|\boldsymbol{w}_i^{(t)}\right\|_2^2$$
$$\overset{(a)}{\leq} \left(|a_i^{(t)}|^2 - \left\|\boldsymbol{w}_i^{(t)}\right\|_2^2\right)\left(\lambda^2 - \left\|2\mathbb{E}_{\boldsymbol{x}}[R(\boldsymbol{x})\sigma'(\overline{\boldsymbol{w}}_i^{(t)\top}\boldsymbol{x})\boldsymbol{x}]\right\|_2^2\right),$$

where (a) due to $|2\mathbb{E}_{\boldsymbol{x}}[R(\boldsymbol{x})\sigma(\overline{\boldsymbol{w}}_i^{(t)\top}\boldsymbol{x})]|^2 \leq \left\|2\mathbb{E}_{\boldsymbol{x}}[R(\boldsymbol{x})\sigma'(\overline{\boldsymbol{w}}_i^{(t)\top}\boldsymbol{x})\boldsymbol{x}]\right\|_2^2$.

Therefore, plug it back to the above equation, we have

$$|a_i^{(t+1)}|^2 - \left\|\boldsymbol{w}_i^{(t+1)}\right\|_2^2 \leq \left(|a_i^{(t)}|^2 - \left\|\boldsymbol{w}_i^{(t)}\right\|_2^2\right)\left(1 + \eta^2\lambda^2 - \eta^2\left\|2\mathbb{E}_{\boldsymbol{x}}[R(\boldsymbol{x})\sigma'(\overline{\boldsymbol{w}}_i^{(t)\top}\boldsymbol{x})\boldsymbol{x}]\right\|_2^2\right)$$
$$\overset{(a)}{\leq} 0,$$

where (a) due to $|a_i^{(t)}|^2 - \left\|\boldsymbol{w}_i^{(t)}\right\|_2^2 \le 0$ and we use $\left\|2\mathbb{E}_{\boldsymbol{x}}[R(\boldsymbol{x})\sigma'(\overline{\boldsymbol{w}}_i^{(t)\top}\boldsymbol{x})]\boldsymbol{x}\right\|_2^2 = O_*(d)$ from Lemma E.4 and $\eta$ is small enough.

Therefore, we can see that $|a_i^{(t)}|^2 - \left\|\boldsymbol{w}_i^{(t)}\right\|_2^2 \le 0$ remains for all $t$. $\qquad\square$

This lemma shows the smoothness of loss function. The proof requires a careful calculations to bound the error terms.

**Lemma E.2** (Smoothness). *Suppose* $|a_i| \le \|\boldsymbol{w}_i\|_2$ *and* $\left\|\mathbb{E}_{\boldsymbol{x}}[R(\boldsymbol{x})\sigma'(\overline{\boldsymbol{w}}_i^{(t)\top}\boldsymbol{x})\boldsymbol{x}]\right\|_2^2 = O_*(d)$ *for all* $i \in [m]$. *If* $\eta = O_*(1/d)$, *then*

$$L_\lambda(\boldsymbol{\theta} - \eta\nabla_{\boldsymbol{\theta}}L_\lambda) \le L_\lambda(\boldsymbol{\theta}) - \eta\left\|\nabla_{\boldsymbol{\theta}}L_\lambda\right\|_F^2 + O_*(\eta^{3/2}d^{3/2})$$

*Proof.* Denote $R_{\boldsymbol{\theta}}(\boldsymbol{x}) = f_{\boldsymbol{\theta}}(\boldsymbol{x}) - f_*(\boldsymbol{x})$ to denote the dependency on $\boldsymbol{\theta}$. For simplicity, we will use $\widetilde{\nabla}_{\boldsymbol{\theta}} = -\eta\nabla_{\boldsymbol{\theta}}L_\lambda$ and same for others. Since $\left\|\mathbb{E}_{\boldsymbol{x}}[R(\boldsymbol{x})\sigma'(\overline{\boldsymbol{w}}_i^{(t)\top}\boldsymbol{x})\boldsymbol{x}]\right\|_2^2 = O_*(d)$, we know $|\widetilde{\nabla}_{a_i}| = O_*(\eta\|\boldsymbol{w}_i\|_2\, d)$ and $\left\|\widetilde{\nabla}_{\boldsymbol{w}_i}\right\|_2 = O_*(\eta|a_i|d)$

We have

$$L_\lambda(\boldsymbol{\theta} - \eta\nabla_{\boldsymbol{\theta}}) - L_\lambda(\boldsymbol{\theta}) + \eta\left\|\nabla_{\boldsymbol{\theta}}\right\|_F^2$$
$$=L_\lambda(\boldsymbol{\theta} - \eta\nabla_{\boldsymbol{\theta}}) - L_\lambda(\boldsymbol{\theta}) - \langle\nabla_{\boldsymbol{\theta}}, -\eta\nabla_{\boldsymbol{\theta}}\rangle$$
$$=\mathbb{E}_{\boldsymbol{x}}[R_{\boldsymbol{\theta}+\widetilde{\nabla}_{\boldsymbol{\theta}}}(\boldsymbol{x})^2] + \frac{\lambda}{2}\left\|\boldsymbol{a} + \widetilde{\nabla}_{\boldsymbol{a}}\right\|_2^2 + \frac{\lambda}{2}\left\|\boldsymbol{W} + \widetilde{\nabla}_{\boldsymbol{W}}\right\|_F^2 - \mathbb{E}_{\boldsymbol{x}}[R_{\boldsymbol{\theta}}(\boldsymbol{x})^2] - \frac{\lambda}{2}\|\boldsymbol{a}\|_2^2 - \frac{\lambda}{2}\|\boldsymbol{W}\|_F^2$$
$$\quad - \sum_{i\in[m]}\mathbb{E}_{\boldsymbol{x}}[R_{\boldsymbol{\theta}}(\boldsymbol{x})\sigma(\boldsymbol{w}_i^\top\boldsymbol{x})\widetilde{\nabla}_{a_i}] - \sum_{i\in[m]}\mathbb{E}_{\boldsymbol{x}}[R_{\boldsymbol{\theta}}(\boldsymbol{x})a_i\sigma'(\boldsymbol{w}_i^\top\boldsymbol{x})\boldsymbol{x}^\top\widetilde{\nabla}_{\boldsymbol{w}_i}] - \mathbb{E}_{\boldsymbol{x}}[R_{\boldsymbol{\theta}}(\boldsymbol{x})\widetilde{\nabla}_\alpha] - \mathbb{E}_{\boldsymbol{x}}[R_{\boldsymbol{\theta}}(\boldsymbol{x})\boldsymbol{x}^\top\widetilde{\nabla}_{\boldsymbol{\beta}}]$$
$$\quad - \lambda\langle\boldsymbol{a}, \widetilde{\nabla}_{\boldsymbol{a}}\rangle - \lambda\langle\boldsymbol{W}, \widetilde{\nabla}_{\boldsymbol{W}}\rangle$$
$$=\underbrace{\mathbb{E}_{\boldsymbol{x}}[(R_{\boldsymbol{\theta}+\widetilde{\nabla}_{\boldsymbol{\theta}}}(\boldsymbol{x}) - R_{\boldsymbol{\theta}}(\boldsymbol{x}))^2]}_{(I)}$$

$$+ 2\mathbb{E}_{\boldsymbol{x}}\underbrace{\left[R_{\boldsymbol{\theta}}(\boldsymbol{x})\left(R_{\boldsymbol{\theta}+\widetilde{\nabla}_{\boldsymbol{\theta}}}(\boldsymbol{x}) - R_{\boldsymbol{\theta}}(\boldsymbol{x}) - \sum_{i\in[m]}\sigma(\boldsymbol{w}_i^\top\boldsymbol{x})\widetilde{\nabla}_{a_i} - \sum_{i\in[m]}a_i\sigma'(\boldsymbol{w}_i^\top\boldsymbol{x})\boldsymbol{x}^\top\widetilde{\nabla}_{\boldsymbol{w}_i} - \widetilde{\nabla}_\alpha - \boldsymbol{x}^\top\widetilde{\nabla}_{\boldsymbol{\beta}}\right)\right]}_{(II)}$$

$$+ \frac{\lambda}{2}\left\|\widetilde{\nabla}_{\boldsymbol{a}}\right\|_2^2 + \frac{\lambda}{2}\left\|\widetilde{\nabla}_{\boldsymbol{W}}\right\|_F^2.$$

The last line is easy to see on $O_*(\eta^2 d^2)$ using norm bound in Lemma F.12, so in below we are going to bound (I) and (II) one by one. The goal is to show they are small in the sense of on order $o(\eta)$.

**Bound (I)** For (I), we can write out the expression as

$$\mathbb{E}_{\boldsymbol{x}}[(R_{\boldsymbol{\theta}+\widetilde{\nabla}_{\boldsymbol{\theta}}}(\boldsymbol{x}) - R_{\boldsymbol{\theta}}(\boldsymbol{x}))^2] = \mathbb{E}_{\boldsymbol{x}}\left[\left(\sum_{i\in[m]}(a_i + \widetilde{\nabla}_{a_i})\sigma((\boldsymbol{w}_i + \widetilde{\nabla}_{\boldsymbol{w}_i})^\top\boldsymbol{x}) - a_i\sigma(\boldsymbol{w}_i^\top\boldsymbol{x}) + \widetilde{\nabla}_\alpha + \boldsymbol{x}^\top\widetilde{\nabla}_{\boldsymbol{\beta}}\right)^2\right]$$

$$\le 2\mathbb{E}_{\boldsymbol{x}}\underbrace{\left[\left(\sum_{i\in[m]}(a_i + \widetilde{\nabla}_{a_i})\sigma((\boldsymbol{w}_i + \widetilde{\nabla}_{\boldsymbol{w}_i})^\top\boldsymbol{x}) - a_i\sigma(\boldsymbol{w}_i^\top\boldsymbol{x})\right)^2\right]}_{(I.i)}$$

$$+ 2\mathbb{E}_{\boldsymbol{x}}\underbrace{\left[\left(\widetilde{\nabla}_\alpha + \boldsymbol{x}^\top\widetilde{\nabla}_{\boldsymbol{\beta}}\right)^2\right]}_{(I.ii)}$$

For (I.i), we can split into 2 terms as

$$\mathbb{E}_{\boldsymbol{x}}\left[\left(\sum_{i\in[m]}(a_i+\widetilde{\nabla}_{a_i})\sigma((\boldsymbol{w}_i+\widetilde{\nabla}_{\boldsymbol{w}_i})^\top\boldsymbol{x})-a_i\sigma(\boldsymbol{w}_i^\top\boldsymbol{x})\right)^2\right]$$

$$\leq 2\mathbb{E}_{\boldsymbol{x}}\left[\left(\sum_{i\in[m]}\widetilde{\nabla}_{a_i}\sigma((\boldsymbol{w}_i+\widetilde{\nabla}_{\boldsymbol{w}_i})^\top\boldsymbol{x})\right)^2\right]+2\mathbb{E}_{\boldsymbol{x}}\left[\left(\sum_{i\in[m]}a_i\sigma((\boldsymbol{w}_i+\widetilde{\nabla}_{\boldsymbol{w}_i})^\top\boldsymbol{x})-a_i\sigma(\boldsymbol{w}_i^\top\boldsymbol{x})\right)^2\right]$$

$$\leq 2\mathbb{E}_{\boldsymbol{x}}\left[\left(\sum_{i\in[m]}|\widetilde{\nabla}_{a_i}||(\boldsymbol{w}_i+\widetilde{\nabla}_{\boldsymbol{w}_i})^\top\boldsymbol{x}|\right)^2\right]+2\mathbb{E}_{\boldsymbol{x}}\left[\left(\sum_{i\in[m]}|a_i||\widetilde{\nabla}_{\boldsymbol{w}_i}^\top\boldsymbol{x}|\right)^2\right].$$

We then can bound them separately as

$$(I.i)\overset{(a)}{\leq}O(1)\left(\sum_{i\in[m]}|\widetilde{\nabla}_{a_i}|\left\|\boldsymbol{w}_i+\widetilde{\nabla}_{\boldsymbol{w}_i}\right\|_2\right)^2+O(1)\left(\sum_{i\in[m]}|a_i|\left\|\widetilde{\nabla}_{\boldsymbol{w}_i}\right\|_2\right)^2$$

$$\overset{(b)}{\leq}O_*(d^2)\left(\sum_{i\in[m]}\eta\left\|\boldsymbol{w}_i\right\|_2^2+\eta^2|a_i|\left\|\boldsymbol{w}_i\right\|_2 d\right)^2+O_*(d^2)\left(\sum_{i\in[m]}\eta a_i^2\right)^2$$

$$\overset{(c)}{\leq}O_*(\eta^2 d^2),$$

where (a) we use Lemma E.5; (b) recall $|\widetilde{\nabla}_{a_i}|=O_*(\eta\left\|\boldsymbol{w}_i\right\|_2 d)$ and $\left\|\widetilde{\nabla}_{\boldsymbol{w}_i}\right\|_2=O_*(\eta|a_i|d)$; (c) $\left\|\boldsymbol{a}\right\|,\left\|\boldsymbol{W}\right\|_F,\sum_{i\in[m]}|a_i|\left\|\boldsymbol{w}_i\right\|_2=O_*(1)$ from Lemma F.12 and Lemma F.4, as well as $\eta$ is small enough.

For (I.ii), we have

$$\mathbb{E}_{\boldsymbol{x}}\left[\left(\widetilde{\nabla}_\alpha+\boldsymbol{x}^\top\widetilde{\nabla}_{\boldsymbol{\beta}}\right)^2\right]\leq O(|\widetilde{\nabla}_\alpha|^2+\left\|\widetilde{\nabla}_{\boldsymbol{\beta}}\right\|_2^2)=O_*(\eta^2 d^2),$$

where we use Lemma F.4.

Combine (I.i) and (I.ii) we know (I)$=O_*(\eta^2 d^2)$.

**Bound (II)** For (II), we have

$$\mathbb{E}_{\boldsymbol{x}}\left[R_{\boldsymbol{\theta}}(\boldsymbol{x})\left(R_{\boldsymbol{\theta}+\widetilde{\nabla}_{\boldsymbol{\theta}}}(\boldsymbol{x})-R_{\boldsymbol{\theta}}(\boldsymbol{x})-\sum_{i\in[m]}\sigma(\boldsymbol{w}_i^\top\boldsymbol{x})\widetilde{\nabla}_{a_i}-\sum_{i\in[m]}a_i\sigma'(\boldsymbol{w}_i^\top\boldsymbol{x})\boldsymbol{x}^\top\widetilde{\nabla}_{\boldsymbol{w}_i}-\widetilde{\nabla}_\alpha-\boldsymbol{x}^\top\widetilde{\nabla}_{\boldsymbol{\beta}}\right)\right]$$

$$=\mathbb{E}_{\boldsymbol{x}}\left[R_{\boldsymbol{\theta}}(\boldsymbol{x})\left(\sum_{i\in[m]}\underbrace{(a_i+\widetilde{\nabla}_{a_i})\sigma((\boldsymbol{w}_i+\widetilde{\nabla}_{\boldsymbol{w}_i})^\top\boldsymbol{x})-a_i\sigma(\boldsymbol{w}_i^\top\boldsymbol{x})-\sigma(\boldsymbol{w}_i^\top\boldsymbol{x})\widetilde{\nabla}_{a_i}-a_i\sigma'(\boldsymbol{w}_i^\top\boldsymbol{x})\boldsymbol{x}^\top\widetilde{\nabla}_{\boldsymbol{w}_i}}_{I_i(\boldsymbol{x})}\right)\right]$$

$$\leq\sum_{i\in[m]}\|R_{\boldsymbol{\theta}}\|\|I_i\|$$

We focus on bound $\|I_i\|$ below. The goal is to show it is $o(\eta)$. For $I_i(\boldsymbol{x})$, we have

$$\|I_i\|_2^2=\mathbb{E}_{\boldsymbol{x}}\left[\left((a_i+\widetilde{\nabla}_{a_i})\sigma((\boldsymbol{w}_i+\widetilde{\nabla}_{\boldsymbol{w}_i})^\top\boldsymbol{x})-a_i\sigma(\boldsymbol{w}_i^\top\boldsymbol{x})-\sigma(\boldsymbol{w}_i^\top\boldsymbol{x})\widetilde{\nabla}_{a_i}-a_i\sigma'(\boldsymbol{w}_i^\top\boldsymbol{x})\boldsymbol{x}^\top\widetilde{\nabla}_{\boldsymbol{w}_i}\right)^2\right]$$

$$\leq\mathbb{E}_{\boldsymbol{x}}\left[2\left(\widetilde{\nabla}_{a_i}(\sigma((\boldsymbol{w}_i+\widetilde{\nabla}_{\boldsymbol{w}_i})^\top\boldsymbol{x})-\sigma(\boldsymbol{w}_i^\top\boldsymbol{x}))\right)^2+2\left(a_i(\sigma((\boldsymbol{w}_i+\widetilde{\nabla}_{\boldsymbol{w}_i})^\top\boldsymbol{x})-\sigma(\boldsymbol{w}_i^\top\boldsymbol{x})-\sigma'(\boldsymbol{w}_i^\top\boldsymbol{x})\boldsymbol{x}^\top\widetilde{\nabla}_{\boldsymbol{w}_i})\right)^2\right]$$

$$\leq 2\underbrace{\mathbb{E}_{\boldsymbol{x}}\left[|\widetilde{\nabla}_{a_i}|^2|\widetilde{\nabla}_{\boldsymbol{w}_i}^\top\boldsymbol{x}|^2\right]}_{(II.i)}+2a_i^2\underbrace{\mathbb{E}_{\boldsymbol{x}}\left[|(\boldsymbol{w}_i+\widetilde{\nabla}_{\boldsymbol{w}_i})^\top\boldsymbol{x}|^2(\sigma'((\boldsymbol{w}_i+\widetilde{\nabla}_{\boldsymbol{w}_i})^\top\boldsymbol{x})-\sigma'(\boldsymbol{w}_i^\top\boldsymbol{x}))^2\right]}_{(II.ii)}$$

For (II.i), recall $|\widetilde{\nabla}_{a_i}| = O_*(\eta \|\boldsymbol{w}_i\|_2 d)$ and $\left\|\widetilde{\nabla}_{\boldsymbol{w}_i}\right\|_2 = O_*(\eta|a_i|d)$ we have

$$\mathbb{E}_{\boldsymbol{x}}\left[|\widetilde{\nabla}_{a_i}|^2 |\widetilde{\nabla}_{\boldsymbol{w}_i}^\top \boldsymbol{x}|^2\right] \leq |\widetilde{\nabla}_{a_i}|^2 \left\|\widetilde{\nabla}_{\boldsymbol{w}_i}\right\|^2 = O_*(\eta^4 |a_i|^2 \|\boldsymbol{w}_i\|_2^2 d^4).$$

For (II.ii), we have

$$\mathbb{E}_{\boldsymbol{x}}\left[|\boldsymbol{w}_i + \widetilde{\nabla}_{\boldsymbol{w}_i}^\top \boldsymbol{x}|^2 (\sigma'((\boldsymbol{w}_i + \widetilde{\nabla}_{\boldsymbol{w}_i})^\top \boldsymbol{x}) - \sigma'(\boldsymbol{w}_i^\top \boldsymbol{x}))^2\right]$$

$$=\mathbb{E}_{\boldsymbol{x}}\left[|(\boldsymbol{w}_i + \widetilde{\nabla}_{\boldsymbol{w}_i})^\top \boldsymbol{x}|^2 \mathbb{1}_{\text{sign}((\boldsymbol{w}_i + \widetilde{\nabla}_{\boldsymbol{w}_i})^\top \boldsymbol{x}) \neq \text{sign}(\boldsymbol{w}_i^\top \boldsymbol{x})}\right]$$

$$\leq O(\left\|\boldsymbol{w}_i + \widetilde{\nabla}_{\boldsymbol{w}_i}\right\|_2^2 \delta^3),$$

where $\delta = \angle(\boldsymbol{w}_i + \widetilde{\nabla}_{\boldsymbol{w}_i}, \boldsymbol{w}_i)$ is the angle between $\boldsymbol{w}_i + \widetilde{\nabla}_{\boldsymbol{w}_i}$ and $\boldsymbol{w}_i$. Since $\left\|\widetilde{\nabla}_{\boldsymbol{w}_i}\right\|_2 = O_*(\eta|a_i|d) = O_*(\eta \|\boldsymbol{w}_i\|_2 d)$, we know $\delta = O(\left\|\widetilde{\nabla}_{\boldsymbol{w}_i}\right\|)$ given $\eta = O_*(1/d)$ to be small enough.

Combine (II.i) and (II.ii) we have

$$\|I_i\|_2^2 \leq O_*(\eta^4 a_i^2 \|\boldsymbol{w}_i\|_2^2 d^4) + O(a_i^2 \left\|\boldsymbol{w}_i + \widetilde{\nabla}_{\boldsymbol{w}_i}\right\|_2^2 \left\|\widetilde{\nabla}_{\boldsymbol{w}_i}\right\|_2^3) \leq O_*(\eta^3 a_i^2 \|\boldsymbol{w}_i\|_2^2 d^3).$$

Since $\|R_{\boldsymbol{\theta}}\| = O_*(1)$, this implies

$$(II) \leq \sum_{i \in [m]} O_*(\eta^{3/2} a_i \|\boldsymbol{w}_i\|_2 d^{3/2}) = O_*(\eta^{3/2} d^{3/2}).$$

**Combine (I)(II)**   Finally, combing (I) and (II) we have

$$L_\lambda(\boldsymbol{\theta} - \eta\nabla_{\boldsymbol{\theta}}) - L_\lambda(\boldsymbol{\theta}) + \eta \|\nabla_{\boldsymbol{\theta}}\|_F^2 = O_*(\eta^{3/2} d^{3/2}).$$

Going back to the beginning of this proof, we get the desired result. $\qquad\square$

### E.2   Technical Lemma

We present technical lemmas that are used in the proof of this section. They mostly follow from direct calculations.

**Lemma E.4.** *We have* $\left\|\mathbb{E}_{\boldsymbol{x}}[R(\boldsymbol{x})\sigma'(\overline{\boldsymbol{w}}_i^{(t)\top} \boldsymbol{x})\boldsymbol{x}]\right\|_2^2 = O_*(d)$

*Proof.* It is easy to see given $\|R\| = O_*(1)$.

$\qquad\square$

**Lemma E.5** (Lemma D.4 in Zhou et al. (2021)). *Consider* $\alpha_i \in \mathbb{R}^d$ *for* $i \in [n]$. *We have*

$$\mathbb{E}_{x \sim N(0,I)}\left[\left(\sum_{i=1}^n |\alpha_i^\top x|\right)^2\right] \leq c_0 \left(\sum_{i=1}^n \|\alpha_i\|\right)^2,$$

*where $c_0$ is a constant.*

## F   Local landscape of population loss

In this section, we are going to show Lemma 6 that characterizing the population local landscape with a fixed $\lambda$ by giving the lower bound of gradient.

**Outline** We generally follow the high-level proof plan that outlines in Section 6. In Section F.1 and Section F.2, we characterize the local geometry as in Lemma 8. Then, we use it to construct descent direction in Section F.3. Finally we give the proof of Lemma 6 in Section F.4.

We start by identifying the structure of (approximated) solution of a closely-related problem in Section F.1 (rewrite (5)):

$$\min_{\mu \in \mathcal{M}(\mathbb{S}^{d-1})} L_\lambda(\mu) := L(\mu) + \lambda |\mu|_1 := \mathbb{E}_{\boldsymbol{x}, \widetilde{y}}[(f_\mu(\boldsymbol{x}) - \widetilde{y})^2] + \lambda |\mu|_1 \tag{6}$$

$$= \mathbb{E}_{\boldsymbol{x}} \left[ \left( \int_{\boldsymbol{w}} \sigma_{\geq 2}(\boldsymbol{w}^\top \boldsymbol{x}) \mathrm{d}\, \mu - \mu_* \right)^2 \right] + \lambda |\mu|_1, \tag{7}$$

where $\mathcal{M}(\mathbb{S}^{d-1})$ is the measure space over unit sphere $\mathbb{S}^{d-1}$, $\mu_* = \sum_{i \in [m_*]} a_i^* \delta_{\boldsymbol{w}_i^*}$ and $\sigma_{\geq 2}(x) = \sigma(x) - 1/\sqrt{2\pi} - x/2$ is the activation that after removing 0th and 1st order term in Hermite expansion. Note that when $\mu$ represents a finite-wdith network, we have $\mu = \sum_{i \in [m]} a_i \|\boldsymbol{w}_i\|_2 \delta_{\overline{\boldsymbol{w}}_i}$ is a empirical measure over the neurons. In particular, when $\mu = \mu_*$, model $f_\mu$ recovers the target $\widetilde{f}_*$.

We call (5) as the ideal loss because the original problem (2) would become the above (5) when we balance the norms ($\|\boldsymbol{w}_i\|_2 = |a_i|$), perfectly fit $\alpha, \beta$ and relax the finite-width constraints to allow infinite-width (see Claim B.1). This is why we slightly abused the notation to use $L_\lambda$ in both (2) and (5).

In Section F.3 we will use the solution structure to construct descent direction that are positively correlated with gradient and also handle the case when norms are not balanced or $\alpha, \beta$ are not fitted well.

**Notation** Denote the optimality gap between the loss at $\mu$ and the optimal distribution $\mu_\lambda^*$ as

$$\zeta(\mu) := L_\lambda(\mu) - L_\lambda(\mu_\lambda^*),$$

where $\mu_\lambda^*$ is the optimal measure that minimize (5). For simplicity denote $\widetilde{a}_i = a_i \|\boldsymbol{w}_i\|_2$ so that $|\mu|_1 = \|\widetilde{\boldsymbol{a}}\|_1$ when $\mu = \sum_{i \in [m]} a_i \|\boldsymbol{w}_i\|_2 \delta_{\overline{\boldsymbol{w}}_i}$. Often we use $\zeta_t = \zeta(\mu_t)$ to denote the optimality gap at time $t$ and just $\zeta$ for simplicity. We slightly abuse the notation to also use $\zeta = L_\lambda(\theta) - L_\lambda(\mu_\lambda^*)$. Finally denote $\mu^* = \sum_{i \in [m_*]} a_i^* \delta_{\boldsymbol{w}_i^*}$ (assuming $\|\boldsymbol{w}_i^*\|_2 = 1$) so that $f_{\mu^*}(\boldsymbol{x}) = \mathbb{E}_{\boldsymbol{w} \sim \mu^*}[\sigma_{\geq 2}(\boldsymbol{w}^\top \boldsymbol{x})]$.

## F.1 Structure of the ideal loss solution

In this section, we will focus on the structure of approximated solution for the $\ell_1$ regularized regression problem (5).

In the rest of this section, we will first introduce the idea of non-degenerate dual certificate and then use it as a tool to characterize the structure of the solutions. The proofs are deferred to Section H.

### F.1.1 Non-degenerate dual certificate

We first recall the definition of non-degenerate dual certificate, which is similar as in (Poon et al., 2023) but slightly adapted for fit our need.

**Definition 1** (Non-degenerate dual certificate). *$\eta(\boldsymbol{w})$ is called a non-degenerate dual certificate if there exists $p(\boldsymbol{x})$ such that $\eta(\boldsymbol{w}) = \mathbb{E}_{\boldsymbol{x}}[p(\boldsymbol{x})\sigma_{\geq 2}(\boldsymbol{w}^\top \boldsymbol{x})]$ for $\boldsymbol{w} \in \mathbb{S}^{d-1}$ and*

*(i) $\eta(\boldsymbol{w}_i^*) = \mathrm{sign}(a_i^*)$ for $i = 1, \ldots, m_*$.*

*(ii) $|\eta(\boldsymbol{w})| \leq 1 - \rho_\eta \delta(\boldsymbol{w}, \boldsymbol{w}_i^*)^2$ if $\boldsymbol{w} \in \mathcal{T}_i$, where $\delta(\boldsymbol{w}, \boldsymbol{w}_i^*) = \angle(\boldsymbol{w}, \boldsymbol{w}_i^*)$.*

We first show that there exist such non-degenerate dual certificate. More discussion and a detailed proof are deferred to Section G.

**Lemma F.1.** *There exists a non-degenerate dual certificate $\eta = \mathbb{E}_{\boldsymbol{x}}[p(\boldsymbol{x})\sigma_{\geq 2}(\boldsymbol{w}^\top \boldsymbol{x})]$ with $\rho_\eta = \Theta(1)$ and $\|p\|_2 \leq \mathrm{poly}(m_*, \Delta)$*

The following lemma (restate of Lemma 9) gives the properties that will be used in the later proofs: the non-degenerate dual certificate $\eta$ allows us to capture the gap between the current position $\mu$ and the target $\mu^*$.

**Lemma F.2.** *Given a non-degenerate dual certificate $\eta$, then*

(i) $\langle \eta, \mu^* \rangle = |\mu^*|_1$

(ii) *For any measure $\mu \in \mathcal{M}(\mathbb{S}^{d-1})$, $|\langle \eta, \mu \rangle| \leq |\mu|_1 - \rho_\eta \sum_{i \in [m_*]} \int_{\mathcal{T}_i} \delta(\boldsymbol{w}, \boldsymbol{w}_i^*)^2 \, \mathrm{d}|\mu|(\boldsymbol{w})$.*

(iii) $\langle \eta, \mu - \mu^* \rangle = \langle p, f_\mu - f_{\mu^*} \rangle$, *where $f_\mu(\boldsymbol{x}) = \mathbb{E}_{\boldsymbol{w} \sim \mu}[\sigma_{\geq 2}(\boldsymbol{w}^\top \boldsymbol{x})]$. Then $|\langle \eta, \mu - \mu^* \rangle| \leq \|p\|_2 \sqrt{L(\mu)}$.*

### F.1.2 Properties of $\mu_\lambda^*$

Given the non-degenerate dual certificate $\eta$, we now are ready to identify several useful properties of $\mu_\lambda^*$. The lemma below essentially says that $\mu_\lambda^*$ is similar to $\mu^*$ in the sense that most of the norm are concentrated in the ground-truth direction and the square loss is small. The proof relies on comparing $\mu_\lambda^*$ with $\mu^*$ using the optimality conditions.

**Lemma F.3.** *We have the following hold*

(i) $|\mu_*|_1 - \lambda \|p\|_2^2 \leq |\mu_\lambda^*|_1 \leq |\mu^*|_1 = \|\boldsymbol{a}^*\|_1$

(ii) $L(\mu_\lambda^*) \leq \lambda^2 \|p\|_2^2 = O_*(\lambda^2)$

(iii) $\sum_{i \in [m_*]} \int_{\mathcal{T}_i} \delta(\boldsymbol{w}, \boldsymbol{w}_i^*)^2 \, \mathrm{d}|\mu_\lambda^*|(\boldsymbol{w}) \leq \lambda \|p\|_2^2 / \rho_\eta = O_*(\lambda)$

### F.1.3 Properties of $\mu$ with optimality gap $\zeta$

We now characterize the structure of $\mu$ when the optimality gap is $\zeta$. The proof mostly relies on comparing $\mu$ with $\mu_\lambda^*$ and the structure of $\mu_\lambda^*$ in previous section.

The following lemma shows the square loss is bounded by the optimality gap and norms are always bounded. Note that the conditions are true under Lemma 6.

**Lemma F.4.** *Recall the optimality gap $\zeta = L_\lambda(\mu) - L_\lambda(\mu_\lambda^*)$. Then, the following holds:*

(i) $L(\mu) \leq 5\lambda^2 \|p\|^2 + 4\zeta = O_*(\lambda^2 + \zeta)$.

(ii) *if $\zeta \leq \lambda|\mu^*|_1$ and $\lambda \leq |\mu^*|_1 / \|p\|_2^2$, then $|\mu|_1 \leq 3|\mu^*|_1 = 3\|\boldsymbol{a}^*\|_1$.*

The following two lemma characterize the structure of $\mu$ using the fact that the square loss is small in previous lemma. The lemma below says that the total norm of far away neuron is small.

**Lemma F.5.** *Recall the optimality gap $\zeta = L_\lambda(\mu) - L_\lambda(\mu_\lambda^*)$. Then, we have*

$$\sum_{i \in [m_*]} \int_{\mathcal{T}_i} \delta(\boldsymbol{w}, \boldsymbol{w}_i^*)^2 \, \mathrm{d}|\mu|(\boldsymbol{w}) \leq (\zeta/\lambda + 2\lambda \|p\|_2^2)/\rho_\eta = O_*(\zeta/\lambda + \lambda).$$

*In particular, when $\mu = \sum_{i \in [m]} a_i \|\boldsymbol{w}_i\|_2 \delta_{\overline{\boldsymbol{w}}_i}$ represents finite number of neurons, we have*

$$\sum_{i \in [m_*]} \sum_{j \in \mathcal{T}_i} |a_j| \|\boldsymbol{w}_j\|_2 \delta_j^2 \leq (\zeta/\lambda + 2\lambda \|p\|_2^2)/\rho_\eta = O_*(\zeta/\lambda + \lambda),$$

*where $\delta_j = \angle(\boldsymbol{w}_j, \boldsymbol{w}_i^*)$ for $j \in \mathcal{T}_i$.*

The lemma below shows there are neurons close to the teacher neurons once the gap is small. The proof idea is similar to Section 5.3 in Zhou et al. (2021) that use test function to lower bound the loss, but now we can handle almost all activation.

**Lemma F.6.** *Under Lemma 6, if the Hermite coefficient of $\sigma$ decays as $|\hat{\sigma}_k| = \Theta(k^{-c_\sigma})$ with some constant $c_\sigma > 0$, then the total mass near each target direction is large, i.e., $\mu(\mathcal{T}_i(\delta)) \, \mathrm{sign}(a_i^*) \geq |a_i^*|/2$ for all $i \in [m_*]$ and any $\delta_{close} \geq \widetilde{\Omega}\left(\left(\frac{L(\mu)}{a_{\min}^2}\right)^{1/(4c_\sigma - 2)}\right)$ with large enough hidden constant. In particular, for $\sigma$ is ReLU or absolute function, $\delta_{close} \geq \widetilde{\Omega}\left(\left(\frac{L(\mu)}{a_{\min}^2}\right)^{1/3}\right)$. Here $a_{\min} = \min |a_i|$ is the smallest entry of $\boldsymbol{a}_*$ in absolute value.*

*As a corollary, if the optimality gap $\zeta = L_\lambda(\mu) - L_\lambda(\mu_\lambda^*)$, then $\delta_{close} \geq \widetilde{\Omega}_*\left((\zeta + \lambda^2)^{1/(4c_\sigma - 2)}\right)$ and for ReLU or absolute $\delta_{close} \geq \widetilde{\Omega}_*\left((\zeta + \lambda^2)^{1/3}\right)$.*

### F.1.4 Residual decomposition and average neuron

In this section, we introduce the residual decomposition and average neuron as in (Zhou et al., 2021) that will be used when proving the existence of descent direction.

Denote the decomposition $R(\boldsymbol{x}) = f_\mu(\boldsymbol{x}) - f_{\mu^*}(\boldsymbol{x}) = R_1(\boldsymbol{x}) + R_2(\boldsymbol{x}) + R_3(\boldsymbol{x})$ (this can be directly verified noticing that $\sigma_{\geq 2}(x) = |x|/2 - 1/\sqrt{2\pi}$),

$$
\begin{aligned}
R_1(\boldsymbol{x}) &= \frac{1}{2} \sum_{i \in [m_*]} \left( \sum_{j \in \mathcal{T}_i} a_j \boldsymbol{w}_j - \boldsymbol{w}_i^* \right)^\top \boldsymbol{x}\, \mathrm{sign}(\boldsymbol{w}_i^{*\top} \boldsymbol{x}), \\
R_2(\boldsymbol{x}) &= \frac{1}{2} \sum_{i \in [m_*]} \sum_{j \in \mathcal{T}_i} a_j \boldsymbol{w}_j^\top \boldsymbol{x}(\mathrm{sign}(\boldsymbol{w}_j^\top \boldsymbol{x}) - \mathrm{sign}(\boldsymbol{w}_i^{*\top} \boldsymbol{x})), \\
R_3(\boldsymbol{x}) &= \frac{1}{\sqrt{2\pi}} \left( \sum_{i \in [m_*]} a_i^* \|\boldsymbol{w}_i^*\|_2 - \sum_{i \in [m]} a_i \|\boldsymbol{w}_i\|_2 \right).
\end{aligned}
\tag{8}
$$

In the following we characterize $R_1, R_2, R_3$ separately. In Lemma F.7 we relate $R_1$ with the average neuron. In Lemma F.8 and Lemma F.9 we bound $R_2$ and $R_3$ respectively.

**Lemma F.7** (Zhou et al. (2021), Lemma 11). $\|R_1\|_2^2 = \Omega(\Delta^3/m_*^3) \sum_{i \in [m_*]} \left\| \sum_{j \in \mathcal{T}_i} a_j \boldsymbol{w}_j - \boldsymbol{w}_i^* \right\|_2^2.$

**Lemma F.8.** *Under Lemma 6, recall the optimality gap $\zeta = L_\lambda(\mu) - L_\lambda(\mu_\lambda^*)$. Then*

$$\|R_2\|_2^2 = O_*((\zeta/\lambda + \lambda)^{3/2}).$$

**Lemma F.9.** *Under Lemma 6 and recall the optimality gap $\zeta = L_\lambda(\mu) - L_\lambda(\mu_\lambda^*)$. If $\hat{\sigma}_0 = 0$ and $\hat{\sigma}_k > 0$ with some $k = \Theta((1/\Delta^2) \log(\zeta/\|\boldsymbol{a}_*\|_1))$, then*

$$\|R_3\|_2 = \widetilde{O}_*((\zeta + \lambda^2)^{1/2}/\hat{\sigma}_k + (\zeta/\lambda + \lambda) + \zeta).$$

Now we are ready to bound the difference between average neuron with its corresponding ground-truth neuron.

**Lemma F.10.** *Under Lemma 6, recall the optimality gap $\zeta = L_\lambda(\mu) - L_\lambda(\mu_\lambda^*)$. Then for any $i \in [m_*]$, $\zeta = \Omega(\lambda^2)$ and $\zeta, \lambda \leq 1/\mathrm{poly}(m_*, \Delta, \|\boldsymbol{a}_*\|_1)$*

$$
\left\| \sum_{j \in \mathcal{T}_i} a_j \boldsymbol{w}_j - \boldsymbol{w}_i^* \right\|_2 \leq \left( \sum_{i \in [m_*]} \left\| \sum_{j \in \mathcal{T}_i} a_j \boldsymbol{w}_j - \boldsymbol{w}_i^* \right\|_2^2 \right)^{1/2} = O_*((\zeta/\lambda)^{3/4}).
$$

### F.2 From ideal loss solution to real loss solution

In previous section, we consider the ideal loss solution that assumes the norms are perfectly balanced ($|a_i| = \|\boldsymbol{w}_i\|_2$) and $\alpha, \boldsymbol{\beta}$ are perfectly fitted. However, during the training we are not able to guarantee achieve these exactly but only approximately. This section is devoted to show that the results in previous section still hold though the conditions are only approximately satisfied. Recall that the original loss

$$L_\lambda(\boldsymbol{\theta}) = L(\boldsymbol{\theta}) + \frac{\lambda}{2} \|\boldsymbol{a}\|_2^2 + \frac{\lambda}{2} \|\boldsymbol{W}\|_F^2$$

so that when norm are balanced and $\alpha, \boldsymbol{\beta}$ are perfectly fitted, $L_\lambda(\boldsymbol{\theta}) = L(\boldsymbol{\theta}) + \lambda \sum_i |a_i| \|\boldsymbol{w}_i\|_2 = L_\lambda(\mu)$.

The lemma below shows that the properties of ideal loss solution in previous section still hold for the solution of original loss, when $\alpha, \boldsymbol{\beta}$ are approximately fitted.

**Lemma F.11.** *Given any $\boldsymbol{\theta} = (\boldsymbol{a}, \boldsymbol{W}, \alpha, \boldsymbol{\beta})$ satisfying $|\alpha - \hat{\alpha}|^2 = O(\zeta)$, $\left\|\boldsymbol{\beta} - \hat{\boldsymbol{\beta}}\right\|_2^2 = O(\zeta)$, where $\hat{\alpha} = -(1/\sqrt{2\pi}) \sum_{i=1}^m a_i \|\boldsymbol{w}_i\|_2$ and $\hat{\boldsymbol{\beta}} = -(1/2) \sum_{i=1}^m a_i \boldsymbol{w}_i$. Let its corresponding balanced*

version $\boldsymbol{\theta}_{bal} = (\boldsymbol{a}_{bal}, \boldsymbol{W}_{bal}, \alpha_{bal}, \boldsymbol{\beta}_{bal})$ as $a_{bal,i} = \text{sign}(a_i)\sqrt{|a_i|\,\|\boldsymbol{w}_i\|_2}$, $\boldsymbol{w}_{bal,i} = \overline{\boldsymbol{w}}_i\sqrt{|a_i|\,\|\boldsymbol{w}_i\|_2}$, $\alpha_{bal} = \hat{\alpha}$ and $\boldsymbol{\beta}_{bal} = \hat{\boldsymbol{\beta}}$. Then, we have

$$L_\lambda(\boldsymbol{\theta}) - L_\lambda(\boldsymbol{\theta}_{bal}) = |\alpha - \hat{\alpha}|^2 + \left\|\boldsymbol{\beta} - \hat{\boldsymbol{\beta}}\right\|_2^2 + \frac{\lambda}{2}\sum_{i\in[m]}(|a_i| - \|\boldsymbol{w}_i\|_2)^2 \geq 0.$$

*Moreover, let the optimality gap $\zeta = L_\lambda(\boldsymbol{\theta}) - L_\lambda(\mu_\lambda^*)$, we have results in Lemma F.4, Lemma F.5, Lemma F.6, Lemma F.7, Lemma F.8, Lemma F.9 and Lemma F.10 still hold for $L_\lambda(\boldsymbol{\theta})$, with the change of $R_3$ in (8) as*

$$R_3(\boldsymbol{x}) = \frac{1}{\sqrt{2\pi}}\left(\sum_{i\in[m_*]}a_i^*\,\|\boldsymbol{w}_i^*\|_2 - \sum_{i\in[m]}a_i\,\|\boldsymbol{w}_i\|_2\right) + \alpha - \hat{\alpha} + (\boldsymbol{\beta} - \hat{\boldsymbol{\beta}})^\top\boldsymbol{x}.$$

The following lemma shows the norm remains bounded.

**Lemma F.12.** *Under Lemma 6, suppose optimality gap $\zeta = L_\lambda(\boldsymbol{\theta}) - L_\lambda(\mu_\lambda^*)$. Then $\|\boldsymbol{a}\|_2^2 + \|\boldsymbol{W}\|_F^2 \leq 3\,\|\boldsymbol{a}_*\|_1$.*

### F.3 Descent direction

In this section, we show that there is a descent direction as long as the optimality gap is small until it reaches $O(\lambda^2)$. We will assume $\zeta = \Omega(\lambda^2)$ in this section for simplicity.

We first show gradient is always large whenever $\alpha, \boldsymbol{\beta}$ are not fitted well. This is a direct corollary of Claim B.1.

**Lemma F.13** (Descent direction, $\alpha$ and $\boldsymbol{\beta}$). *We have*

$$|\nabla_\alpha L_\lambda|^2 = 4(\alpha - \hat{\alpha})^2, \quad \|\nabla_{\boldsymbol{\beta}} L_\lambda\|_2^2 = 4\left\|\boldsymbol{\beta} - \hat{\boldsymbol{\beta}}\right\|_2^2.$$

Before proceeding to the following descent direction, we first make a simplification assumption that

**Assumption F.1.** *For every $\mathcal{T}_i$, for all neuron $\boldsymbol{w}_j \in \mathcal{T}_i$, assume $\boldsymbol{w}_j^\top\boldsymbol{w}_i^* \geq 0$.*

This is because due to the linear term $\boldsymbol{\beta}$, the effective activation is symmetry $\sigma_{\geq 2}(x) = \sigma_{\geq 2}(-x)$. This introduce the ambiguity of the sign of neurons. Such assumption clarifies the ambiguity of neurons' direction.

As the lemma below shows, there always exists a set of parameter (by flipping the sign of neurons) that satisfy the assumption and gives almost same gradient norm. Thus, making such assumption will not cause any issue when $\alpha, \beta$ are perfectly fitted.

**Lemma F.14.** *Suppose $(\alpha - \hat{\alpha})^2, \left\|\boldsymbol{\beta} - \hat{\boldsymbol{\beta}}\right\|_2^2 \leq \tau$ to be small enough and $\|\boldsymbol{a}\|_2, \|\boldsymbol{W}\|_F = O_*(1)$. Then, given any parameter $\boldsymbol{\theta}$, there exists another set of parameter $\widetilde{\boldsymbol{\theta}}$ that satisfies Assumption F.1 such that $f_{\boldsymbol{\theta}} = f_{\widetilde{\boldsymbol{\theta}}}$ and $|\,\|\nabla_{\boldsymbol{\theta}} L_\lambda\| - \|\nabla_{\widetilde{\boldsymbol{\theta}}} L_\lambda\|_F| \leq O_*(\sqrt{\tau})$.*

*Proof.* Denote $\boldsymbol{\theta} = (\boldsymbol{a}, \boldsymbol{w}_1, \ldots, \boldsymbol{w}_m, \alpha, \boldsymbol{\beta})$. We first construct $\widetilde{\boldsymbol{\theta}} = (\widetilde{\boldsymbol{a}}, \widetilde{\boldsymbol{w}}_1, \ldots, \widetilde{\boldsymbol{w}}_m, \widetilde{\alpha}, \widetilde{\boldsymbol{\beta}})$.

Let $\widetilde{\boldsymbol{a}} = \boldsymbol{a}$. For $\widetilde{\boldsymbol{w}}_i$, there exists such sign vector $\boldsymbol{s} = (s_1, \ldots, s_m) \in \{\pm1\}^m$ so that by flipping the sign of neurons we have $\widetilde{\boldsymbol{w}}_i = s_i\boldsymbol{w}_i$ satisfies Assumption F.1. Let $\widetilde{\alpha} = \alpha$ and $\widetilde{\boldsymbol{\beta}} = \boldsymbol{\beta} + \sum_{i:s_i=-1}a_i\boldsymbol{w}_i$.

One can verify that $f_{\boldsymbol{\theta}} = f_{\widetilde{\boldsymbol{\theta}}}$. Moreover, for the gradient of $\alpha, \boldsymbol{\beta}$ we have

$$\nabla_\alpha L_\lambda = \nabla_{\widetilde{\alpha}} L_\lambda, \nabla_{\boldsymbol{\beta}} L_\lambda = \nabla_{\widetilde{\boldsymbol{\beta}}} L_\lambda,$$

For gradient of $\boldsymbol{a}, \boldsymbol{w}_i$, when $s_i = 1$ we know they are the same. When $s_i = -1$, note that

$$\nabla_{a_i} L_\lambda - \nabla_{\widetilde{a}_i} L_\lambda = 2\mathbb{E}_{\boldsymbol{x}}[R(\boldsymbol{x})(\sigma(\boldsymbol{w}_i^\top\boldsymbol{x}) - \sigma(\widetilde{\boldsymbol{w}}_i^\top\boldsymbol{x}))] = 2(\boldsymbol{\beta} - \hat{\boldsymbol{\beta}})^\top\boldsymbol{w}_i$$

$$\nabla_{\boldsymbol{w}_i} L_\lambda + \nabla_{\widetilde{\boldsymbol{w}}_i} L_\lambda = 2a_i\mathbb{E}_{\boldsymbol{x}}[R(\boldsymbol{x})(\sigma'(\boldsymbol{w}_i^\top\boldsymbol{x}) + \sigma'(\widetilde{\boldsymbol{w}}_i^\top\boldsymbol{x}))\boldsymbol{x}] = 2a_i(\boldsymbol{\beta} - \hat{\boldsymbol{\beta}}).$$

Therefore, we get the desired result by noting the norm bound.

$\square$

We then show that if norms are not balanced or norm cancellation happens for neurons with similar direction, then one can always adjust the norm to decrease the loss due to the regularization term.

**Lemma F.15** (Descent direction, norm balance). *We have*

$$\sum_i \sum_{j \in T_i} \left| \langle \nabla_{a_j} L_\lambda, -a_j \rangle + \langle \nabla_{\boldsymbol{w}_j} L_\lambda, \boldsymbol{w}_j \rangle \right| = \lambda \sum_{i \in [m_*]} \left| a_i^2 - \|\boldsymbol{w}_i\|_2^2 \right|$$

$$\geq \max \left\{ \lambda | \|\boldsymbol{a}\|_2^2 - \|\boldsymbol{W}\|_F^2 |, \lambda \sum_{i \in [m_*]} (|a_i| - \|\boldsymbol{w}_i\|_2)^2 \right\}$$

**Lemma F.16** (Descent direction, norm cancellation). *Under Lemma 6 and Assumption F.1, suppose the optimality gap $\zeta = L_\lambda(\boldsymbol{\theta}) - L_\lambda(\mu_\lambda^*)$. For any $\boldsymbol{w}_i^*$, consider $\delta_{\text{sign}}$ such that $\delta_{close} < \delta_{\text{sign}} = O(\lambda/\zeta^{1/2})$ with small enough hidden constant ($\delta_{close}$ defined in Lemma F.6), then*

$$\sum_{s \in \{+,-\}} \sum_{j \in T_{i,s}(\delta_{\text{sign}})} \left\langle \nabla_{a_j} L_\lambda, \frac{a_j}{\sum_{j \in T_{i,s}(\delta_{\text{sign}})} |a_j| \|\boldsymbol{w}_j\|_2} \right\rangle + \left\langle \nabla_{\boldsymbol{w}_j} L_\lambda, \frac{\boldsymbol{w}_j}{\sum_{j \in T_{i,s}(\delta_{\text{sign}})} |a_j| \|\boldsymbol{w}_j\|_2} \right\rangle = \Omega(\lambda).$$

*where $T_{i,+}(\delta_{\text{sign}}) = \{j \in T_i : \delta(\boldsymbol{w}_j, \boldsymbol{w}_i^*) \leq \delta_{\text{sign}}, \text{sign}(a_j) = \text{sign}(a_i^*)\}$, $T_{i,-}(\delta_{\text{sign}}) = \{j \in T_i : \delta(\boldsymbol{w}_j, \boldsymbol{w}_i^*) \leq \delta_{\text{sign}}, \text{sign}(a_j) \neq \text{sign}(a_i^*)\}$ are the set of neurons that close to $\boldsymbol{w}_i^*$ with/without same sign of $a_i^*$.*

*As a result,*

$$\|\nabla_{\boldsymbol{a}} L_\lambda\|_2^2 + \|\nabla_{\boldsymbol{W}} L_\lambda\|_F^2 \geq \lambda^2 \sum_{j \in T_{i,-}(\delta_{\text{sign}})} |a_j| \|\boldsymbol{w}_j\|_2$$

Now given the above lemmas, it suffices to consider the remaining case that $\alpha, \boldsymbol{\beta}$ are well fitted, norms are balanced and no cancellation. In this case, the loss landscape is roughly the same as the ideal loss (5) from Lemma F.11. Thus, we could leverage these detailed characterization of the solution (far-away neurons are small and average neuron is close to corresponding ground-truth neuron) to construct descent direction.

**Lemma F.17** (Descent direction). *Under Lemma 6 and Assumption F.1, suppose the optimality gap $\zeta = L_\lambda(\boldsymbol{\theta}) - L_\lambda(\mu_\lambda^*)$. Suppose*

(i) *norms are (almost) balanced:* $| \|\boldsymbol{W}\|_F^2 - \|\boldsymbol{a}\|_2^2 | \leq \zeta/\lambda$, $\sum_{i \in [m]} (|a_j| - \|\boldsymbol{w}_j\|_2)^2 = O_*(\zeta^2/\lambda^2)$

(ii) *(almost) no norm cancellation: consider all neurons $\boldsymbol{w}_j$ that are $\delta_{\text{sign}}$-close w.r.t. teacher neuron $\boldsymbol{w}_i^*$ but has a different sign, i.e., $\text{sign}(a_j) \neq \text{sign}(a_i^*)$ with $\delta_{\text{sign}} = \Theta_*(\lambda/\zeta^{1/2})$, we have $\sum_{j \in T_{i,-}(\delta_{\text{sign}})} |a_j| \|\boldsymbol{w}_j\|_2 \leq \tau = O_*(\zeta^{5/6}/\lambda)$ with small enough hidden constant, where $T_{i,-}(\delta)$ defined in Lemma F.16.*

(iii) $\alpha, \boldsymbol{\beta}$ *are well fitted:* $|\alpha - \hat{\alpha}|^2 = O_*(\zeta)$, $\left\| \boldsymbol{\beta} - \hat{\boldsymbol{\beta}} \right\|_2^2 = O_*(\zeta)$ *with small enough hidden factor.*

*Then, we can construct the following descent direction*

$$(\alpha + \alpha_*)\nabla_\alpha L_\lambda + \langle \nabla_{\boldsymbol{\beta}} L_\lambda, \boldsymbol{\beta} + \boldsymbol{\beta}_* \rangle + \sum_{i \in [m_*]} \sum_{j \in \mathcal{T}_i} \langle \nabla_{\boldsymbol{w}_i} L_\lambda, \boldsymbol{w}_j - q_{ij} \boldsymbol{w}_i^* \rangle = \Omega(\zeta),$$

*where $q_{ij}$ satisfy the following conditions with $\delta_{close} < \delta_{\text{sign}}$ and $\delta_{close} = O_*(\zeta^{1/3})$: (1) $\sum_{j \in \mathcal{T}_i} a_j q_{ij} = a_i^*$; (2) $q_{ij} \geq 0$; (3) $q_{ij} = 0$ when $\text{sign}(a_j) \neq \text{sign}(a_i^*)$ or $\delta_j > \delta_{close}$. (4) $\sum_{i \in [m_*]} \sum_{j \in \mathcal{T}_i} q_{ij}^2 = O_*(1)$.*

## F.4 Proof of Lemma 6

Now we are ready to prove the gradient lower bound (Lemma 6) by combining all descent direction lemma in the previous section together.

**Lemma 6** (Gradient lower bound). *When $\Omega_*(\lambda^2) \leq \zeta \leq O_*(\lambda^{9/5})$ and $\lambda \leq O_*(1)$, we have*
$$\|\nabla_{\boldsymbol{\theta}} L_\lambda\|_F^2 \geq \Omega_*(\zeta^4/\lambda^2).$$

*Proof.* We check the assumption of Lemma F.17 one by one. We first assume Assumption F.1 holds to get a gradient lower bound.

For assumption (i) (norm balance) in Lemma F.17, whenever $\sum_{i\in[m_*]} \left|a_i^2 - \|\boldsymbol{w}_i\|_2^2\right| = \Omega_*(\zeta^2/\lambda^2)$, by Lemma F.15 we know
$$\sum_i \sum_{j\in T_i} \left|\langle \nabla_{a_j} L_\lambda, -a_j \rangle + \langle \nabla_{\boldsymbol{w}_j} L_\lambda, \boldsymbol{w}_j \rangle\right| \geq \Omega_*(\zeta^2/\lambda).$$

With Lemma F.12, this implies
$$\sqrt{\|\nabla_{\boldsymbol{a}} L_\lambda\|_2^2 + \|\nabla_{\boldsymbol{W}} L_\lambda\|_F^2} \cdot O(\|\boldsymbol{a}_*\|_1) \geq \sqrt{\|\nabla_{\boldsymbol{a}} L_\lambda\|_2^2 + \|\nabla_{\boldsymbol{W}} L_\lambda\|_F^2} \sqrt{\|\boldsymbol{a}\|_2^2 + \|\boldsymbol{W}\|_F^2} = \Omega_*(\zeta^2/\lambda),$$

which means
$$\|\nabla_{\boldsymbol{\theta}} L_\lambda\|_F^2 \geq \|\nabla_{\boldsymbol{a}} L_\lambda\|_2^2 + \|\nabla_{\boldsymbol{W}} L_\lambda\|_F^2 \geq \Omega_*(\zeta^4/\lambda^2)$$

For assumption (ii) (norm cancellation) in Lemma F.17, whenever it does not hold, by Lemma F.16 we know
$$\|\nabla_{\boldsymbol{\theta}} L_\lambda\|_F^2 \geq \|\nabla_{\boldsymbol{a}} L_\lambda\|_2^2 + \|\nabla_{\boldsymbol{W}} L_\lambda\|_F^2 \geq \lambda^2 \sum_{j\in T_{i,-}(\delta_{\text{sign}})} |a_j| \|\boldsymbol{w}_j\|_2 \geq \Omega_*(\zeta^{5/6}\lambda).$$

For assumption (iii) $(\alpha, \boldsymbol{\beta})$ in Lemma F.17, whenever it does not hold, by Lemma F.13 we know
$$|\nabla_\alpha L_\lambda|^2 = (\alpha - \hat{\alpha})^2 = \Omega_*(\zeta^2), \quad \|\nabla_{\boldsymbol{\beta}} L_\lambda\|_2^2 = 4\left\|\boldsymbol{\beta} - \hat{\boldsymbol{\beta}}\right\|_2^2 = \Omega_*(\zeta^2),$$

which implies
$$\|\nabla_{\boldsymbol{\theta}} L_\lambda\|_F^2 \geq |\nabla_\alpha L_\lambda|^2 + \|\nabla_{\boldsymbol{\beta}} L_\lambda\|_2^2 = \Omega_*(\zeta^2).$$

Thus, the remaining case is the one that all assumption (i)-(iii) in Lemma F.17 hold and also $\sum_{i\in[m_*]} \left|a_i^2 - \|\boldsymbol{w}_i\|_2^2\right| = O_*(\zeta^2/\lambda^2)$, we choose
$$q_{ij} = \begin{cases} \frac{a_j a_i^*}{\sum_{j\in T_{i,+}(\delta_{close})} a_j^2} & , \text{if } j \in T_{i,+}(\delta_{close}) \\ 0 & , \text{otherwise} \end{cases}$$

so that condition (1)-(4) on $q_{ij}$ all hold: condition (1)-(3) are easy to check, Lemma H.4 shows condition (4) holds. Now we know from Lemma F.17 that
$$(\alpha + \alpha_*)\nabla_\alpha L_\lambda + \langle \nabla_{\boldsymbol{\beta}} L_\lambda, \boldsymbol{\beta} + \boldsymbol{\beta}_* \rangle + \sum_{i\in[m_*]}\sum_{j\in\mathcal{T}_i} \langle \nabla_{\boldsymbol{w}_i} L_\lambda, \boldsymbol{w}_j - q_{ij}\boldsymbol{w}_i^* \rangle = \Omega(\zeta).$$

Note that
$$(\alpha + \alpha_*)\nabla_\alpha L_\lambda + \langle \nabla_{\boldsymbol{\beta}} L_\lambda, \boldsymbol{\beta} + \boldsymbol{\beta}_* \rangle + \sum_{i\in[m_*]}\sum_{j\in\mathcal{T}_i} \langle \nabla_{\boldsymbol{w}_i} L_\lambda, \boldsymbol{w}_j - q_{ij}\boldsymbol{w}_i^* \rangle$$
$$\leq \sqrt{|\nabla_\alpha L_\lambda|^2 + \|\nabla_{\boldsymbol{\beta}} L_\lambda\|_2^2 + \|\nabla_{\boldsymbol{a}} L_\lambda\|_2^2 + \|\nabla_{\boldsymbol{W}} L_\lambda\|_F^2} \sqrt{(\alpha + \alpha_*)^2 + \|\boldsymbol{\beta} + \boldsymbol{\beta}_*\|_2^2 + \sum_{i\in[m_*]}\sum_{j\in\mathcal{T}_i} \|\boldsymbol{w}_j - q_{ij}\boldsymbol{w}_i^*\|_2^2}$$

and
$$|\alpha + \alpha_*| \leq |\hat{\alpha}| + |\alpha_*| + O_*(\zeta) \overset{(a)}{\leq} O_*(1)$$
$$\|\boldsymbol{\beta} + \boldsymbol{\beta}_*\|_2 \leq \left\|\hat{\boldsymbol{\beta}}\right\|_2 + \|\boldsymbol{\beta}_*\|_2 + O_*(\zeta) \overset{(b)}{\leq} O_*(1)$$
$$\sum_{i\in[m_*]}\sum_{j\in\mathcal{T}_i} \|\boldsymbol{w}_j - q_{ij}\boldsymbol{w}_i^*\|_2^2 \leq 2\sum_{i\in[m_*]}\sum_{j\in\mathcal{T}_i} \|\boldsymbol{w}_j\|_2^2 + q_{ij}^2 \|\boldsymbol{w}_i^*\|_2^2 \overset{(c)}{\leq} O_*(1),$$

where (a)(b) by Lemma F.4; (c) we use Lemma F.12 and condition (4) on $q_{ij}$.

Therefore, we get

$$\|\nabla_{\boldsymbol{\theta}} L_\lambda\|_F^2 = |\nabla_\alpha L_\lambda|^2 + \|\nabla_{\boldsymbol{\beta}} L_\lambda\|_2^2 + \|\nabla_{\boldsymbol{a}} L_\lambda\|_2^2 + \|\nabla_{\boldsymbol{W}} L_\lambda\|_F^2 = \Omega_*(\zeta^2).$$

Combine all cases above, we know

$$\|\nabla_{\boldsymbol{a}} L_\lambda\|_2^2 + \|\nabla_{\boldsymbol{W}} L_\lambda\|_F^2 = \Omega_*(\min\{\zeta^4/\lambda^2, \zeta^{5/6}\lambda, \zeta^2\}) = \Omega_*(\zeta^4/\lambda^2),$$

as long as $\zeta = O(\lambda^{9/5}/\operatorname{poly}(r, m_*, \Delta, \|\boldsymbol{a}_*\|_1, a_{\min}))$.

We now use Lemma F.14 to show when Assumption F.1 is not true, we can get similar gradient lower bound. Denote the above gradient lower bound as $\tau_0 = \Omega_*(\zeta^4/\lambda^2)$. Let $\tau = \tau_0/2$.

When $(\alpha - \hat{\alpha})^2 \geq \tau$ or $\left\|\boldsymbol{\beta} - \hat{\boldsymbol{\beta}}\right\|_2^2 \geq \tau$, from Lemma F.13 we know $\|\nabla_{\boldsymbol{\theta}} L_\lambda\|_F^2 \geq \tau$.

When $(\alpha - \hat{\alpha})^2, \left\|\boldsymbol{\beta} - \hat{\boldsymbol{\beta}}\right\|_2^2 \leq \tau$, using Lemma F.14 we know there exists $\widetilde{\boldsymbol{\theta}}$ such that $\left\|\nabla_{\widetilde{\boldsymbol{\theta}}} L_\lambda\right\|_F^2 \geq \tau_0$ and $\left| \left\|\nabla_{\widetilde{\boldsymbol{\theta}}} L_\lambda\right\|_F - \|\nabla_{\boldsymbol{\theta}} L_\lambda\|_F \right| \leq \sqrt{\tau}$. Thus, we know $\|\nabla_{\boldsymbol{\theta}} L_\lambda\|_F^2 \geq 0.1\tau$.

Therefore, combine above we can show $\|\nabla_{\boldsymbol{\theta}} L_\lambda\|_F^2 = \Omega_*(\zeta^4/\lambda^2)$. $\qquad\square$

# G  Non-degenerate dual certificate

In this section, we show that there indeed exists a non-degenerate dual certificate that satisfies Definition 1 and therefore proving Lemma F.1.

**Lemma F.1.** *There exists a non-degenerate dual certificate $\eta = \mathbb{E}_{\boldsymbol{x}}[p(\boldsymbol{x})\sigma_{\geq 2}(\boldsymbol{w}^\top \boldsymbol{x})]$ with $\rho_\eta = \Theta(1)$ and $\|p\|_2 \leq \operatorname{poly}(m_*, \Delta)$*

Recall that we want to use the dual certificate $\eta$ to characterize the (approximate) solution for the following regression problem:

$$\min_{\mu \in \mathcal{M}(\mathbb{S}^{d-1})} L_\lambda(\mu) = \mathbb{E}_{\boldsymbol{x},\widetilde{y}}[(f_\mu(\boldsymbol{x}) - \widetilde{y})^2] + \lambda|\mu|_1 = \mathbb{E}_{\boldsymbol{x}}\left[\left(\int_{\boldsymbol{w}} \sigma_{\geq 2}(\boldsymbol{w}^\top \boldsymbol{x})\mathrm{d}\,\mu - \mu_*\right)^2\right] + \lambda|\mu|_1,$$

where $\sigma_{\geq 2}$ is the ReLU activation after removing 0th and 1st order (corresponding to $\alpha$ and $\beta$ terms) and $\mu_* = \sum_{i \in [m_*]} a_i^* \delta_{\boldsymbol{w}_i^*}$ is the ground-truth.

**Notation**  We need to first introduce few notations before proceeding to the proof. Denote the kernel $K_{\geq \ell}(\boldsymbol{w}, \boldsymbol{u}) = \mathbb{E}_{\boldsymbol{x} \sim N(0, \boldsymbol{I})}[\overline{\sigma_{\geq \ell}}(\overline{\boldsymbol{w}}^\top \boldsymbol{x})\overline{\sigma_{\geq \ell}}(\overline{\boldsymbol{u}}^\top \boldsymbol{x})]$ as the kernel induced by activation $\sigma_{\geq \ell}(x)$, where $\overline{\sigma_{\geq \ell}}(x) = \sum_{k \geq \ell} \hat{\sigma}_k h_k(x)/Z_\sigma$, $Z_\sigma = \|\sigma_{\geq \ell}\|_2 = \sqrt{\sum_{k \geq \ell} \hat{\sigma}_k^2} = \Theta(\ell^{-3/4})$ is the normalizing factor, $h_k(x)$ is the normalized $k$-th (probabilistic) Hermite polynomial and $\hat{\sigma}_k$ is the corresponding Hermite coefficient. We will specify the value of $\ell$ later and use $K$ instead of $K_{\geq \ell}$ for simplicity.

We will construct the dual certificate $\eta$ following the proof strategy in Poon et al. (2023) with the form below (the difference is that we now only keep high order terms that are at least $\ell$):

$$\eta(\boldsymbol{w}) = \sum_{j \in [m_*]} \alpha_{1,j} K(\boldsymbol{w}_j^*, \boldsymbol{w}) + \sum_{j \in [m_*]} \boldsymbol{\alpha}_{2,j}^\top \nabla_1 K(\boldsymbol{w}_j^*, \boldsymbol{w})$$

such that it satisfies

$$\eta(\boldsymbol{w}_i^*) = \operatorname{sign}(a_i^*) \text{ and } \nabla\eta(\boldsymbol{w}_i^*) = 0 \text{ for all } i \in [m_*]. \tag{9}$$

Here $\boldsymbol{\alpha}_1 = (\alpha_1, \ldots, \alpha_{m_*})^\top \in \mathbb{R}^{m_*}, \boldsymbol{\alpha}_2 = (\boldsymbol{\alpha}_{2,1}^\top, \ldots, \boldsymbol{\alpha}_{2,m_*}^\top)^\top \in \mathbb{R}^{m_* d}$ are the parameters that we are going to solve and $\nabla_i$ means the gradient w.r.t. $i$-th variable (for example, $\nabla_1 K(\boldsymbol{x}, \boldsymbol{y})$ means gradient with respect to $\boldsymbol{x}$).

One can rewrite the above constraints (9) into the matrix form:

$$\boldsymbol{\Upsilon} \begin{pmatrix} \boldsymbol{\alpha}_1 \\ \boldsymbol{\alpha}_2 \end{pmatrix} = \boldsymbol{b}, \tag{10}$$

where $\boldsymbol{b} = (\text{sign}(a_1^*), \ldots, \text{sign}(a_{m_*}^*), \boldsymbol{0}_{m^*d}^\top)^\top \in \mathbb{R}^{m_*(d+1)}$, $\boldsymbol{\Upsilon} = \mathbb{E}_{\boldsymbol{x}}[\boldsymbol{\gamma}(\boldsymbol{x})\boldsymbol{\gamma}(\boldsymbol{x})^\top] \in \mathbb{R}^{m_*(d+1)\times m_*(d+1)}$,

$$\boldsymbol{\gamma}(\boldsymbol{x}) = (\overline{\sigma_{\geq\ell}}(\boldsymbol{w}_1^{*\top}\boldsymbol{x}), \ldots, \overline{\sigma_{\geq\ell}}(\boldsymbol{w}_{m_*}^{*\top}\boldsymbol{x}), \nabla_{\boldsymbol{w}}\overline{\sigma_{\geq\ell}}(\overline{\boldsymbol{w}}_1^{*\top}\boldsymbol{x})^\top, \ldots, \nabla_{\boldsymbol{w}}\overline{\sigma_{\geq\ell}}(\overline{\boldsymbol{w}}_{m_*}^{*\top}\boldsymbol{x})^\top)^\top \in \mathbb{R}^{m_*(d+1)}.$$

Here $\nabla_{\boldsymbol{w}}\overline{\sigma_{\geq\ell}}(\overline{\boldsymbol{w}}_i^{*\top}\boldsymbol{x}) = \boldsymbol{P}_{\boldsymbol{w}_i^*}\overline{\sigma_{\geq\ell}}'(\boldsymbol{w}_i^{*\top}\boldsymbol{x})\boldsymbol{x} \in \mathbb{R}^d$, where $\boldsymbol{P}_{\boldsymbol{w}_i^*}$ is the projection matrix defined below.

**Notions on the unit sphere** As we could see, the kernel $K$ is invariant under the change of norms, so it suffices to focus on the input on the unit sphere $\mathbb{S}^{d-1}$. On the unite sphere, we could compute the gradient and hessian of a function $f(\boldsymbol{w})$ on the sphere (e.g., Absil et al. (2013))

$$\text{grad}\, f(\boldsymbol{w}) = \boldsymbol{P}_{\boldsymbol{w}}\nabla f(\boldsymbol{w}),$$
$$\text{H}\, f(\boldsymbol{w})[\boldsymbol{z}] = \boldsymbol{P}_{\boldsymbol{w}}(\nabla^2 f(\boldsymbol{w}) - \overline{\boldsymbol{w}}^\top\nabla f(\boldsymbol{w})\boldsymbol{I})\boldsymbol{z} \quad \text{for all tangent vector } \boldsymbol{z} \text{ that } \boldsymbol{z}^\top\boldsymbol{w} = 0,$$

where $\boldsymbol{P}_{\boldsymbol{w}} = \boldsymbol{I} - \boldsymbol{w}\boldsymbol{w}^\top$ is the projection matrix.

Then, we could define the derivative as in Poon et al. (2023); Absil et al. (2008): for tangent vectors $\boldsymbol{z}, \boldsymbol{z}'$

$$\text{D}_0\, f(\boldsymbol{w}) := f(\boldsymbol{w})$$
$$\text{D}_1\, f(\boldsymbol{w})[\boldsymbol{z}] := \langle \boldsymbol{z}, \text{grad}\, f(\boldsymbol{w})\rangle = \boldsymbol{z}^\top\boldsymbol{P}_{\boldsymbol{w}}\nabla f(\boldsymbol{w})$$
$$\text{D}_2\, f(\boldsymbol{w})[\boldsymbol{z}, \boldsymbol{z}'] := \langle \text{H}\, f(\boldsymbol{w})[\boldsymbol{z}], \boldsymbol{z}'\rangle = \boldsymbol{z}^\top\boldsymbol{P}_{\boldsymbol{w}}(\nabla^2 f(\boldsymbol{w}) - \overline{\boldsymbol{w}}^\top\nabla f(\boldsymbol{w})\boldsymbol{I})\boldsymbol{P}_{\boldsymbol{w}}\boldsymbol{z}',$$

and their associated norms

$$\|\text{D}_1\, f(\boldsymbol{w})\|_{\boldsymbol{w}} := \sup_{\|\boldsymbol{z}\|_{\boldsymbol{w}}=1} \text{D}_1\, f(\boldsymbol{w})[\boldsymbol{z}] = \|\boldsymbol{P}_{\boldsymbol{w}}\nabla f(\boldsymbol{w})\|_2,$$
$$\|\text{D}_2\, f(\boldsymbol{w})\|_{\boldsymbol{w}} := \sup_{\|\boldsymbol{z}\|_{\boldsymbol{w}},\|\boldsymbol{z}'\|_{\boldsymbol{w}}=1} \text{D}_2\, f(\boldsymbol{w})[\boldsymbol{z}, \boldsymbol{z}'] = \|\boldsymbol{P}_{\boldsymbol{w}}\,\text{H}\, f(\boldsymbol{w})\boldsymbol{P}_{\boldsymbol{w}}\|_2,$$

where $\|\boldsymbol{z}\|_{\boldsymbol{w}} = \|\boldsymbol{P}_{\boldsymbol{w}}\boldsymbol{z}\|_2$.

For simplicity, we will use $K^{(ij)}(\boldsymbol{w}, \boldsymbol{u})$ to denote $\nabla_1^i\nabla_2^j K(\boldsymbol{w}, \boldsymbol{u})$. One can check that this is in fact the same as the one defined Poon et al. (2023) under our specific kernel $K$, $i + j \leq 3$ and $i, j \leq 2$. Let

$$\left\|K^{(ij)}(\boldsymbol{w}, \boldsymbol{u})\right\|_{\boldsymbol{w},\boldsymbol{u}} := \sup_{\substack{\|\boldsymbol{z}_{\boldsymbol{w}}^{(p)}\|_{\boldsymbol{w}}=\|\boldsymbol{z}_{\boldsymbol{u}}^{(q)}\|_{\boldsymbol{u}}=1, \\ \boldsymbol{w}^\top\boldsymbol{z}_{\boldsymbol{w}}^{(p)}=\boldsymbol{u}^\top\boldsymbol{z}_{\boldsymbol{u}}^{(q)}=0\,\forall p\in[i],q\in[j]}} K^{(ij)}(\boldsymbol{w}, \boldsymbol{u})[\boldsymbol{z}_{\boldsymbol{w}}^{(1)}, \ldots, \boldsymbol{z}_{\boldsymbol{u}}^{(j)}],$$

where $\boldsymbol{z}_{\boldsymbol{w}}^{(p)}$ applies to the dimension corresponding to $\boldsymbol{w}$ and similarly $\boldsymbol{z}_{\boldsymbol{u}}^{(q)}$ for $\boldsymbol{u}$.

Before solving (10), we first present some useful proprieties of kernel $K$ that will be used later (see Section I for the proofs). The lemma below shows that kernel $K(\boldsymbol{w}, \boldsymbol{u})$ is non-degenerate in the sense that it decays at least quadratic at each ground-truth direction ($\boldsymbol{w} \approx \boldsymbol{u} \approx \boldsymbol{w}_i^*$) and contributes almost nothing when $\boldsymbol{w}, \boldsymbol{u}$ are away.

**Lemma G.1** (Non-degeneracy of kernel $K$). *For any $h > 0$, let $\ell \geq \Theta(\Delta^{-2}\log(m_*\ell/h\Delta))$, kernel $K_{\geq\ell}$ is non-degenerate in the sense that there exists $r = \Theta(\ell^{-1/2}), \rho_1 = \Theta(1), \rho_2 = \Theta(\ell)$ such that following hold:*

*(i)* $K(\boldsymbol{w}, \boldsymbol{u}) \leq 1 - \rho_1$ *for all* $\delta(\boldsymbol{w}, \boldsymbol{u}) := \angle(\boldsymbol{w}, \boldsymbol{u}) \geq r$.

*(ii)* $K^{(20)}(\boldsymbol{w}, \boldsymbol{u})[\boldsymbol{z}, \boldsymbol{z}] \leq -\rho_2 \|\boldsymbol{z}\|^2$ *for tangent vector $\boldsymbol{z}$ that $\boldsymbol{z}^\top\boldsymbol{w} = 0$ and $\delta(\boldsymbol{w}, \boldsymbol{u}) \leq r$.*

*(iii)* $\left\|K^{(ij)}(\boldsymbol{w}_1^*, \boldsymbol{w}_k^*)\right\|_{\boldsymbol{w}_i^*,\boldsymbol{w}_k^*} \leq h/m_*^2$ *for* $(i, j) \in \{0, 1\} \times \{0, 1, 2\}$

The following lemma shows that $K$ and its derivatives are bounded.

**Lemma G.2** (Regularity conditions on kernel $K$). *Let $B_{ij} := \sup_{\boldsymbol{w},\boldsymbol{u}} \|K^{(ij)}(\boldsymbol{w}, \boldsymbol{u})\|_{\boldsymbol{w},\boldsymbol{u}}$ and $B_0 = B_{00} + B_{10} + 1$, $B_2 = B_{20} + B_{21} + 1$. We have $B_{00} = O(1)$, $B_{10} = O(\ell^{1/2})$, $B_{11} = O(\ell)$, $B_{20} = O(\ell)$, $B_{21} = O(\ell^{3/2})$, and therefore $B_0 = O(\ell^{1/2})$, $B_2 = O(\ell^{3/2})$.*

The following lemma from Poon et al. (2023) connects the non-degeneracy of kernel $K$ to the dual certificate $\eta$ that we are interested in.

**Lemma G.3** (Lemma 2, Poon et al. (2023), adapted in our setting). *Let $a \in \{\pm 1\}$. Suppose that for some $\rho > 0$, $B > 0$ and $0 < r \leq B^{-1/2}$ we have: for all $\delta(\boldsymbol{w}, \boldsymbol{w}_0)$ and $\boldsymbol{z} \in \mathbb{R}^d$ with $\boldsymbol{z}^\top \boldsymbol{w} = 0$, it holds that $-K^{(02)}(\boldsymbol{w}_0, \boldsymbol{w})[\boldsymbol{z}, \boldsymbol{z}] > \rho \|\boldsymbol{z}\|_2^2$ and $\left\|K^{(02)}(\boldsymbol{w}_0, \boldsymbol{w})\right\|_{\boldsymbol{w}} \leq B$. Let $\eta$ be a smooth function. If $\eta(\boldsymbol{w}_0) = a$, $\nabla \eta(\boldsymbol{w}_0) = 0$ and $\left\|a \operatorname{D}_2 \eta(\boldsymbol{w}) - K^{(02)}(\boldsymbol{w}_0, \boldsymbol{w})\right\|_{\boldsymbol{w}} \leq \tau$ for all $\delta(\boldsymbol{w}, \boldsymbol{w}_0) \leq r$ with $\tau < \rho/2$, then we have $|\eta(\boldsymbol{w})| \leq 1 - ((\rho - 2\tau)/2)\delta(\boldsymbol{w}, \boldsymbol{w}_0)^2$ for all $\delta(\boldsymbol{w}, \boldsymbol{w}_0) \leq r$.*

We now are ready to proof the main result in this section Lemma F.1 that shows the non-degenerate dual certificate exists. Roughly speaking, following the same proof as in Poon et al. (2023), we can show that $\boldsymbol{\alpha} \approx \operatorname{sign}(\boldsymbol{a}_*)$ and $\boldsymbol{\alpha}_2 \approx \boldsymbol{0}$ and therefore we can transfer the non-degeneracy of kernel $K$ to the dual certificate $\eta$ with Lemma G.3.

**Lemma F.1.** *There exists a non-degenerate dual certificate $\eta = \mathbb{E}_{\boldsymbol{x}}[p(\boldsymbol{x})\sigma_{\geq 2}(\boldsymbol{w}^\top \boldsymbol{x})]$ with $\rho_\eta = \Theta(1)$ and $\|p\|_2 \leq \operatorname{poly}(m_*, \Delta)$.*

*Proof.* Note that $\boldsymbol{\Upsilon} = \boldsymbol{S}\boldsymbol{D}\widetilde{\boldsymbol{\Upsilon}}\boldsymbol{D}\boldsymbol{S}$, where

$$
\boldsymbol{D} = \begin{pmatrix} \boldsymbol{I}_{m_*} & & & \\ & \boldsymbol{P}_{\boldsymbol{w}_1^*} & & \\ & & \ddots & \\ & & & \boldsymbol{P}_{\boldsymbol{w}_{m_*}^*} \end{pmatrix}, \quad \boldsymbol{S} = \begin{pmatrix} \boldsymbol{I}_{m_*} & & & \\ & (Z_{\sigma'}/Z_\sigma)\boldsymbol{I}_{m_*} & & \\ & & \ddots & \\ & & & (Z_{\sigma'}/Z_\sigma)\boldsymbol{I}_{m_*} \end{pmatrix}
$$

are block diagonal matrices, $\widetilde{\boldsymbol{\Upsilon}} = \mathbb{E}_{\boldsymbol{x}}[\widetilde{\boldsymbol{\gamma}}(\boldsymbol{x})\widetilde{\boldsymbol{\gamma}}(\boldsymbol{x})^\top] \in \mathbb{R}^{m_*(d+1) \times m_*(d+1)}$,

$$
\widetilde{\boldsymbol{\gamma}}(\boldsymbol{x}) = (\overline{\sigma_{\geq \ell}}(\boldsymbol{w}_1^{*\top}\boldsymbol{x}), \ldots, \overline{\sigma_{\geq \ell}}(\boldsymbol{w}_{m_*}^{*\top}\boldsymbol{x}), (Z_\sigma/Z_{\sigma'})\overline{\sigma_{\geq \ell}}'(\boldsymbol{w}_1^{*\top}\boldsymbol{x})\boldsymbol{x}^\top, \ldots, (Z_\sigma/Z_{\sigma'})\overline{\sigma_{\geq \ell}}'(\boldsymbol{w}_{m_*}^{*\top}\boldsymbol{x})\boldsymbol{x}^\top)^\top \in \mathbb{R}^{m_*(d+1)},
$$

$Z_{\sigma'} = \sqrt{\sum_{k \geq \ell} \hat{\sigma}_k^2 k} = \Theta(\ell^{-1/4})$ is the normalizing factor so that the diagonal of $\widetilde{\boldsymbol{\Upsilon}}$ are all 1.

Thus, to solve (10), it is sufficient to solve the following: denote $\widetilde{\boldsymbol{K}} = \boldsymbol{D}\widetilde{\boldsymbol{\Upsilon}}\boldsymbol{D}$

$$
\widetilde{\boldsymbol{K}}\begin{pmatrix} \widetilde{\boldsymbol{\alpha}}_1 \\ \widetilde{\boldsymbol{\alpha}}_2 \end{pmatrix} = \boldsymbol{b}, \tag{11}
$$

and let $\boldsymbol{\alpha}_1 = \widetilde{\boldsymbol{\alpha}}_1$, $\boldsymbol{\alpha}_{2,i} = (Z_\sigma/Z_{\sigma'})\widetilde{\boldsymbol{\alpha}}_{2,i}$ to get the solution of (10).

In the following, we are going to first show that $\widetilde{\boldsymbol{K}} \approx \boldsymbol{D}\boldsymbol{D}$ because all the off-diagonal terms of $\widetilde{\boldsymbol{\Upsilon}}$ are small due to Lemma G.1 (iii) (we can choose $h$ to be small enough, and we will choose it later). Specifically, we have

$$
\left\|\widetilde{\boldsymbol{K}} - \boldsymbol{D}\boldsymbol{D}\right\|_2 = \sup_{\|\boldsymbol{z}\|_2=1} |\boldsymbol{z}^\top(\widetilde{\boldsymbol{K}} - \boldsymbol{D}\boldsymbol{D})\boldsymbol{z}|
$$

$$
= \sup_{\|\boldsymbol{z}\|_2=1} \left| \sum_{i,j} z_{1,i}K(\boldsymbol{w}_i^*, \boldsymbol{w}_j^*)z_{1,j} + 2(Z_\sigma/Z_{\sigma'})\sum_{i,j} z_{1,i}\nabla_1 K(\boldsymbol{w}_i^*, \boldsymbol{w}_j^*)^\top \boldsymbol{z}_{2,j} \right.
$$

$$
\left. + (Z_\sigma/Z_{\sigma'})^2 \sum_{i,j} \boldsymbol{z}_{2,i}^\top \nabla_1\nabla_2 K(\boldsymbol{w}_i^*, \boldsymbol{w}_j^*)^\top \boldsymbol{z}_{2,j} \right|
$$

$$
\leq \sum_{i,j} |K(\boldsymbol{w}_i^*, \boldsymbol{w}_j^*)| + \Theta(\ell^{-1/2})\left\|K^{(10)}(\boldsymbol{w}_i^*, \boldsymbol{w}_j^*)\right\|_{\boldsymbol{w}_i^*} + \Theta(\ell^{-1})\left\|K^{(11)}(\boldsymbol{w}_i^*, \boldsymbol{w}_j^*)\right\|_{\boldsymbol{w}_i^*, \boldsymbol{w}_j^*} \leq 2h,
$$

where $\boldsymbol{z} = (\boldsymbol{z}_1^\top, \boldsymbol{z}_2^\top)^\top$, $\boldsymbol{z}_1 = (z_{1,1}, \ldots, z_{1,m_*})^\top$ and $\boldsymbol{z}_2 = (\boldsymbol{z}_{2,1}^\top, \ldots, \boldsymbol{z}_{2,m_*}^\top)^\top$ has the same block structure as $(\boldsymbol{\alpha}_1, \boldsymbol{\alpha}_2)$ and we use Lemma G.1 and Lemma G.2 in the last line.

Note that $\boldsymbol{D}\boldsymbol{D}$ has exactly $m_*d$ eigenvalues of 1 and $m_*$ eigenvalues of 0, and $\widetilde{\boldsymbol{K}}$ also has $m_*$ eigenvalues of 0. By Weyl's inequality, we know $|\gamma_i - 1| \leq 2h$ where $\widetilde{\boldsymbol{K}} = \sum_{i \in [m_*d]} \gamma_i \boldsymbol{v}_i \boldsymbol{v}_i^\top$ is its eigendecomposition. Here $\boldsymbol{v}_i^\top \boldsymbol{v}_\perp = 0$ for all $\boldsymbol{v}_\perp \in V_\perp = \operatorname{span}\{(\boldsymbol{0}, \boldsymbol{w}_1^*, \boldsymbol{0}, \ldots, \boldsymbol{0})^\top, \ldots (\boldsymbol{0}, \ldots, \boldsymbol{0}, \boldsymbol{w}_{m_*}^*)^\top\}$

in the null space of $\boldsymbol{D}$. Since $\boldsymbol{b}^\top \boldsymbol{v}_\perp = 0$ for all $\boldsymbol{v}_\perp \in V_\perp$, we have

$$
\begin{pmatrix} \widetilde{\boldsymbol{\alpha}}_1 \\ \widetilde{\boldsymbol{\alpha}}_2 \end{pmatrix} = \widetilde{\boldsymbol{K}}^\dagger \boldsymbol{b} = \sum_{i \in [m_* d]} \gamma_i^{-1} \boldsymbol{v}_i \boldsymbol{v}_i^\top \boldsymbol{b} = \sum_{i \in [m_* d]} (\gamma_i^{-1} - 1) \boldsymbol{v}_i \boldsymbol{v}_i^\top \boldsymbol{b} + \sum_{i \in [m_* d]} \boldsymbol{v}_i \boldsymbol{v}_i^\top \boldsymbol{b}
$$

$$
= \sum_{i \in [m_* d]} (\gamma_i^{-1} - 1) \boldsymbol{v}_i \boldsymbol{v}_i^\top \boldsymbol{b} + \boldsymbol{b}.
$$

Therefore,

$$
\left\| \begin{pmatrix} \widetilde{\boldsymbol{\alpha}}_1 \\ \widetilde{\boldsymbol{\alpha}}_2 \end{pmatrix} - \boldsymbol{b} \right\|_2 \leq \left\| \sum_{i \in [m_* d]} (\gamma_i^{-1} - 1) \boldsymbol{v}_i \boldsymbol{v}_i^\top \boldsymbol{b} \right\|_2 \leq \max_i |\gamma_i^{-1} - 1| \sqrt{m_*} = O(h \sqrt{m_*}) =: h'.
$$

This implies $\|\boldsymbol{\alpha}_1 - \text{sign}(\boldsymbol{a}_*)\|_\infty = \|\widetilde{\boldsymbol{\alpha}}_1 - \text{sign}(\boldsymbol{a}_*)\|_\infty \leq h'$, $\|\boldsymbol{\alpha}_1\|_\infty = \|\widetilde{\boldsymbol{\alpha}}_1\|_\infty \leq 1 + h'$ and $\|\boldsymbol{\alpha}_2\|_2 = (Z_\sigma / Z_{\sigma'}) \|\widetilde{\boldsymbol{\alpha}}_{2,i}\|_2 \leq \Theta(h' \ell^{-1/2})$.

Now, given the $\boldsymbol{\alpha}_1, \boldsymbol{\alpha}_2$, we can show the corresponding $\eta$ is non-degenerate. Choosing $h = O(m_*^{-1/2})$ and $\ell = \Theta(\Delta^{-2} \log(m_*/\Delta))$ so that the condition in Lemma G.1 holds.

Consider $\boldsymbol{w} \in \mathcal{T}_i$, when $\delta(\boldsymbol{w}, \boldsymbol{w}_i^*) \geq r = \Theta(\ell^{-1/2})$, using Lemma G.1 and Lemma G.2 we have

$$
|\eta(\boldsymbol{w})| = \left| \sum_{j \in [m_*]} \alpha_{1,j} K(\boldsymbol{w}_j^*, \boldsymbol{w}) + \sum_{j \in [m_*]} \boldsymbol{\alpha}_{2,j}^\top \nabla_1 K(\boldsymbol{w}_j^*, \boldsymbol{w}) \right|
$$

$$
\leq \sum_{j \in [m_*]} |\alpha_{1,j}| |K(\boldsymbol{w}_j^*, \boldsymbol{w})| + \sum_{j \in [m_*]} \|\boldsymbol{\alpha}_{2,j}\|_{\boldsymbol{w}_j^*} \|\nabla_1 K(\boldsymbol{w}_j^*, \boldsymbol{w})\|_{\boldsymbol{w}_j^*}
$$

$$
\leq (1 + h')(1 - \rho_1 + h) + \Theta(h' \ell^{-1/2})(B_{10} + h) \leq 1 - \rho_1/2 \leq 1 - \Theta(\rho_1) \delta(\boldsymbol{w}, \boldsymbol{w}_i^*)^2,
$$

where we choose $h = O(m_*^{-1/2})$ to be small enough.

When $\delta(\boldsymbol{w}, \boldsymbol{w}_i^*) \leq r = \Theta(\ell^{-1/2})$, again using Lemma G.1 and Lemma G.2 we have

$$
\left\| a_i^* \operatorname{D}_2 \eta(\boldsymbol{w}) - K^{(02)}(\boldsymbol{w}_i^*, \boldsymbol{w}) \right\|_{\boldsymbol{w}}
$$

$$
\leq \left\| \alpha_{1,i} K^{(02)}(\boldsymbol{w}_i^*, \boldsymbol{w}) - K^{(02)}(\boldsymbol{w}_i^*, \boldsymbol{w}) \right\|_{\boldsymbol{w}} + \sum_{j \neq i} \left\| \alpha_{1,j} K^{(02)}(\boldsymbol{w}_j^*, \boldsymbol{w}) \right\|_{\boldsymbol{w}} + \sum_{j \in [m_*]} \|\boldsymbol{\alpha}_{2,j}\|_{\boldsymbol{w}_j^*} \left\| K^{(12)}(\boldsymbol{w}_j^*, \boldsymbol{w}) \right\|_{\boldsymbol{w}_j^*, \boldsymbol{w}}
$$

$$
\leq h' B_{02} + (1 + h')h + \Theta(h' \ell^{-1/2})(B_{21} + h) \leq \rho_2/16,
$$

where again due to our choice of small $h$. Using Lemma G.3 we know that $|\eta(\boldsymbol{w})| \leq 1 - (\rho_2/4)\delta(\boldsymbol{w}, \boldsymbol{w}_i^*)^2$.

Combine the above two cases, we have $|\eta(\boldsymbol{w})| \leq 1 - \Theta(1)\delta(\boldsymbol{w}, \boldsymbol{w}_i^*)^2$ and $\eta(\boldsymbol{w}) = \mathbb{E}_{\boldsymbol{x}}[p(\boldsymbol{x})\sigma(\boldsymbol{w}^\top \boldsymbol{x})]$ with

$$
p(\boldsymbol{x}) = \frac{1}{Z_\sigma^2} \left( \sum_{j \in [m_*]} \alpha_{1,j} \sigma_{\geq \ell}(\boldsymbol{w}_j^{*\top} \boldsymbol{x}) + \sum_{j \in [m_*]} \boldsymbol{\alpha}_{2,j}^\top (\boldsymbol{I} - \boldsymbol{w}_i^* \boldsymbol{w}_i^{*\top}) \boldsymbol{x} \sigma_{\geq \ell}'(\boldsymbol{w}_i^{*\top} \boldsymbol{x}) \right).
$$

We have $\|p\| = O(\ell^{3/4} m_* + m_* h' \ell^{-1/2} \ell^{5/4}) = \widetilde{O}(\Delta^{-3/2} m_*)$. $\qquad\square$

## H   Proofs in Section F

In this section, we give the omitted proofs in Section F.

### H.1   Omitted proofs in Section F.1

We give the proofs for these results that characterize the structure of ideal loss solution.

The following proof follows from the definition of non-degenerate dual certificate $\eta$.

**Lemma F.2.** *Given a non-degenerate dual certificate $\eta$, then*

*(i)* $\langle \eta, \mu^* \rangle = |\mu^*|_1$

*(ii) For any measure $\mu \in \mathcal{M}(\mathbb{S}^{d-1})$, $|\langle \eta, \mu \rangle| \leq |\mu|_1 - \rho_\eta \sum_{i \in [m_*]} \int_{\mathcal{T}_i} \delta(\boldsymbol{w}, \boldsymbol{w}_i^*)^2 \, \mathrm{d}|\mu|(\boldsymbol{w})$.*

*(iii)* $\langle \eta, \mu - \mu^* \rangle = \langle p, f_\mu - f_{\mu^*} \rangle$, *where* $f_\mu(\boldsymbol{x}) = \mathbb{E}_{\boldsymbol{w} \sim \mu}[\sigma_{\geq 2}(\boldsymbol{w}^\top \boldsymbol{x})]$. *Then* $|\langle \eta, \mu - \mu^* \rangle| \leq \|p\|_2 \sqrt{L(\mu)}$.

*Proof.* We show the results one by one.

**Part (i)(ii)** We have

$$|\langle \eta, \mu \rangle| \leq \int_{\mathbb{S}^{d-1}} |\eta(\boldsymbol{w})| \, \mathrm{d}|\mu|(\boldsymbol{w}) = \sum_{i \in [m_*]} \int_{\mathcal{T}_i} |\eta(\boldsymbol{w})| \, \mathrm{d}|\mu|(\boldsymbol{w}) \leq |\mu|_1 - \rho_\eta \sum_{i \in [m_*]} \int_{\mathcal{T}_i} \delta(\boldsymbol{w}, \boldsymbol{w}_i^*)^2 \, \mathrm{d}|\mu|(\boldsymbol{w}).$$

where the last inequality follows the property of non-degenerate dual certificate (Definition 1). The other part then follows directly by the definition of $\mu^*$.

**Part (iii)** We have

$$\langle \eta, \mu - \mu^* \rangle = \int_{\mathbb{S}^{d-1}} \eta(\boldsymbol{w}) \, \mathrm{d}(\mu - \mu^*)(\boldsymbol{w}) = \int_{\mathbb{S}^{d-1}} \mathbb{E}_{\boldsymbol{x}}[p(\boldsymbol{x})\sigma_{\geq 2}(\boldsymbol{w}^\top \boldsymbol{x})] \, \mathrm{d}(\mu - \mu^*)(\boldsymbol{w})$$

$$= \mathbb{E}_{\boldsymbol{x}} \left[ p(\boldsymbol{x}) \int_{\mathbb{S}^{d-1}} \sigma_{\geq 2}(\boldsymbol{w}^\top \boldsymbol{x}) \, \mathrm{d}(\mu - \mu^*)(\boldsymbol{w}) \right]$$

$$= \mathbb{E}_{\boldsymbol{x}}[p(\boldsymbol{x})(f_\mu(\boldsymbol{x}) - f_{\mu^*}(\boldsymbol{x}))].$$

Note that $L(\mu) = \|f_\mu - f_{\mu^*}\|_2^2$, this leads to $|\langle \eta, \mu - \mu^* \rangle| \leq \|p\|_2 \sqrt{L(\mu)}$. $\qquad \square$

Given the above lemma and the optimality of $\mu_\lambda^*$, we are able to characterize the structure of $\mu_\lambda^*$ as below: norm is bounded, square loss is small and far-away neurons are small.

**Lemma F.3.** *We have the following hold*

*(i)* $|\mu_*|_1 - \lambda \|p\|_2^2 \leq |\mu_\lambda^*|_1 \leq |\mu^*|_1 = \|\boldsymbol{a}^*\|_1$

*(ii)* $L(\mu_\lambda^*) \leq \lambda^2 \|p\|_2^2 = O_*(\lambda^2)$

*(iii)* $\sum_{i \in [m_*]} \int_{\mathcal{T}_i} \delta(\boldsymbol{w}, \boldsymbol{w}_i^*)^2 \, \mathrm{d}|\mu_\lambda^*|(\boldsymbol{w}) \leq \lambda \|p\|_2^2 / \rho_\eta = O_*(\lambda)$

*Proof.* We show the results one by one.

**Part (i)** Due to the optimality of $\mu_\lambda^*$, we have

$$L(\mu_\lambda^*) + \lambda|\mu_\lambda^*|_1 = L_\lambda(\mu_\lambda^*) \leq L_\lambda(\mu^*) = L(\mu^*) + \lambda|\mu^*|_1.$$

Rearranging the terms, we have

$$\lambda|\mu_\lambda^*|_1 - \lambda|\mu^*|_1 \leq L(\mu^*) - L(\mu_\lambda^*) = -L(\mu_\lambda^*) \leq 0.$$

For the lower bound, with Lemma F.2 we have

$$0 \leq |\mu_\lambda^*|_1 - |\mu^*|_1 - \langle \eta, \mu_\lambda^* - \mu^* \rangle \leq |\mu_\lambda^*|_1 - |\mu^*|_1 + \|p\|_2 \sqrt{L(\mu_\lambda^*)}.$$

Using part (ii) we get the desired lower bound.

**Part (ii)**   We first have the following inequality due to the optimality of $\mu_\lambda^*$ and adding $\lambda\langle\eta, \mu_\lambda^* - \mu^*\rangle$ on both side:

$$L(\mu_\lambda^*) + \underbrace{\lambda(|\mu_\lambda^*|_1 - |\mu^*|_1) - \lambda\langle\eta, \mu_\lambda^* - \mu^*\rangle}_{(I)} \leq L(\mu^*) - \lambda\langle\eta, \mu_\lambda^* - \mu^*\rangle.$$

For $(I)$, we have

$$(I) = \lambda(|\mu_\lambda^*|_1 - \langle\eta, \mu_\lambda^*\rangle) + \lambda(\langle\eta, \mu^*\rangle - |\mu^*|_1) \geq 0,$$

where we use Lemma F.2 in the last inequality.

Therefore, the above inequality leads to

$$L(\mu_\lambda^*) \leq L(\mu^*) - \lambda\langle\eta, \mu_\lambda^* - \mu^*\rangle \leq \lambda\|p\|_2 \sqrt{L(\mu_\lambda^*)},$$

where we again use Lemma F.2. This further leads to $L(\mu_\lambda^*) \leq \lambda^2 \|p\|_2^2$.

**Part (iii)**   Using part (i) we have

$$|\mu_\lambda^*|_1 - |\mu^*|_1 - \langle\eta, \mu_\lambda^* - \mu^*\rangle \leq -\langle\eta, \mu_\lambda^* - \mu^*\rangle.$$

With Lemma F.2, LHS and RHS become

$$\text{LHS} = |\mu_\lambda^*|_1 - \langle\eta, \mu_\lambda^*\rangle \geq \rho_\eta \sum_{i\in[m_*]} \int_{\mathcal{T}_i} \delta(\boldsymbol{w}, \boldsymbol{w}_i^*)^2 \, \mathrm{d}|\mu_\lambda^*|(\boldsymbol{w})$$

$$\text{RHS} \leq \|p\|_2 \sqrt{L(\mu_\lambda^*)}.$$

Then using part (ii) we have the desired result. $\qquad\square$

We are now ready to characterize the approximated solution by comparing $\mu$ and $\mu_\lambda^*$.

**Lemma F.4.** *Recall the optimality gap $\zeta = L_\lambda(\mu) - L_\lambda(\mu_\lambda^*)$. Then, the following holds:*

*(i) $L(\mu) \leq 5\lambda^2 \|p\|^2 + 4\zeta = O_*(\lambda^2 + \zeta)$.*

*(ii) if $\zeta \leq \lambda|\mu^*|_1$ and $\lambda \leq |\mu^*|_1/ \|p\|_2^2$, then $|\mu|_1 \leq 3|\mu^*|_1 = 3\|\boldsymbol{a}^*\|_1$.*

*Proof.*   We show the results one by one.

**Part (i)**   By the definition of the optimality gap $\zeta$ and adding $-\lambda\langle\eta, \mu - \mu^*\rangle$ on both side, we have

$$L(\mu) + \lambda(|\mu|_1 - |\mu_\lambda^*|_1) - \lambda\langle\eta, \mu - \mu^*\rangle \leq L(\mu_\lambda^*) + \zeta - \lambda\langle\eta, \mu - \mu^*\rangle.$$

Note that on LHS,

$$\lambda(|\mu|_1 - |\mu_\lambda^*|_1) - \lambda\langle\eta, \mu - \mu^*\rangle = \lambda(|\mu|_1 - \langle\eta, \mu\rangle) + \lambda(|\mu^*|_1 - |\mu_\lambda^*|_1) \geq 0,$$

where we use Lemma F.2 and Lemma F.3.

Therefore, with Lemma F.2 and Lemma F.3 we get

$$L(\mu) \leq L(\mu_\lambda^*) + \zeta - \lambda\langle\eta, \mu - \mu^*\rangle \leq \lambda^2 \|p\|_2^2 + \zeta + \lambda\|p\|_2 \sqrt{L(\mu)}.$$

Solving the above inequality on $L(\mu)$ gives $L(\mu) \leq 5\lambda^2 \|p\|_2^2 + 4\zeta$.

**Part (ii)**   Again from the definition of the optimality gap $\zeta$, we have

$$\lambda|\mu|_1 \leq L(\mu_\lambda^*) + \lambda|\mu_\lambda^*|_1 + \zeta - L(\mu) \leq \lambda^2 \|p\|_2^2 + \lambda|\mu^*|_1 + \zeta,$$

where we use Lemma F.3. Thus, $|\mu|_1 \leq \lambda\|p\|_2^2 + |\mu^*|_1 + \zeta/\lambda \leq 3|\mu^*|_1$. $\qquad\square$

The lemma below shows that far-away neurons are still small even for the approximated solution. Intutively, we use the non-degenerate dual certificate to certify the gap between $\mu$ and $\mu_\lambda^*$ and give a bound for it.

**Lemma F.5.** *Recall the optimality gap $\zeta = L_\lambda(\mu) - L_\lambda(\mu_\lambda^*)$. Then, we have*

$$\sum_{i \in [m_*]} \int_{\mathcal{T}_i} \delta(\boldsymbol{w}, \boldsymbol{w}_i^*)^2 \, \mathrm{d}|\mu|(\boldsymbol{w}) \leq (\zeta/\lambda + 2\lambda \|p\|_2^2)/\rho_\eta = O_*(\zeta/\lambda + \lambda).$$

*In particular, when $\mu = \sum_{i \in [m]} a_i \|\boldsymbol{w}_i\|_2 \delta_{\overline{\boldsymbol{w}}_i}$ represents finite number of neurons, we have*

$$\sum_{i \in [m_*]} \sum_{j \in \mathcal{T}_i} |a_j| \|\boldsymbol{w}_j\|_2 \delta_j^2 \leq (\zeta/\lambda + 2\lambda \|p\|_2^2)/\rho_\eta = O_*(\zeta/\lambda + \lambda),$$

*where $\delta_j = \angle(\boldsymbol{w}_j, \boldsymbol{w}_i^*)$ for $j \in \mathcal{T}_i$.*

*Proof.* By the definition of the optimality gap $\zeta$, we have

$$L(\mu) + \lambda|\mu|_1 = L(\mu_\lambda^*) + \lambda|\mu_\lambda^*|_1 + \zeta.$$

Rearranging the terms and adding $-\langle \eta, \mu - \mu^* \rangle$ on both side, we get

$$|\mu|_1 - |\mu_\lambda^*|_1 - \langle \eta, \mu - \mu^* \rangle = \frac{1}{\lambda}(L(\mu_\lambda^*) - L(\mu) + \zeta) - \langle \eta, \mu - \mu^* \rangle.$$

For LHS, with Lemma F.2 and Lemma F.3 we have

$$\mathrm{LHS} = |\mu|_1 - \langle \eta, \mu \rangle - |\mu_\lambda^*|_1 + |\mu^*|_1 \geq \rho_\eta \sum_{i \in [m_*]} \int_{\mathcal{T}_i} \delta(\boldsymbol{w}, \boldsymbol{w}_i^*)^2 \, \mathrm{d}|\mu|(\boldsymbol{w}).$$

For RHS, with Lemma F.2 and Lemma F.3 we have

$$\mathrm{RHS} \leq \frac{1}{\lambda}(\lambda^2 \|p\|_2^2 - L(\mu) + \zeta) + \|p\|_2 \sqrt{L(\mu)} = \frac{\zeta}{\lambda} + \lambda \|p\|_2^2 - \frac{L(\mu)}{\lambda} + \|p\|_2 \sqrt{L(\mu)}.$$

When $L(\mu) \geq \lambda^2 \|p\|_2^2$, we have $\mathrm{RHS} \leq \zeta/\lambda + \lambda \|p\|_2^2$. When $L(\mu) \leq \lambda^2 \|p\|_2^2$, we have $\mathrm{RHS} \leq \zeta/\lambda + 2\lambda \|p\|_2^2$. Thus, in summary $\mathrm{RHS} \leq \zeta/\lambda + 2\lambda \|p\|_2^2$.

Combine the bounds on LHS and RHS we have

$$\rho_\eta \sum_{i \in [m_*]} \int_{\mathcal{T}_i} \delta(\boldsymbol{w}, \boldsymbol{w}_i^*)^2 \, \mathrm{d}|\mu|(\boldsymbol{w}) \leq \zeta/\lambda + 2\lambda \|p\|_2^2.$$

$\square$

The following lemma shows that every teacher neuron must have at least one close-by student neuron within angle $O_*(\zeta^{1/3})$. This generalize and greatly simplify the previous results Lemma 9 in Zhou et al. (2021). In particular, we design a new test function using the Hermite expansion to achieve this.

**Lemma F.6.** *Under Lemma 6, if the Hermite coefficient of $\sigma$ decays as $|\hat{\sigma}_k| = \Theta(k^{-c_\sigma})$ with some constant $c_\sigma > 0$, then the total mass near each target direction is large, i.e., $\mu(\mathcal{T}_i(\delta)) \operatorname{sign}(a_i^*) \geq |a_i^*|/2$ for all $i \in [m_*]$ and any $\delta_{close} \geq \widetilde{\Omega}\left((\frac{L(\mu)}{a_{\min}^2})^{1/(4c_\sigma - 2)}\right)$ with large enough hidden constant. In particular, for $\sigma$ is ReLU or absolute function, $\delta_{close} \geq \widetilde{\Omega}\left((\frac{L(\mu)}{a_{\min}^2})^{1/3}\right)$. Here $a_{\min} = \min |a_i|$ is the smallest entry of $\boldsymbol{a}_*$ in absolute value.*

*As a corollary, if the optimality gap $\zeta = L_\lambda(\mu) - L_\lambda(\mu_\lambda^*)$, then $\delta_{close} \geq \widetilde{\Omega}_*\left((\zeta + \lambda^2)^{1/(4c_\sigma - 2)}\right)$ and for ReLU or absolute $\delta_{close} \geq \widetilde{\Omega}_*\left((\zeta + \lambda^2)^{1/3}\right)$.*

*Proof.* Assume towards contradiction that there exists some $i \in [m_*]$ with some $\delta_{close} \geq \widetilde{\Omega}\left((\frac{L(\mu)}{a_{\min}^2})^{1/(4c_\sigma - 2)}\right)$ with large enough hidden constant such that $\mu(\mathcal{T}_i(\delta)) \operatorname{sign}(a_i^*) \leq |a_i^*|/2$. For simplicity, we will use $\delta$ for $\delta_{close}$ in the following.

Let $g(x) = \sum_{\ell \leq k < 2\ell} \operatorname{sign}(a_i^*) \operatorname{sign}(\hat{\sigma}_k) h_k(\boldsymbol{w}_i^{*\top} \boldsymbol{x})$ be a test function, where $h_k(x)$ is the $k$-th normalized probabilistic Hermite polynomial and $\ell$ will be chosen later.

Denote $R(\boldsymbol{x}) = f_\mu(\boldsymbol{x}) - f_{\mu^*}(\boldsymbol{x})$ so that $\|R\|_2^2 = L(\mu)$. We have

$$\sqrt{L(\mu)}\,\|g\|_2 \geq \langle -R, g\rangle$$
$$= \mathbb{E}_{\boldsymbol{x}}\left[\left(a_i^*\sigma(\boldsymbol{w}_i^{*\top}\boldsymbol{x}) - \int_{\mathcal{T}_i(\delta)} \sigma(\boldsymbol{w}^\top\boldsymbol{x})\,\mathrm{d}\mu(\boldsymbol{w})\right)g(\boldsymbol{x})\right]$$
$$+ \mathbb{E}_{\boldsymbol{x}}\left[\left(\sum_{j\neq i} a_j^*\sigma(\boldsymbol{w}_j^{*\top}\boldsymbol{x}) - \int_{\mathbb{S}^{d-1}\setminus\mathcal{T}_i(\delta)} \sigma(\boldsymbol{w}^\top\boldsymbol{x})\,\mathrm{d}\mu(\boldsymbol{w})\right)g(\boldsymbol{x})\right].$$

Recall the Hermite expansion of $\sigma(x) = \sum_{k\geq 0}\hat{\sigma}_k h_k(x)$ and its property in Claim A.1. For the first term, it becomes

$$\sum_{\ell\leq k<2\ell}\left(|a_i^*||\hat{\sigma}_k| - \int_{\mathcal{T}_i(\delta)}|\hat{\sigma}_k|\,\mathrm{sign}(a_i^*)(\boldsymbol{w}^\top\boldsymbol{w}_i^*)^k\,\mathrm{d}\mu(\boldsymbol{w})\right) \geq \frac{1}{2}|a_i^*|\sum_{\ell\leq k<2\ell}|\hat{\sigma}_k|.$$

For the second term, it becomes

$$\sum_{\ell\leq k<2\ell}\left(\sum_{j\neq i}a_j^*|\hat{\sigma}_k|\,\mathrm{sign}(a_i^*)(\boldsymbol{w}_j^{*\top}\boldsymbol{w}_i^*)^k - \int_{\mathbb{S}^{d-1}\setminus\mathcal{T}_i(\delta)}|\hat{\sigma}_k|\,\mathrm{sign}(a_i^*)(\boldsymbol{w}^\top\boldsymbol{w}_i^*)^k\,\mathrm{d}\mu(\boldsymbol{w})\right)$$
$$\leq (\|\boldsymbol{a}^*\|_1 + |\mu|_1)\sum_{\ell\leq k\leq 2\ell}|\hat{\sigma}_k|\max_{\angle(\boldsymbol{w},\boldsymbol{w}_i^*)\geq\delta}(\boldsymbol{w}^\top\boldsymbol{w}_i^*)^k$$
$$\leq (\|\boldsymbol{a}^*\|_1 + |\mu|_1)\sum_{\ell\leq k<2\ell}|\hat{\sigma}_k|(1-\delta^2/5)^\ell$$
$$\leq 4\|\boldsymbol{a}^*\|_1(1-\delta^2/5)^\ell\sum_{\ell\leq k<2\ell}|\hat{\sigma}_k| \leq \frac{1}{4}|a_i^*|\sum_{\ell\leq k<2\ell}|\hat{\sigma}_k|,$$

where (i) in the third line we use $\cos\delta \leq 1-\delta^2/5$ for $\delta \in [0,\pi/2]$ and (ii) in the last line we use Lemma F.4 and choose $\ell = \lceil (5/\delta^2)\log(16\|\boldsymbol{a}^*\|_1/|a_i^*|)\rceil$.

Thus, given $|\hat{\sigma}_k| = \Theta(k^{-c_\sigma})$ we have

$$\sqrt{L(\mu)}\sqrt{\ell} = \sqrt{L(\mu)}\,\|g\|_2 \geq \frac{1}{4}|a_i^*|\sum_{\ell\leq k<2\ell}|\hat{\sigma}_k| = \frac{1}{4}|a_i^*|\sum_{\ell\leq k<2\ell}\Theta(k^{-c_\sigma}) = |a_i^*|\Theta(\ell^{1-c_\sigma}).$$

With the choice of $\ell = \widetilde{\Theta}(1/\delta^2)$, we have $\delta = \widetilde{O}\left(\left(\frac{L(\mu)}{|a_i^*|^2}\right)^{1/(4c_\sigma-2)}\right)$. Since $\delta \geq \widetilde{\Omega}\left((\frac{L(\mu)}{a_{\min}^2})^{1/(4c_\sigma-2)}\right)$ with a large enough hidden constant, we know this is a contradiction.

As a corollary, with Lemma F.4 that $L(\mu) = 4\zeta + 5\lambda^2\|p\|_2^2$, we have $\delta \geq \widetilde{\Omega}\left((\frac{4\zeta+5\lambda^2\|p\|_2^2}{a_{\min}^2})^{1/(4c_\sigma-2)}\right)$.

For the activation $\sigma$ is ReLU or absolute function, by Lemma A.1 we know $c_\sigma = 5/4$, which gives the desired result. $\qquad\square$

The lemma below bounds $R_2$ using the fact that it is spiky (has small non-zero support).

**Lemma F.8.** *Under Lemma 6, recall the optimality gap $\zeta = L_\lambda(\mu) - L_\lambda(\mu_\lambda^*)$. Then*

$$\|R_2\|_2^2 = O_*((\zeta/\lambda + \lambda)^{3/2}).$$

*Proof.* Using the same calculation as in Lemma 12 in Zhou et al. (2021), we have

$$\|R_2\|_2^2 \leq O(m_*)\sum_{i\in[m_*]}\left(\sum_{j\in\mathcal{T}_i}|a_j|\,\|\boldsymbol{w}_j\|_2\right)^{1/2}\left(\sum_{j\in\mathcal{T}_i}|a_j|\,\|\boldsymbol{w}_j\|_2\,\delta_j^2\right)^{3/2}$$

With Lemma F.4 and Lemma F.5, we have $\|R_2\|_2^2 = O(m_*^2|\mu^*|^{1/2}(\zeta/\lambda + \lambda)^{3/2})$. $\qquad\square$

The following lemma bounds $R_3$. In fact, in the view of expressing the loss as a sum of tensor decomposition problem, $R_3$ corresponds to the 0-th order term in the expansion. It would become small when high-order terms become small, as shown in the proof below.

**Lemma F.9.** *Under Lemma 6 and recall the optimality gap* $\zeta = L_\lambda(\mu) - L_\lambda(\mu_\lambda^*)$. *If* $\hat\sigma_0 = 0$ *and* $\hat\sigma_k > 0$ *with some* $k = \Theta((1/\Delta^2)\log(\zeta/\|\boldsymbol{a}_*\|_1))$, *then*

$$\|R_3\|_2 = \widetilde{O}_*((\zeta + \lambda^2)^{1/2}/\hat\sigma_k + (\zeta/\lambda + \lambda) + \zeta).$$

*Proof.* As shown in Ge et al. (2018); Li et al. (2020), we can write the loss $L(\mu)$ as sum of tensor decomposition problem (recall $\|\boldsymbol{w}_i^*\|_2 = 1$):

$$L(\mu) = \sum_{k \geq 0} \hat\sigma_k^2 \left\| \int_{\boldsymbol{w}\in\mathbb{S}^{d-1}} \boldsymbol{w}^{\otimes k} \, \mathrm{d}\mu(\boldsymbol{w}) - \sum_{i\in[m_*]} a_i^* \|\boldsymbol{w}_i^*\|_2 \, \boldsymbol{w}_i^{*\otimes k} \right\|_F^2.$$

Thus, we know for any $k \geq 1$,

$$\left\| \int_{\boldsymbol{w}\in\mathbb{S}^{d-1}} \boldsymbol{w}^{\otimes k} \, \mathrm{d}\mu(\boldsymbol{w}) - \sum_{i\in[m_*]} a_i^* \|\boldsymbol{w}_i^*\|_2 \, \boldsymbol{w}_i^{*\otimes k} \right\|_F^2 \leq L(\mu)/\hat\sigma_k^2.$$

Given any $\boldsymbol{w}_j^*$ and even $k$, we have

$$\left\| \int_{\boldsymbol{w}\in\mathbb{S}^{d-1}} \boldsymbol{w}^{\otimes k} \, \mathrm{d}\mu(\boldsymbol{w}) - \sum_{i\in[m_*]} a_i^* \|\boldsymbol{w}_i^*\|_2 \, \boldsymbol{w}_i^{*\otimes k} \right\|_F$$

$$\geq \left| \left\langle \sum_{i\in[m_*]} a_i^* \|\boldsymbol{w}_i^*\|_2 \, \boldsymbol{w}_i^{*\otimes k} - \int_{\boldsymbol{w}\in\mathbb{S}^{d-1}} \boldsymbol{w}^{\otimes k} \, \mathrm{d}\mu(\boldsymbol{w}), \boldsymbol{w}_j^{*\otimes k} \right\rangle \right|$$

$$\geq \left| a_j^* \|\boldsymbol{w}_j^*\|_2 - \int_{\mathcal{T}_j} \langle \boldsymbol{w}, \boldsymbol{w}_j^* \rangle^k \, \mathrm{d}\mu(\boldsymbol{w}) \right| - \left| \sum_{i\neq j} a_i^* \|\boldsymbol{w}_i^*\|_2 \langle \boldsymbol{w}_i^*, \boldsymbol{w}_j^* \rangle^k - \int_{\mathbb{S}^{d-1}\setminus\mathcal{T}_j} \langle \boldsymbol{w}, \boldsymbol{w}_j^* \rangle^k \, \mathrm{d}\mu(\boldsymbol{w}) \right|$$

$$\geq \left| a_j^* \|\boldsymbol{w}_j^*\|_2 - \int_{\mathcal{T}_j} \mathrm{d}\mu(\boldsymbol{w}) \right| - \left| \int_{\mathcal{T}_j} \mathrm{d}\mu(\boldsymbol{w}) - \int_{\mathcal{T}_j} \langle \boldsymbol{w}, \boldsymbol{w}_j^* \rangle^k \, \mathrm{d}\mu(\boldsymbol{w}) \right|$$

$$- \left| \sum_{i\neq j} a_i^* \|\boldsymbol{w}_i^*\|_2 \langle \boldsymbol{w}_i^*, \boldsymbol{w}_j^* \rangle^k - \int_{\mathbb{S}^{d-1}\setminus\mathcal{T}_j} \langle \boldsymbol{w}, \boldsymbol{w}_j^* \rangle^k \, \mathrm{d}\mu(\boldsymbol{w}) \right|$$

We show the last 2 terms are small.

For the second term on RHS, we have

$$\left| \int_{\mathcal{T}_j} \mathrm{d}\mu(\boldsymbol{w}) - \int_{\mathcal{T}_j} \langle \boldsymbol{w}, \boldsymbol{w}_j^* \rangle^k \, \mathrm{d}\mu(\boldsymbol{w}) \right| \leq \int_{\mathcal{T}_j} \left( 1 - \langle \boldsymbol{w}, \boldsymbol{w}_j^* \rangle^k \right) \, \mathrm{d}|\mu|(\boldsymbol{w}) \overset{(a)}{\leq} \int_{\mathcal{T}_j} 1 - (1 - \delta(\boldsymbol{w}, \boldsymbol{w}_j^*)^2/2)^k \, \mathrm{d}|\mu|(\boldsymbol{w})$$

$$\overset{(b)}{\leq} \int_{\mathcal{T}_j, \delta(\boldsymbol{w},\boldsymbol{w}_j^*)^2 \leq 1} O(k) \cdot \delta(\boldsymbol{w}, \boldsymbol{w}_j^*)^2 \, \mathrm{d}|\mu|(\boldsymbol{w}) + \int_{\mathcal{T}_j, \delta(\boldsymbol{w},\boldsymbol{w}_j^*)^2 > 1} \mathrm{d}|\mu|(\boldsymbol{w})$$

$$\leq O(k) \int_{\mathcal{T}_j} \delta(\boldsymbol{w}, \boldsymbol{w}_j^*)^2 \, \mathrm{d}|\mu|(\boldsymbol{w}),$$

where (a) $\cos\delta \geq 1 - \delta^2/2$ for $\delta \in [0, \pi/2]$; (b) $(1-x)^k \geq 1 - kx$ for $x \in [0, 1]$.

For the third term on RHS, we have

$$\left| \sum_{i\neq j} a_i^* \|\boldsymbol{w}_i^*\|_2 \langle \boldsymbol{w}_i^*, \boldsymbol{w}_j^* \rangle^k - \int_{\mathbb{S}^{d-1}\setminus\mathcal{T}_j} \langle \boldsymbol{w}, \boldsymbol{w}_j^* \rangle^k \, \mathrm{d}\mu(\boldsymbol{w}) \right| \leq (\|\boldsymbol{a}_*\|_1 + |\mu|_1) \max_{\angle(\boldsymbol{w},\boldsymbol{w}_j^*) \geq \Delta/2} (\boldsymbol{w}^\top \boldsymbol{w}_j^*)^k$$

$$\overset{(a)}{\leq} (\|\boldsymbol{a}_*\|_1 + |\mu|_1)(1 - \Delta^2/10)^k \overset{(b)}{\leq} O(\zeta),$$

where (a) $\cos \delta \leq 1 - \delta^2/5$ for $\delta \in [0, \pi/2]$; (b) we choose $k = \Theta((1/\Delta^2) \log(\zeta/\|\boldsymbol{a}_*\|_1))$ and Lemma F.4.

Therefore, we have

$$\left\| \int_{\boldsymbol{w} \in \mathbb{S}^{d-1}} \boldsymbol{w}^{\otimes k} \mu(\boldsymbol{w}) - \sum_{i \in [m_*]} a_i^* \|\boldsymbol{w}_i^*\|_2 \, \boldsymbol{w}_i^{*\otimes k} \right\|_F$$

$$\geq \left| a_j^* \|\boldsymbol{w}_j^*\|_2 - \int_{\mathcal{T}_j} \mu(\boldsymbol{w}) \right| - O(k) \int_{\mathcal{T}_j} \delta(\boldsymbol{w}, \boldsymbol{w}_j^*)^2 |\mu|(\boldsymbol{w}) - O(\zeta).$$

This implies that

$$m_* \sqrt{L(\mu)}/\hat{\sigma}_k \geq \sum_{j \in [m_*]} \left| a_j^* \|\boldsymbol{w}_j^*\|_2 - \int_{\mathcal{T}_j} \mu(\boldsymbol{w}) \right| - O(k) \sum_{j \in [m_*]} \int_{\mathcal{T}_j} \delta(\boldsymbol{w}, \boldsymbol{w}_j^*)^2 |\mu|(\boldsymbol{w}) - O(m_* \zeta)$$

$$\geq \left| \sum_{i \in [m_*]} a_i^* \|\boldsymbol{w}_i^*\|_2 - \int_{\mathbb{S}^{d-1}} \mu(\boldsymbol{w}) \right| - \widetilde{O}_*(\zeta/\lambda + \lambda) - O(m_* \zeta),$$

where we use Lemma F.5. Rearranging the terms and recalling $L(\mu) = O_*(\zeta + \lambda^2)$ from Lemma F.4, we get the bound.

$\square$

The following lemma gives the bound on the average neuron to its corresponding teacher neuron. It follows directly from the residual decomposition and previous lemmas that characterize $R_1, R_2, R_3$ respectively.

**Lemma F.10.** *Under Lemma 6, recall the optimality gap $\zeta = L_\lambda(\mu) - L_\lambda(\mu_\lambda^*)$. Then for any $i \in [m_*]$, $\zeta = \Omega(\lambda^2)$ and $\zeta, \lambda \leq 1/\operatorname{poly}(m_*, \Delta, \|\boldsymbol{a}_*\|_1)$*

$$\left\| \sum_{j \in \mathcal{T}_i} a_j \boldsymbol{w}_j - \boldsymbol{w}_i^* \right\|_2 \leq \left( \sum_{i \in [m_*]} \left\| \sum_{j \in \mathcal{T}_i} a_j \boldsymbol{w}_j - \boldsymbol{w}_i^* \right\|_2^2 \right)^{1/2} = O_*((\zeta/\lambda)^{3/4}).$$

*Proof.* With the relation of residual decomposition, Lemma F.7, Lemma F.8 and Lemma F.9, we have for any $i \in [m_*]$

$$\Omega(\Delta^{3/2}/m_*^{3/2}) \left( \sum_{i \in [m_*]} \left\| \sum_{j \in \mathcal{T}_i} a_j \boldsymbol{w}_j - \boldsymbol{w}_i^* \right\|_2^2 \right)^{1/2} \leq \|R_1\|_2 \leq \|R\|_2 + \|R_2\|_2 + \|R_3\|_2$$

$$= O_*((\zeta + \lambda^2)^{1/2} + (\zeta/\lambda + \lambda)^{3/4}) + \widetilde{O}_*((\zeta + \lambda^2)^{1/2} + (\zeta/\lambda + \lambda) + \zeta).$$

Rearranging the terms, we get the result. $\square$

## H.2 Omitted proofs in Section F.2

In this section, we give the omitted proofs in Section F.2. The key observation used in the proofs is that balancing the norm and setting $\alpha, \boldsymbol{\beta}$ perfectly to their target values only decrease the optimality gap.

**Lemma F.11.** *Given any $\boldsymbol{\theta} = (\boldsymbol{a}, \boldsymbol{W}, \alpha, \boldsymbol{\beta})$ satisfying $|\alpha - \hat{\alpha}|^2 = O(\zeta)$, $\left\| \boldsymbol{\beta} - \hat{\boldsymbol{\beta}} \right\|_2^2 = O(\zeta)$, where $\hat{\alpha} = -(1/\sqrt{2\pi}) \sum_{i=1}^m a_i \|\boldsymbol{w}_i\|_2$ and $\hat{\boldsymbol{\beta}} = -(1/2) \sum_{i=1}^m a_i \boldsymbol{w}_i$. Let its corresponding balanced version $\boldsymbol{\theta}_{bal} = (\boldsymbol{a}_{bal}, \boldsymbol{W}_{bal}, \alpha_{bal}, \boldsymbol{\beta}_{bal})$ as $a_{bal,i} = \operatorname{sign}(a_i)\sqrt{|a_i| \|\boldsymbol{w}_i\|_2}$, $\boldsymbol{w}_{bal,i} = \overline{\boldsymbol{w}}_i \sqrt{|a_i| \|\boldsymbol{w}_i\|_2}$, $\alpha_{bal} = \hat{\alpha}$ and $\boldsymbol{\beta}_{bal} = \hat{\boldsymbol{\beta}}$. Then, we have*

$$L_\lambda(\boldsymbol{\theta}) - L_\lambda(\boldsymbol{\theta}_{bal}) = |\alpha - \hat{\alpha}|^2 + \left\| \boldsymbol{\beta} - \hat{\boldsymbol{\beta}} \right\|_2^2 + \frac{\lambda}{2} \sum_{i \in [m]} (|a_i| - \|\boldsymbol{w}_i\|_2)^2 \geq 0.$$

*Moreover, let the optimality gap $\zeta = L_\lambda(\boldsymbol{\theta}) - L_\lambda(\mu_\lambda^*)$, we have results in Lemma F.4, Lemma F.5, Lemma F.6, Lemma F.7, Lemma F.8, Lemma F.9 and Lemma F.10 still hold for $L_\lambda(\boldsymbol{\theta})$, with the change of $R_3$ in (8) as*

$$R_3(\boldsymbol{x}) = \frac{1}{\sqrt{2\pi}} \left( \sum_{i \in [m_*]} a_i^* \left\| \boldsymbol{w}_i^* \right\|_2 - \sum_{i \in [m]} a_i \left\| \boldsymbol{w}_i \right\|_2 \right) + \alpha - \hat{\alpha} + (\boldsymbol{\beta} - \hat{\boldsymbol{\beta}})^\top \boldsymbol{x}.$$

*Proof.* Recall in Claim B.1 we have

$$L(\boldsymbol{\theta}) = |\alpha - \hat{\alpha}|^2 + \left\| \boldsymbol{\beta} - \hat{\boldsymbol{\beta}} \right\|_2^2 + \sum_{k \geq 2} \hat{\sigma}_k^2 \left\| \sum_{i \in [m]} a_i \left\| \boldsymbol{w}_i \right\|_2 \overline{\boldsymbol{w}}_i^{\otimes k} - \sum_{i \in [m_*]} a_i^* \left\| \boldsymbol{w}_i^* \right\|_2 \boldsymbol{w}_i^{*\otimes k} \right\|_F^2.$$

Note that $|a_i| \left\| \boldsymbol{w}_i \right\|_2 = |a_{bal,i}| \left\| \boldsymbol{w}_{bal,i} \right\|_2$ so that $L(\boldsymbol{\theta}) = L(\boldsymbol{\theta}_{bal}) + |\alpha - \hat{\alpha}|^2 + \left\| \boldsymbol{\beta} - \hat{\boldsymbol{\beta}} \right\|_2^2$. We then have

$$\begin{aligned}
L_\lambda(\boldsymbol{\theta}) - L_\lambda(\boldsymbol{\theta}_{bal}) =& |\alpha - \hat{\alpha}|^2 + \left\| \boldsymbol{\beta} - \hat{\boldsymbol{\beta}} \right\|_2^2 + \frac{\lambda}{2} \left\| \boldsymbol{a} \right\|_2^2 + \frac{\lambda}{2} \left\| \boldsymbol{W} \right\|_2^2 - \frac{\lambda}{2} \left\| \boldsymbol{a}_{bal} \right\|_2^2 - \frac{\lambda}{2} \left\| \boldsymbol{W}_{bal} \right\|_2^2 \\
=& |\alpha - \hat{\alpha}|^2 + \left\| \boldsymbol{\beta} - \hat{\boldsymbol{\beta}} \right\|_2^2 + \frac{\lambda}{2} \sum_{i \in [m]} (|a_i| - \left\| \boldsymbol{w}_i \right\|_2)^2.
\end{aligned}$$

Therefore, we have the optimality gap $\zeta = L_\lambda(\boldsymbol{\theta}) - L_\lambda(\mu_\lambda^*) \geq L_\lambda(\boldsymbol{\theta}_{bal}) - L_\lambda(\mu_\lambda^*) = \zeta_{bal}$. Note that $\boldsymbol{\theta}_{bal}$ corresponds to a network that has perfect balanced norms and fitted $\alpha, \boldsymbol{\beta}$, thus all results in Lemma F.4, Lemma F.5, Lemma F.6, Lemma F.7, Lemma F.8, Lemma F.9 and Lemma F.10 hold for $\boldsymbol{\theta}_{bal}$. Since $\zeta \geq \zeta_{bal}$, $|a_i| \left\| \boldsymbol{w}_i \right\|_2 = |a_{bal,i}| \left\| \boldsymbol{w}_{bal,i} \right\|_2$ and $L(\boldsymbol{\theta}) = L(\boldsymbol{\theta}_{bal}) + O(\zeta)$, we can easily check that all of them also hold for $\boldsymbol{\theta}$. For the bound on $R_3$, note that

$$\left\| R_3 \right\|_2 \leq \frac{1}{\sqrt{2\pi}} \left| \sum_{i \in [m_*]} a_i^* \left\| \boldsymbol{w}_i^* \right\|_2 - \sum_{i \in [m]} a_i \left\| \boldsymbol{w}_i \right\|_2 \right| + |\alpha - \hat{\alpha}| + \left\| \boldsymbol{\beta} - \hat{\boldsymbol{\beta}} \right\|_2$$

so that the same bound still hold for $R_3$. $\qquad\square$

**Lemma F.12.** *Under Lemma 6, suppose optimality gap $\zeta = L_\lambda(\boldsymbol{\theta}) - L_\lambda(\mu_\lambda^*)$. Then $\left\| \boldsymbol{a} \right\|_2^2 + \left\| \boldsymbol{W} \right\|_F^2 \leq 3 \left\| \boldsymbol{a}_* \right\|_1$.*

*Proof.* We have

$$\frac{\lambda}{2} \left\| \boldsymbol{a} \right\|_2^2 + \frac{\lambda}{2} \left\| \boldsymbol{W} \right\|_F^2 = \zeta + L(\mu_\lambda^*) + \lambda |\mu_\lambda^*|_1 - L(\boldsymbol{\theta}) \leq \zeta + \lambda^2 \left\| p \right\|_2^2 + \lambda |\mu_\lambda^*|_1,$$

where we use Lemma F.3. Rearranging the terms, we get the result by noting that $|\mu_\lambda^*|_1 \leq \left\| \boldsymbol{a}_* \right\|_1$. $\qquad\square$

### H.3  Omitted proofs in Section F.3

In this section, we give the omitted proofs in Section F.3. We will consider them case by case.

The lemma below says that one can always decrease the loss if norms are not balanced.

**Lemma F.15** (Descent direction, norm balance). *We have*

$$\sum_i \sum_{j \in T_i} \left| \langle \nabla_{a_j} L_\lambda, -a_j \rangle + \langle \nabla_{\boldsymbol{w}_j} L_\lambda, \boldsymbol{w}_j \rangle \right| = \lambda \sum_{i \in [m_*]} \left| a_i^2 - \left\| \boldsymbol{w}_i \right\|_2^2 \right|$$

$$\geq \max \left\{ \lambda \left| \left\| \boldsymbol{a} \right\|_2^2 - \left\| \boldsymbol{W} \right\|_F^2 \right|, \lambda \sum_{i \in [m_*]} (|a_i| - \left\| \boldsymbol{w}_i \right\|_2)^2 \right\}$$

*Proof.* We have

$$\sum_{i \in [m]} \left| \langle \nabla_{a_j} L_\lambda, -a_j \rangle + \langle \nabla_{\boldsymbol{w}_j} L_\lambda, \boldsymbol{w}_j \rangle \right|$$

$$= \sum_{i \in [m]} \left| -2\mathbb{E}_{\boldsymbol{x}}[(f(\boldsymbol{x}) - f_*(\boldsymbol{x}))a_j \sigma(\boldsymbol{w}_j^\top \boldsymbol{x})] - \lambda a_j^2 + 2\mathbb{E}_{\boldsymbol{x}}[(f(\boldsymbol{x}) - f_*(\boldsymbol{x}))a_j \sigma(\boldsymbol{w}_j^\top \boldsymbol{x})] + \lambda \|\boldsymbol{w}_i\|_2^2 \right|$$

$$= \lambda \sum_{i \in [m]} \left| a_i^2 - \|\boldsymbol{w}_i\|_2^2 \right|$$

Note that $|a_i| + \|\boldsymbol{w}_i\|_2 \geq | |a_i| - \|\boldsymbol{w}_i\|_2 |$, we get the result. $\qquad \square$

The following lemma shows that one can always decrease the loss if there are close-by neurons that cancels with others. Intuitively, reducing such norm cancellation decrease the regularization term while keeping the square loss term, which decreasing the total loss as a whole.

**Lemma F.16** (Descent direction, norm cancellation). *Under Lemma 6 and Assumption F.1, suppose the optimality gap $\zeta = L_\lambda(\boldsymbol{\theta}) - L_\lambda(\mu_\lambda^*)$. For any $\boldsymbol{w}_i^*$, consider $\delta_{\text{sign}}$ such that $\delta_{close} < \delta_{\text{sign}} = O(\lambda/\zeta^{1/2})$ with small enough hidden constant ($\delta_{close}$ defined in Lemma F.6), then*

$$\sum_{s \in \{+,-\}} \sum_{j \in T_{i,s}(\delta_{\text{sign}})} \left\langle \nabla_{a_j} L_\lambda, \frac{a_j}{\sum_{j \in T_{i,s}(\delta_{\text{sign}})} |a_j| \|\boldsymbol{w}_j\|_2} \right\rangle + \left\langle \nabla_{\boldsymbol{w}_j} L_\lambda, \frac{\boldsymbol{w}_j}{\sum_{j \in T_{i,s}(\delta_{\text{sign}})} |a_j| \|\boldsymbol{w}_j\|_2} \right\rangle = \Omega(\lambda).$$

*where $T_{i,+}(\delta_{\text{sign}}) = \{j \in T_i : \delta(\boldsymbol{w}_j, \boldsymbol{w}_i^*) \leq \delta_{\text{sign}}, \text{sign}(a_j) = \text{sign}(a_i^*)\}$, $T_{i,-}(\delta_{\text{sign}}) = \{j \in T_i : \delta(\boldsymbol{w}_j, \boldsymbol{w}_i^*) \leq \delta_{\text{sign}}, \text{sign}(a_j) \neq \text{sign}(a_i^*)\}$ are the set of neurons that close to $\boldsymbol{w}_i^*$ with/without same sign of $a_i^*$.*

*As a result,*

$$\|\nabla_{\boldsymbol{a}} L_\lambda\|_2^2 + \|\nabla_{\boldsymbol{W}} L_\lambda\|_F^2 \geq \lambda^2 \sum_{j \in T_{i,-}(\delta_{\text{sign}})} |a_j| \|\boldsymbol{w}_j\|_2$$

*Proof.* Denote $R(\boldsymbol{x}) = f(\boldsymbol{x}) - \widetilde{f}_*(\boldsymbol{x})$. We have

$$\sum_{s \in \{+,-\}} \sum_{j \in T_{i,s}(\delta_{\text{sign}})} \left\langle \nabla_{a_j} L_\lambda, \frac{a_j}{\sum_{j \in T_{i,s}(\delta_{\text{sign}})} |a_j| \|\boldsymbol{w}_j\|_2} \right\rangle + \left\langle \nabla_{\boldsymbol{w}_j} L_\lambda, \frac{\boldsymbol{w}_j}{\sum_{j \in T_{i,s}(\delta_{\text{sign}})} |a_j| \|\boldsymbol{w}_j\|_2} \right\rangle$$

$$= \sum_{s \in \{+,-\}} \sum_{j \in T_{i,s}(\delta_{\text{sign}})} \frac{a_j \|\boldsymbol{w}_j\|_2}{\sum_{j \in T_{i,s}(\delta_{\text{sign}})} |a_j| \|\boldsymbol{w}_j\|_2} \cdot 2\mathbb{E}_{\boldsymbol{x}}[R(\boldsymbol{x})\sigma(\overline{\boldsymbol{w}}_j^\top \boldsymbol{x})] + \frac{\lambda a_j^2}{\sum_{j \in T_{i,s}(\delta_{\text{sign}})} |a_j| \|\boldsymbol{w}_j\|_2}$$

$$+ \sum_{s \in \{+,-\}} \sum_{j \in T_{i,s}(\delta_{\text{sign}})} \frac{a_j \|\boldsymbol{w}_j\|_2}{\sum_{j \in T_{i,s}(\delta_{\text{sign}})} |a_j| \|\boldsymbol{w}_j\|_2} \cdot 2\mathbb{E}_{\boldsymbol{x}}[R(\boldsymbol{x})\sigma(\overline{\boldsymbol{w}}_j^\top \boldsymbol{x})] + \frac{\lambda \|\boldsymbol{w}_j\|_2^2}{\sum_{j \in T_{i,s}(\delta_{\text{sign}})} |a_j| \|\boldsymbol{w}_j\|_2}$$

We split the above into two terms (depending on square loss or regularization). WLOG, assume $\text{sign}(a_i^*) = 1$. For the first term that depends on gradient on square loss,

$$(I) = 4 \sum_{s \in \{+,-\}} \sum_{j \in T_{i,s}(\delta_{\text{sign}})} \frac{a_j \|\boldsymbol{w}_j\|_2}{\sum_{j \in T_{i,s}(\delta_{\text{sign}})} |a_j| \|\boldsymbol{w}_j\|_2} \cdot \mathbb{E}_{\boldsymbol{x}}[R(\boldsymbol{x})\sigma(\overline{\boldsymbol{w}}_j^\top \boldsymbol{x})]$$

$$= 4 \sum_{j \in T_{i,+}(\delta_{\text{sign}})} \frac{|a_j| \|\boldsymbol{w}_j\|_2}{\sum_{j \in T_{i,+}(\delta_{\text{sign}})} |a_j| \|\boldsymbol{w}_j\|_2} \mathbb{E}_{\boldsymbol{x}}[R(\boldsymbol{x})\sigma(\overline{\boldsymbol{w}}_j^\top \boldsymbol{x})]$$

$$- 4 \sum_{j \in T_{i,-}(\delta_{\text{sign}})} \frac{|a_j| \|\boldsymbol{w}_j\|_2}{\sum_{j \in T_{i,-}(\delta_{\text{sign}})} |a_j| \|\boldsymbol{w}_j\|_2} \mathbb{E}_{\boldsymbol{x}}[R(\boldsymbol{x})\sigma(\overline{\boldsymbol{w}}_j^\top \boldsymbol{x})]$$

$$= 4 \sum_{j \in T_{i,+}(\delta_{\text{sign}})} \frac{|a_j| \|\boldsymbol{w}_j\|_2}{\sum_{j \in T_{i,+}(\delta_{\text{sign}})} |a_j| \|\boldsymbol{w}_j\|_2} \mathbb{E}_{\boldsymbol{x}}[R(\boldsymbol{x})(\sigma(\overline{\boldsymbol{w}}_j^\top \boldsymbol{x}) - \sigma(\overline{\boldsymbol{w}}_i^{*\top} \boldsymbol{x}))]$$

$$- 4 \sum_{j \in T_{i,-}(\delta_{\text{sign}})} \frac{|a_j| \|\boldsymbol{w}_j\|_2}{\sum_{j \in T_{i,-}(\delta_{\text{sign}})} |a_j| \|\boldsymbol{w}_j\|_2} \mathbb{E}_{\boldsymbol{x}}[R(\boldsymbol{x})(\sigma(\overline{\boldsymbol{w}}_j^\top \boldsymbol{x}) - \sigma(\overline{\boldsymbol{w}}_i^{*\top} \boldsymbol{x}))]$$

Since $\overline{\boldsymbol{w}}_j$ is $\delta_{\text{sign}}$-close to $\boldsymbol{w}_i^*$ and $\|R\|_2^2 = L(\boldsymbol{\theta})$, we have

$$|(I)| \leq O(\delta_{\text{sign}}) \|R\|_2 = O_*(\delta_{\text{sign}} \zeta^{1/2}),$$

where we use Lemma F.11 that $L(\boldsymbol{\theta}) = O_*(\zeta)$.

For the second term that depends on regularization, we have

$$(II) = \lambda \sum_{s \in \{+,-\}} \frac{\sum_{j \in T_{i,s}(\delta_{\text{sign}})} a_j^2 + \|\boldsymbol{w}_j\|_2^2}{\sum_{j \in T_{i,s}(\delta_{\text{sign}})} |a_j| \|\boldsymbol{w}_j\|_2} \geq 2\lambda + 2\lambda = 4\lambda.$$

Therefore, when $(I) \leq 2\lambda$, i.e., $\delta_{\text{sign}} = O_*(\lambda/\zeta^{1/2})$, we have

$$\sum_{s \in \{+,-\}} \sum_{j \in T_{i,s}(\delta_{\text{sign}})} \left\langle \nabla_{a_j} L_\lambda, \frac{\text{sign}(a_j)|a_j|}{\sum_{j \in T_{i,s}(\delta_{\text{sign}})} |a_j| \|\boldsymbol{w}_j\|_2} \right\rangle + \left\langle \nabla_{\boldsymbol{w}_j} L_\lambda, \frac{\boldsymbol{w}_j}{\sum_{j \in T_{i,s}(\delta_{\text{sign}})} |a_j| \|\boldsymbol{w}_j\|_2} \right\rangle$$

$$\geq \frac{\lambda}{2} \sum_{s \in \{+,-\}} \frac{\sum_{j \in T_{i,s}(\delta_{\text{sign}})} a_j^2 + \|\boldsymbol{w}_j\|_2^2}{\sum_{j \in T_{i,s}(\delta_{\text{sign}})} |a_j| \|\boldsymbol{w}_j\|_2}.$$

We compute a upper bound for LHS. Note that

$$\sum_{s \in \{+,-\}} \sum_{j \in T_{i,s}(\delta_{\text{sign}})} \left\langle \nabla_{a_j} L_\lambda, \frac{a_j}{\sum_{j \in T_{i,s}(\delta_{\text{sign}})} |a_j| \|\boldsymbol{w}_j\|_2} \right\rangle + \left\langle \nabla_{\boldsymbol{w}_j} L_\lambda, \frac{\boldsymbol{w}_j}{\sum_{j \in T_{i,s}(\delta_{\text{sign}})} |a_j| \|\boldsymbol{w}_j\|_2} \right\rangle$$

$$\leq \sqrt{\sum_{s \in \{+,-\}} \sum_{j \in T_{i,s}(\delta_{\text{sign}})} (\nabla_{a_j} L_\lambda)^2 + \|\nabla_{\boldsymbol{w}_j} L_\lambda\|_2^2} \sqrt{\sum_{s \in \{+,-\}} \sum_{j \in T_{i,s}(\delta_{\text{sign}})} \frac{a_j^2 + \|\boldsymbol{w}_j\|_2^2}{(\sum_{j \in T_{i,s}(\delta_{\text{sign}})} |a_j| \|\boldsymbol{w}_j\|_2)^2}}$$

$$\leq \sqrt{\|\nabla_{\boldsymbol{a}} L_\lambda\|_2^2 + \|\nabla_{\boldsymbol{W}} L_\lambda\|_F^2} \sqrt{\sum_{s \in \{+,-\}} \frac{\sum_{j \in T_{i,s}(\delta_{\text{sign}})} a_j^2 + \|\boldsymbol{w}_j\|_2^2}{(\sum_{j \in T_{i,s}(\delta_{\text{sign}})} |a_j| \|\boldsymbol{w}_j\|_2)^2}}$$

$$\leq \sqrt{\|\nabla_{\boldsymbol{a}} L_\lambda\|_2^2 + \|\nabla_{\boldsymbol{W}} L_\lambda\|_F^2} \sqrt{\sum_{s \in \{+,-\}} \frac{\sum_{j \in T_{i,s}(\delta_{\text{sign}})} a_j^2 + \|\boldsymbol{w}_j\|_2^2}{\sum_{j \in T_{i,s}(\delta_{\text{sign}})} |a_j| \|\boldsymbol{w}_j\|_2} \frac{1}{\sqrt{\sum_{j \in T_{i,-}(\delta_{\text{sign}})} |a_j| \|\boldsymbol{w}_j\|_2}}},$$

where the last line we use Lemma F.6: $\sum_{j \in T_{i,-}(\delta_{\text{sign}})} |a_j| \|\boldsymbol{w}_j\|_2 < \sum_{j \in T_{i,+}(\delta_{\text{sign}})} |a_j| \|\boldsymbol{w}_j\|_2$ because $\mu(T_i(\delta)) = \sum_{j \in T_i(\delta_{\text{sign}})} a_j \|\boldsymbol{w}_j\|_2 > 0$.

Combine with the above descent direction, we have

$$\sqrt{\|\nabla_{\boldsymbol{a}} L_\lambda\|_2^2 + \|\nabla_{\boldsymbol{W}} L_\lambda\|_F^2} \sqrt{\sum_{s \in \{+,-\}} \frac{\sum_{j \in T_{i,s}(\delta_{\text{sign}})} a_j^2 + \|\boldsymbol{w}_j\|_2^2}{\sum_{j \in T_{i,s}(\delta_{\text{sign}})} |a_j| \|\boldsymbol{w}_j\|_2} \frac{1}{\sqrt{\sum_{j \in T_{i,-}(\delta_{\text{sign}})} |a_j| \|\boldsymbol{w}_j\|_2}}}$$

$$\geq \frac{\lambda}{2} \sum_{s \in \{+,-\}} \frac{\sum_{j \in T_{i,s}(\delta_{\text{sign}})} a_j^2 + \|\boldsymbol{w}_j\|_2^2}{\sum_{j \in T_{i,s}(\delta_{\text{sign}})} |a_j| \|\boldsymbol{w}_j\|_2},$$

which implies

$$\|\nabla_{\boldsymbol{a}} L_\lambda\|_2^2 + \|\nabla_{\boldsymbol{W}} L_\lambda\|_F^2 \geq \lambda^2 \sum_{j \in T_{i,-}(\delta_{\text{sign}})} |a_j| \|\boldsymbol{w}_j\|_2$$

$\square$

The lemma below shows that when all previous cases are not hold, then there is a descent direction that move all close-by neurons towards their corresponding teacher neuron. The proof relies on calculations that generalize Lemma 8 in Zhou et al. (2021).

**Lemma F.17** (Descent direction). *Under Lemma 6 and Assumption F.1, suppose the optimality gap* $\zeta = L_\lambda(\boldsymbol{\theta}) - L_\lambda(\mu_\lambda^*)$. *Suppose*

(i) *norms are (almost) balanced:* $|\|\boldsymbol{W}\|_F^2 - \|\boldsymbol{a}\|_2^2| \leq \zeta/\lambda$, $\sum_{i\in[m]}(|a_j| - \|\boldsymbol{w}_j\|_2)^2 = O_*(\zeta^2/\lambda^2)$

(ii) *(almost) no norm cancellation: consider all neurons $\boldsymbol{w}_j$ that are $\delta_{\text{sign}}$-close w.r.t. teacher neuron $\boldsymbol{w}_i^*$ but has a different sign, i.e.,* $\text{sign}(a_j) \neq \text{sign}(a_i^*)$ *with* $\delta_{\text{sign}} = \Theta_*(\lambda/\zeta^{1/2})$*, we have* $\sum_{j\in T_{i,-}(\delta_{\text{sign}})} |a_j| \|\boldsymbol{w}_j\|_2 \leq \tau = O_*(\zeta^{5/6}/\lambda)$ *with small enough hidden constant, where $T_{i,-}(\delta)$ defined in Lemma F.16.*

(iii) $\alpha, \boldsymbol{\beta}$ *are well fitted:* $|\alpha - \hat{\alpha}|^2 = O_*(\zeta)$, $\left\|\boldsymbol{\beta} - \hat{\boldsymbol{\beta}}\right\|_2^2 = O_*(\zeta)$ *with small enough hidden factor.*

*Then, we can construct the following descent direction*

$$(\alpha + \alpha_*)\nabla_\alpha L_\lambda + \langle \nabla_{\boldsymbol{\beta}} L_\lambda, \boldsymbol{\beta} + \boldsymbol{\beta}_* \rangle + \sum_{i\in[m_*]}\sum_{j\in\mathcal{T}_i} \langle \nabla_{\boldsymbol{w}_i} L_\lambda, \boldsymbol{w}_j - q_{ij}\boldsymbol{w}_i^* \rangle = \Omega(\zeta),$$

*where $q_{ij}$ satisfy the following conditions with $\delta_{close} < \delta_{\text{sign}}$ and $\delta_{close} = O_*(\zeta^{1/3})$: (1) $\sum_{j\in\mathcal{T}_i} a_j q_{ij} = a_i^*$; (2) $q_{ij} \geq 0$; (3) $q_{ij} = 0$ when $\text{sign}(a_j) \neq \text{sign}(a_i^*)$ or $\delta_j > \delta_{close}$. (4) $\sum_{i\in[m_*]}\sum_{j\in\mathcal{T}_i} q_{ij}^2 = O_*(1)$.*

*Proof.* Recall residual $R(\boldsymbol{x}) = f(\boldsymbol{x}) - \widetilde{f}_*(\boldsymbol{x})$. We have

$$(\alpha + \alpha_*)\nabla_\alpha L_\lambda + \langle \nabla_{\boldsymbol{\beta}} L_\lambda, \boldsymbol{\beta} + \boldsymbol{\beta}_* \rangle + \sum_{i\in[m_*]}\sum_{j\in\mathcal{T}_i} \langle \nabla_{\boldsymbol{w}_i} L_\lambda, \boldsymbol{w}_j - q_{ij}\boldsymbol{w}_i^* \rangle$$

$$\overset{(a)}{=} 2\mathbb{E}_{\boldsymbol{x}}[R(\boldsymbol{x})(\alpha + \alpha_*)] + 2\mathbb{E}_{\boldsymbol{x}}[R(\boldsymbol{x})(\boldsymbol{\beta} + \boldsymbol{\beta}_*)^\top \boldsymbol{x}]$$

$$+ 2\sum_{i\in[m_*]}\sum_{j\in\mathcal{T}_i}\mathbb{E}_{\boldsymbol{x}}[R(\boldsymbol{x})a_j\sigma(\boldsymbol{w}_j^\top \boldsymbol{x})] - 2\sum_{i\in[m_*]}\sum_{j\in\mathcal{T}_i}\mathbb{E}_{\boldsymbol{x}}[R(\boldsymbol{x})a_j q_{ij}\sigma(\boldsymbol{w}_i^{*\top} \boldsymbol{x})]$$

$$+ 2\sum_{i\in[m_*]}\sum_{j\in\mathcal{T}_i}\mathbb{E}_{\boldsymbol{x}}[R(\boldsymbol{x})a_j q_{ij}\boldsymbol{w}_i^{*\top}\boldsymbol{x}(\sigma'(\boldsymbol{w}_i^{*\top}\boldsymbol{x}) - \sigma'(\boldsymbol{w}_i^\top\boldsymbol{x}))]$$

$$+ \lambda\sum_{i\in[m]}\|\boldsymbol{w}_j\|_2^2 - \lambda\sum_{i\in[m_*]}\sum_{j\in\mathcal{T}_i} q_{ij}\boldsymbol{w}_j^\top\boldsymbol{w}_i^*$$

$$\overset{(b)}{=} 2\|R\|_2^2 + \lambda\|\boldsymbol{W}\|_F^2 - \lambda\sum_{i\in[m_*]}\sum_{j\in\mathcal{T}_i} q_{ij}\boldsymbol{w}_j^\top\boldsymbol{w}_i^*$$

$$+ 2\sum_{i\in[m_*]}\sum_{j\in\mathcal{T}_i}\mathbb{E}_{\boldsymbol{x}}[R(\boldsymbol{x})a_j q_{ij}\boldsymbol{w}_i^{*\top}\boldsymbol{x}(\sigma'(\boldsymbol{w}_i^{*\top}\boldsymbol{x}) - \sigma'(\boldsymbol{w}_j^\top\boldsymbol{x}))]$$

$$\overset{(c)}{\geq} L_\lambda(\mu_\lambda^*) + \zeta + \frac{\lambda}{2}(\|\boldsymbol{W}\|_F^2 - \|\boldsymbol{a}\|_2^2) - \lambda\sum_{i\in[m_*]}\sum_{j\in\mathcal{T}_i} q_{ij}\|\boldsymbol{w}_j\|_2$$

$$+ 2\sum_{i\in[m_*]}\sum_{j\in\mathcal{T}_i}\mathbb{E}_{\boldsymbol{x}}[R(\boldsymbol{x})a_j q_{ij}\boldsymbol{w}_i^{*\top}\boldsymbol{x}(\sigma'(\boldsymbol{w}_i^{*\top}\boldsymbol{x}) - \sigma'(\boldsymbol{w}_j^\top\boldsymbol{x}))], \tag{12}$$

where (a) we plug in the gradient expression and add and minus the term $2\sum_{i\in[m_*]}\sum_{j\in\mathcal{T}_i}\mathbb{E}_{\boldsymbol{x}}[R(\boldsymbol{x})a_j q_{ij}\sigma(\boldsymbol{w}_i^{*\top}\boldsymbol{x})]$; (b) rearranging the terms; (c) using $L_\lambda(\boldsymbol{\theta}) = \|R\|_2^2 + (\lambda/2)\|\boldsymbol{W}\|_F^2 + (\lambda/2)\|\boldsymbol{a}\|_2^2 = L_\lambda(\mu_\lambda^*) + \zeta$.

For the first line on RHS of (12), we have

$$L_\lambda(\mu_\lambda^*) + \zeta + \frac{\lambda}{2}(\|\boldsymbol{W}\|_F^2 - \|\boldsymbol{a}\|_2^2) - \lambda \sum_{i\in[m_*]} \sum_{j\in\mathcal{T}_i} q_{ij} \|\boldsymbol{w}_j\|_2$$

$$\overset{(a)}{\geq} \zeta/2 + L(\mu_\lambda^*) + \lambda|\mu_\lambda^*| - \lambda \sum_{i\in[m_*]} \sum_{j\in\mathcal{T}_i} q_{ij} \|\boldsymbol{w}_j\|_2$$

$$\overset{(b)}{\geq} \zeta/2 + \lambda|\mu_\lambda^*| - \lambda \|\boldsymbol{a}_*\|_1 + \lambda \sum_{i\in[m_*]} \sum_{j\in\mathcal{T}_i} q_{ij}(|a_j| - \|\boldsymbol{w}_j\|_2)$$

$$\overset{(c)}{\geq} \zeta/2 - O_*(\lambda^2) - \lambda \left(\sum_{i\in[m_*]} \sum_{j\in\mathcal{T}_i} q_{ij}^2\right)^{1/2} \left(\sum_{i\in[m]} (|a_j| - \|\boldsymbol{w}_j\|_2)^2\right)^{1/2} \overset{(d)}{\geq} \zeta/4,$$

where (a) due to assumption that norms are balanced; (b) we ignore $L(\mu_\lambda^*)$ and add and minus $\lambda \|\boldsymbol{a}_*\|_1$; (c) due to Lemma F.3; (d) due to assumption that norms are balanced and the choice of $q_{ij}$.

In the following, we will lower bound the last term of (12) to show it is no smaller than $-\zeta/8$ so that we get the desired lower bound. Recall the residual decomposition (8) that $R(\boldsymbol{x}) = R_1(\boldsymbol{x}) + R_2(\boldsymbol{x}) + R_3(\boldsymbol{x})$, we have

$$\sum_{i\in[m_*]} \sum_{j\in\mathcal{T}_i} \mathbb{E}_{\boldsymbol{x}}[R(\boldsymbol{x})a_j q_{ij} \boldsymbol{w}_i^{*\top}\boldsymbol{x}(\sigma'(\boldsymbol{w}_j^\top\boldsymbol{x}) - \sigma'(\boldsymbol{w}_i^{*\top}\boldsymbol{x}))]$$

$$= \underbrace{\sum_{i\in[m_*]} \sum_{j\in\mathcal{T}_i} \mathbb{E}_{\boldsymbol{x}}[R_1(\boldsymbol{x})a_j q_{ij} \boldsymbol{w}_i^{*\top}\boldsymbol{x}(\sigma'(\boldsymbol{w}_i^{*\top}\boldsymbol{x}) - \sigma'(\boldsymbol{w}_j^\top\boldsymbol{x}))]}_{(I)}$$

$$+ \underbrace{\sum_{i\in[m_*]} \sum_{j\in\mathcal{T}_i} \mathbb{E}_{\boldsymbol{x}}[R_2(\boldsymbol{x})a_j q_{ij} \boldsymbol{w}_i^{*\top}\boldsymbol{x}(\sigma'(\boldsymbol{w}_i^{*\top}\boldsymbol{x}) - \sigma'(\boldsymbol{w}_j^\top\boldsymbol{x}))]}_{(II)}$$

$$+ \underbrace{\sum_{i\in[m_*]} \sum_{j\in\mathcal{T}_i} \mathbb{E}_{\boldsymbol{x}}[R_3(\boldsymbol{x})a_j q_{ij} \boldsymbol{w}_i^{*\top}\boldsymbol{x}(\sigma'(\boldsymbol{w}_i^{*\top}\boldsymbol{x}) - \sigma'(\boldsymbol{w}_j^\top\boldsymbol{x}))]}_{(III)}$$

**Bound (I)** For (I), recall $R_1(\boldsymbol{x}) = (1/2)\sum_{i\in[m_*]} \boldsymbol{v}_i^\top\boldsymbol{x}\,\mathrm{sign}(\boldsymbol{w}_i^{*\top}\boldsymbol{x})$, where $\boldsymbol{v}_i = \sum_{j\in\mathcal{T}_i} a_j\boldsymbol{w}_j - \boldsymbol{w}_i^*$ is the difference between average neuron and corresponding ground-truth and $(\sum_{i\in[m_*]} \|\boldsymbol{v}_i\|_2^2)^{1/2} = O_*((\zeta/\lambda)^{3/4})$ from Lemma F.10 and Lemma F.11. We have

$$\sum_{i\in[m_*]} \sum_{j\in\mathcal{T}_i} \mathbb{E}_{\boldsymbol{x}}[R_1(\boldsymbol{x})a_j q_{ij} \boldsymbol{w}_i^{*\top}\boldsymbol{x}(\sigma'(\boldsymbol{w}_i^{*\top}\boldsymbol{x}) - \sigma'(\boldsymbol{w}_j^\top\boldsymbol{x}))]$$

$$\overset{(a)}{\geq} -\frac{1}{2} \sum_{i\in[m_*]} \sum_{j\in\mathcal{T}_i} \sum_{k\in[m_*]} \mathbb{E}_{\boldsymbol{x}}[|\boldsymbol{v}_k^\top\boldsymbol{x}||a_j q_{ij}||\boldsymbol{w}_i^{*\top}\boldsymbol{x}|\mathbb{1}_{\mathrm{sign}(\boldsymbol{w}_j^\top\boldsymbol{x})\neq\mathrm{sign}(\boldsymbol{w}_i^{*\top}\boldsymbol{x})}]$$

$$\overset{(b)}{=} -\frac{1}{2} \sum_{i\in[m_*]} \sum_{j\in\mathcal{T}_i} \sum_{k\in[m_*]} |a_j q_{ij}| \|\boldsymbol{v}_k\|_2 \mathbb{E}_{\widetilde{\boldsymbol{x}}}[|\overline{\boldsymbol{v}}_k^\top\widetilde{\boldsymbol{x}}||\boldsymbol{w}_i^{*\top}\widetilde{\boldsymbol{x}}|\mathbb{1}_{\mathrm{sign}(\boldsymbol{w}_j^\top\widetilde{\boldsymbol{x}})\neq\mathrm{sign}(\boldsymbol{w}_i^{*\top}\widetilde{\boldsymbol{x}})}]$$

$$\overset{(c)}{\geq} -\frac{1}{2} \sum_{i\in[m_*]} \sum_{j\in\mathcal{T}_i} \sum_{k\in[m_*]} |a_j q_{ij}| \|\boldsymbol{v}_k\|_2 \delta_j \mathbb{E}_{\widetilde{\boldsymbol{x}}}[\|\widetilde{\boldsymbol{x}}\|_2^2 \mathbb{1}_{\mathrm{sign}(\boldsymbol{w}_j^\top\widetilde{\boldsymbol{x}})\neq\mathrm{sign}(\boldsymbol{w}_i^{*\top}\widetilde{\boldsymbol{x}})}]$$

$$\overset{(d)}{\geq} -\frac{1}{2} \sum_{i\in[m_*]} \sum_{j\in\mathcal{T}_i} \sum_{k\in[m_*]} |a_j q_{ij}| \|\boldsymbol{v}_k\|_2 \Theta(\delta_j^2)$$

$$\overset{(e)}{\geq} -\Theta_*((\zeta/\lambda)^{3/4}\delta_{close}^2) \sum_{i\in[m_*]} \sum_{j\in\mathcal{T}_i} |a_j q_{ij}| = -\Theta_*((\zeta/\lambda)^{3/4}\delta_{close}^2),$$

where in (a) we plug in the definition of $R_1$ and using the fact that $\boldsymbol{w}_i^{*\top}\boldsymbol{x}(\sigma'(\boldsymbol{w}_i^{*\top}\boldsymbol{x}) - \sigma'(\boldsymbol{w}_j^\top\boldsymbol{x})) = |\boldsymbol{w}_i^{*\top}\boldsymbol{x}|\mathbb{1}_{\text{sign}(\boldsymbol{w}_j^\top\boldsymbol{x})\neq\text{sign}(\boldsymbol{w}_i^{*\top}\boldsymbol{x})}$; (b) $\widetilde{\boldsymbol{x}}$ is a 3-dimensional Gaussian since the expectation only depends on $\boldsymbol{v}_k, \boldsymbol{w}_i^*, \boldsymbol{w}_j$; (c) $|\boldsymbol{w}_i^{*\top}\widetilde{\boldsymbol{x}}| \leq \delta_j\,\|\widetilde{\boldsymbol{x}}\|_2$ when $\text{sign}(\boldsymbol{w}_j^\top\widetilde{\boldsymbol{x}}) \neq \text{sign}(\boldsymbol{w}_i^{*\top}\widetilde{\boldsymbol{x}})$; (d) a direct calculation bound as Lemma H.2; (e) definition of $q_{ij}$.

**Bound (II)** For (II), recall

$$R_2(\boldsymbol{x}) = \frac{1}{2}\sum_{i\in[m_*]}\sum_{j\in\mathcal{T}_i} a_j\boldsymbol{w}_j^\top\boldsymbol{x}(\text{sign}(\boldsymbol{w}_j^\top\boldsymbol{x}) - \text{sign}(\boldsymbol{w}_i^{*\top}\boldsymbol{x})) = \sum_{i\in[m_*]}\sum_{j\in\mathcal{T}_i} a_j|\boldsymbol{w}_j^\top\boldsymbol{x}|\mathbb{1}_{\text{sign}(\boldsymbol{w}_j^\top\boldsymbol{x})\neq\text{sign}(\boldsymbol{w}_i^{*\top}\boldsymbol{x}))}.$$

For each term in (II) with $j \in \mathcal{T}_i$, we can split it into two terms that corresponding to $\mathcal{T}_i$ and other $\mathcal{T}_k$'s.

$$\mathbb{E}_{\boldsymbol{x}}[R_2(\boldsymbol{x})a_jq_{ij}\boldsymbol{w}_i^{*\top}\boldsymbol{x}(\sigma'(\boldsymbol{w}_i^{*\top}\boldsymbol{x}) - \sigma'(\boldsymbol{w}_j^\top\boldsymbol{x}))]$$

$$= \sum_{k\in[m_*]}\sum_{\ell\in\mathcal{T}_k}\mathbb{E}_{\boldsymbol{x}}[a_\ell|\boldsymbol{w}_\ell^\top\boldsymbol{x}|\mathbb{1}_{\text{sign}(\boldsymbol{w}_\ell^\top\boldsymbol{x})\neq\text{sign}(\boldsymbol{w}_k^{*\top}\boldsymbol{x})} \cdot a_jq_{ij}\boldsymbol{w}_i^{*\top}\boldsymbol{x}(\sigma'(\boldsymbol{w}_i^{*\top}\boldsymbol{x}) - \sigma'(\boldsymbol{w}_j^\top\boldsymbol{x}))]$$

$$= \sum_{k\in[m_*]}\sum_{\ell\in\mathcal{T}_k}a_\ell a_jq_{ij}\mathbb{E}_{\boldsymbol{x}}[|\boldsymbol{w}_\ell^\top\boldsymbol{x}|\mathbb{1}_{\text{sign}(\boldsymbol{w}_\ell^\top\boldsymbol{x})\neq\text{sign}(\boldsymbol{w}_k^{*\top}\boldsymbol{x})} \cdot |\boldsymbol{w}_i^{*\top}\boldsymbol{x}|\mathbb{1}_{\text{sign}(\boldsymbol{w}_i^{*\top}\boldsymbol{x})\neq\text{sign}(\boldsymbol{w}_j^\top\boldsymbol{x})}]$$

$$= \underbrace{\sum_{\ell\in\mathcal{T}_i}a_\ell a_jq_{ij}\mathbb{E}_{\boldsymbol{x}}[|\boldsymbol{w}_\ell^\top\boldsymbol{x}||\boldsymbol{w}_i^{*\top}\boldsymbol{x}|\mathbb{1}_{\text{sign}(\boldsymbol{w}_\ell^\top\boldsymbol{x})\neq\text{sign}(\boldsymbol{w}_i^{*\top}\boldsymbol{x})} \cdot \mathbb{1}_{\text{sign}(\boldsymbol{w}_i^{*\top}\boldsymbol{x})\neq\text{sign}(\boldsymbol{w}_j^\top\boldsymbol{x})}]}_{(II.i)}$$

$$+ \underbrace{\sum_{k\neq i}\sum_{\ell\in\mathcal{T}_k}a_\ell a_jq_{ij}\mathbb{E}_{\boldsymbol{x}}[|\boldsymbol{w}_\ell^\top\boldsymbol{x}||\boldsymbol{w}_i^{*\top}\boldsymbol{x}|\mathbb{1}_{\text{sign}(\boldsymbol{w}_\ell^\top\boldsymbol{x})\neq\text{sign}(\boldsymbol{w}_k^{*\top}\boldsymbol{x})} \cdot \mathbb{1}_{\text{sign}(\boldsymbol{w}_i^{*\top}\boldsymbol{x})\neq\text{sign}(\boldsymbol{w}_j^\top\boldsymbol{x})}]}_{(II.ii)}. \quad (13)$$

For (II.i), we further split neurons into $\mathcal{T}_i(\delta_{\text{sign}})$ and others:

$$(II.i) = \sum_{\ell\in\mathcal{T}_i(\delta_{\text{sign}})}a_\ell a_jq_{ij}\mathbb{E}_{\boldsymbol{x}}[|\boldsymbol{w}_\ell^\top\boldsymbol{x}||\boldsymbol{w}_i^{*\top}\boldsymbol{x}|\mathbb{1}_{\text{sign}(\boldsymbol{w}_\ell^\top\boldsymbol{x})\neq\text{sign}(\boldsymbol{w}_i^{*\top}\boldsymbol{x})} \cdot \mathbb{1}_{\text{sign}(\boldsymbol{w}_i^{*\top}\boldsymbol{x})\neq\text{sign}(\boldsymbol{w}_j^\top\boldsymbol{x})}]$$

$$+ \sum_{\ell\in\mathcal{T}_i\backslash\mathcal{T}_i(\delta_{\text{sign}})}a_\ell a_jq_{ij}\mathbb{E}_{\boldsymbol{x}}[|\boldsymbol{w}_\ell^\top\boldsymbol{x}||\boldsymbol{w}_i^{*\top}\boldsymbol{x}|\mathbb{1}_{\text{sign}(\boldsymbol{w}_\ell^\top\boldsymbol{x})\neq\text{sign}(\boldsymbol{w}_i^{*\top}\boldsymbol{x})} \cdot \mathbb{1}_{\text{sign}(\boldsymbol{w}_i^{*\top}\boldsymbol{x})\neq\text{sign}(\boldsymbol{w}_j^\top\boldsymbol{x})}]$$

$$(14)$$

Consider the first line of (14), from the choice of $q_{ij}$ we know $a_jq_{ij}a_i^* \geq 0$. For $\ell \in \mathcal{T}_{i,+}(\delta_{\text{sign}})$, we know $\text{sign}(a_\ell) = \text{sign}(a_i^*)$, which implies $a_\ell a_jq_{ij} \geq 0$ for these terms. We thus only need to deal with neurons in $T_{i,-}(\delta_{\text{sign}})$, we have the first line is bounded as

$$\sum_{\ell\in\mathcal{T}_i(\delta_{\text{sign}})}a_\ell a_jq_{ij}\mathbb{E}_{\boldsymbol{x}}[|\boldsymbol{w}_\ell^\top\boldsymbol{x}||\boldsymbol{w}_i^{*\top}\boldsymbol{x}|\mathbb{1}_{\text{sign}(\boldsymbol{w}_\ell^\top\boldsymbol{x})\neq\text{sign}(\boldsymbol{w}_i^{*\top}\boldsymbol{x})} \cdot \mathbb{1}_{\text{sign}(\boldsymbol{w}_i^{*\top}\boldsymbol{x})\neq\text{sign}(\boldsymbol{w}_j^\top\boldsymbol{x})}]$$

$$\geq \sum_{\ell\in\mathcal{T}_{i,-}(\delta_{\text{sign}})}a_\ell a_jq_{ij}\mathbb{E}_{\boldsymbol{x}}[|\boldsymbol{w}_\ell^\top\boldsymbol{x}||\boldsymbol{w}_i^{*\top}\boldsymbol{x}|\mathbb{1}_{\text{sign}(\boldsymbol{w}_\ell^\top\boldsymbol{x})\neq\text{sign}(\boldsymbol{w}_i^{*\top}\boldsymbol{x})} \cdot \mathbb{1}_{\text{sign}(\boldsymbol{w}_i^{*\top}\boldsymbol{x})\neq\text{sign}(\boldsymbol{w}_j^\top\boldsymbol{x})}]$$

$$\overset{(a)}{\geq} -|a_jq_{ij}|\sum_{\ell\in\mathcal{T}_{i,-}(\delta_{\text{sign}})}|a_\ell|\,\|\boldsymbol{w}_\ell\|_2\,\mathbb{E}_{\boldsymbol{x}}[|\overline{\boldsymbol{w}}_\ell^\top\widetilde{\boldsymbol{x}}||\boldsymbol{w}_i^{*\top}\widetilde{\boldsymbol{x}}|\mathbb{1}_{\text{sign}(\boldsymbol{w}_\ell^\top\widetilde{\boldsymbol{x}})\neq\text{sign}(\boldsymbol{w}_i^{*\top}\widetilde{\boldsymbol{x}})} \cdot \mathbb{1}_{\text{sign}(\boldsymbol{w}_i^{*\top}\widetilde{\boldsymbol{x}})\neq\text{sign}(\boldsymbol{w}_j^\top\widetilde{\boldsymbol{x}})}]$$

$$\overset{(b)}{\geq} -|a_jq_{ij}|\sum_{\ell\in\mathcal{T}_{i,-}(\delta_{\text{sign}})}|a_\ell|\,\|\boldsymbol{w}_\ell\|_2\,\delta_\ell\delta_j\mathbb{E}_{\boldsymbol{x}}[\|\widetilde{\boldsymbol{x}}\|_2^2\,\mathbb{1}_{\text{sign}(\boldsymbol{w}_\ell^\top\widetilde{\boldsymbol{x}})\neq\text{sign}(\boldsymbol{w}_i^{*\top}\widetilde{\boldsymbol{x}})} \cdot \mathbb{1}_{\text{sign}(\boldsymbol{w}_i^{*\top}\widetilde{\boldsymbol{x}})\neq\text{sign}(\boldsymbol{w}_j^\top\widetilde{\boldsymbol{x}})}]$$

$$\overset{(c)}{\geq} -|a_jq_{ij}|\sum_{\ell\in\mathcal{T}_{i,-}(\delta_{\text{sign}})}|a_\ell|\,\|\boldsymbol{w}_\ell\|_2\,O(\delta_\ell\delta_j^2)$$

$$\overset{(d)}{\geq} -|a_jq_{ij}|O(\tau\delta_{\text{sign}}\delta_{close}^2),$$

where (a) $\widetilde{\boldsymbol{x}}$ is a 3-dimensional Gaussian since the expectation only depends on $\boldsymbol{w}_\ell, \boldsymbol{w}_j, \boldsymbol{w}_i^*$; (b) $|\overline{\boldsymbol{w}}_\ell^\top \widetilde{\boldsymbol{x}}| \leq \delta_\ell \|\widetilde{\boldsymbol{x}}\|_2$ when $\text{sign}(\boldsymbol{w}_i^{*\top}\widetilde{\boldsymbol{x}}) \neq \text{sign}(\boldsymbol{w}_\ell^\top \widetilde{\boldsymbol{x}})$ and $|\boldsymbol{w}_i^{*\top}\widetilde{\boldsymbol{x}}| \leq \delta_j \|\widetilde{\boldsymbol{x}}\|_2$ when $\text{sign}(\boldsymbol{w}_i^{*\top}\widetilde{\boldsymbol{x}}) \neq \text{sign}(\boldsymbol{w}_j^\top \widetilde{\boldsymbol{x}})$; (c) a direct calculation as in Lemma H.2; (d) assumption that norm cancellation is small.

For the second term of (14), similar as above, we have

$$2 \sum_{\ell \in \mathcal{T}_i \setminus \mathcal{T}_i(\delta_{\text{sign}})} a_\ell a_j q_{ij} \mathbb{E}_{\boldsymbol{x}}[|\boldsymbol{w}_\ell^\top \boldsymbol{x}||\boldsymbol{w}_i^{*\top}\boldsymbol{x}| \mathbb{1}_{\text{sign}(\boldsymbol{w}_\ell^\top \boldsymbol{x}) \neq \text{sign}(\boldsymbol{w}_i^{*\top}\boldsymbol{x})} \cdot \mathbb{1}_{\text{sign}(\boldsymbol{w}_i^{*\top}\boldsymbol{x}) \neq \text{sign}(\boldsymbol{w}_j^\top \boldsymbol{x})}]$$

$$\overset{(a)}{\geq} -2|a_j q_{ij}| \sum_{\ell \in \mathcal{T}_i \setminus \mathcal{T}_i(\delta_{\text{sign}})} |a_\ell| \|\boldsymbol{w}_\ell\|_2 \mathbb{E}_{\widetilde{\boldsymbol{x}}}[|\overline{\boldsymbol{w}}_\ell^\top \widetilde{\boldsymbol{x}}||\boldsymbol{w}_i^{*\top}\widetilde{\boldsymbol{x}}| \mathbb{1}_{\text{sign}(\boldsymbol{w}_\ell^\top \widetilde{\boldsymbol{x}}) \neq \text{sign}(\boldsymbol{w}_i^{*\top}\widetilde{\boldsymbol{x}})} \cdot \mathbb{1}_{\text{sign}(\boldsymbol{w}_i^{*\top}\widetilde{\boldsymbol{x}}) \neq \text{sign}(\boldsymbol{w}_j^\top \widetilde{\boldsymbol{x}})}]$$

$$\overset{(b)}{\geq} -2|a_j q_{ij}| \sum_{\ell \in \mathcal{T}_i \setminus \mathcal{T}_i(\delta_{\text{sign}})} |a_\ell| \|\boldsymbol{w}_\ell\|_2 \delta_\ell \delta_j \mathbb{E}_{\widetilde{\boldsymbol{x}}}[\|\widetilde{\boldsymbol{x}}\|_2^2 \mathbb{1}_{\text{sign}(\boldsymbol{w}_i^{*\top}\widetilde{\boldsymbol{x}}) \neq \text{sign}(\boldsymbol{w}_j^\top \widetilde{\boldsymbol{x}})}]$$

$$\overset{(c)}{\geq} -2|a_j q_{ij}|O(\delta_j^2) \sum_{\ell \in \mathcal{T}_i \setminus \mathcal{T}_i(\delta_{\text{sign}})} |a_\ell| \|\boldsymbol{w}_\ell\|_2 \delta_\ell$$

$$\overset{(d)}{\geq} -2|a_j q_{ij}|O_*(\delta_{close}^2 \zeta \lambda^{-1} \delta_{\text{sign}}^{-1}),$$

where (a) $\widetilde{\boldsymbol{x}}$ is 3-dimensional Gaussian vector since the expectation only depends on $\boldsymbol{w}_\ell, \boldsymbol{w}_j, \boldsymbol{w}_i^*$; (b) $|\overline{\boldsymbol{w}}_\ell^\top \widetilde{\boldsymbol{x}}| \leq \delta_\ell \|\widetilde{\boldsymbol{x}}\|_2$ when $\text{sign}(\boldsymbol{w}_i^{*\top}\widetilde{\boldsymbol{x}}) \neq \text{sign}(\boldsymbol{w}_\ell^\top \widetilde{\boldsymbol{x}})$ and $|\boldsymbol{w}_i^{*\top}\widetilde{\boldsymbol{x}}| \leq \delta_j \|\widetilde{\boldsymbol{x}}\|_2$ when $\text{sign}(\boldsymbol{w}_i^{*\top}\widetilde{\boldsymbol{x}}) \neq \text{sign}(\boldsymbol{w}_j^\top \widetilde{\boldsymbol{x}})$; (c) a direct calculation as in Lemma H.2; (d) choice of $q_{ij}$ and Lemma F.5 and Lemma F.11 that far-away neurons are small.

Thus, for (II.i) we have

$$(II.i) \geq -2|a_j q_{ij}|O_*(\tau \delta_{\text{sign}} \delta_{close}^2 + \delta_{close}^2 \zeta \lambda^{-1} \delta_{\text{sign}}^{-1}).$$

For (II.ii), we have

$$|(II.ii)| \leq 2 \sum_{k \neq i} \sum_{\ell \in \mathcal{T}_k} |a_\ell||a_j q_{ij}| \mathbb{E}_{\boldsymbol{x}}[|\boldsymbol{w}_\ell^\top \boldsymbol{x}||\boldsymbol{w}_i^{*\top}\boldsymbol{x}| \mathbb{1}_{\text{sign}(\boldsymbol{w}_\ell^\top \boldsymbol{x}) \neq \text{sign}(\boldsymbol{w}_k^{*\top}\boldsymbol{x})} \cdot \mathbb{1}_{\text{sign}(\boldsymbol{w}_i^{*\top}\boldsymbol{x}) \neq \text{sign}(\boldsymbol{w}_j^\top \boldsymbol{x})}]$$

$$\overset{(a)}{\leq} 2 \sum_{k \neq i} \sum_{\ell \in \mathcal{T}_k} |a_\ell||a_j q_{ij}| \|\boldsymbol{w}_\ell\|_2 \delta_\ell \delta_j \mathbb{E}_{\widetilde{\boldsymbol{x}}}[\|\widetilde{\boldsymbol{x}}\|_2^2 \mathbb{1}_{\text{sign}(\boldsymbol{w}_\ell^\top \widetilde{\boldsymbol{x}}) \neq \text{sign}(\boldsymbol{w}_k^{*\top}\widetilde{\boldsymbol{x}})} \cdot \mathbb{1}_{\text{sign}(\boldsymbol{w}_i^{*\top}\widetilde{\boldsymbol{x}}) \neq \text{sign}(\boldsymbol{w}_j^\top \widetilde{\boldsymbol{x}})}]$$

$$\overset{(b)}{\leq} 2 \sum_{k \neq i} \sum_{\ell \in \mathcal{T}_k} |a_\ell||a_j q_{ij}| \|\boldsymbol{w}_\ell\|_2 \delta_\ell \delta_j \mathbb{E}_{\widetilde{\boldsymbol{x}}}[\|\widetilde{\boldsymbol{x}}\|_2^2 \mathbb{1}_{|\boldsymbol{w}_k^{*\top}\widetilde{\boldsymbol{x}}| \leq \delta_\ell \|\widetilde{\boldsymbol{x}}\|_2} \cdot \mathbb{1}_{|\boldsymbol{w}_i^{*\top}\widetilde{\boldsymbol{x}}| \leq \delta_j \|\widetilde{\boldsymbol{x}}\|_2}]$$

$$\overset{(c)}{\leq} 2|a_j q_{ij}|\delta_j \sum_{k \neq i} \sum_{\ell \in \mathcal{T}_k} |a_\ell| \|\boldsymbol{w}_\ell\|_2 \delta_\ell \cdot O(\delta_\ell \delta_j / \Delta)$$

$$\overset{(d)}{=} 2|a_j q_{ij}|O_*(\delta_{close}^2 \zeta \lambda^{-1} \Delta^{-1}),$$

where (a)(b) $\widetilde{\boldsymbol{x}}$ is a 4-dimensional Gaussian vector, $|\overline{\boldsymbol{w}}_\ell^\top \widetilde{\boldsymbol{x}}| \leq \delta_\ell \|\widetilde{\boldsymbol{x}}\|_2$ when $\text{sign}(\boldsymbol{w}_i^{*\top}\widetilde{\boldsymbol{x}}) \neq \text{sign}(\boldsymbol{w}_\ell^\top \widetilde{\boldsymbol{x}})$ and $|\boldsymbol{w}_i^{*\top}\widetilde{\boldsymbol{x}}| \leq \delta_j \|\widetilde{\boldsymbol{x}}\|_2$ when $\text{sign}(\boldsymbol{w}_i^{*\top}\widetilde{\boldsymbol{x}}) \neq \text{sign}(\boldsymbol{w}_j^\top \widetilde{\boldsymbol{x}})$; (c) by Lemma H.1; (d) choice of $q_{ij}$ and Lemma F.5 and Lemma F.11 that far-away neurons are small.

Combine (II.i) (II.ii), we have for (13)

$$\mathbb{E}_{\boldsymbol{x}}[R_2(\boldsymbol{x})a_j q_{ij} \boldsymbol{w}_i^{*\top}\boldsymbol{x}(\sigma'(\boldsymbol{w}_i^{*\top}\boldsymbol{x}) - \sigma'(\boldsymbol{w}_j^\top \boldsymbol{x}))] \geq -2|a_j q_{ij}|O(\tau \delta_{\text{sign}} \delta_{close}^2 + \delta_{close}^2 \zeta \lambda^{-1} \delta_{\text{sign}}^{-1}).$$

This further gives the lower bound on (II):

$$\sum_{i \in [m_*]} \sum_{j \in \mathcal{T}_i} \mathbb{E}_{\boldsymbol{x}}[R_2(\boldsymbol{x})a_j q_{ij} \boldsymbol{w}_i^{*\top}\boldsymbol{x}(\sigma'(\boldsymbol{w}_i^{*\top}\boldsymbol{x}) - \sigma'(\boldsymbol{w}_j^\top \boldsymbol{x}))]$$

$$\geq -2 \sum_{i \in [m_*]} \sum_{j \in \mathcal{T}_i} |a_j q_{ij}|O(\tau \delta_{\text{sign}} \delta_{close}^2 + \delta_{close}^2 \zeta \lambda^{-1} \delta_{\text{sign}}^{-1})$$

$$= -O_*(\tau \delta_{\text{sign}} \delta_{close}^2 + \delta_{close}^2 \zeta \lambda^{-1} \delta_{\text{sign}}^{-1})$$

**Bound (III)** For (III), recall $R_3(\boldsymbol{x}) = \frac{1}{\sqrt{2\pi}}\left(\sum_{i\in[m_*]} a_i^* \|\boldsymbol{w}_i^*\|_2 - \sum_{i\in[m]} a_i \|\boldsymbol{w}_i\|_2\right) + \alpha - \hat{\alpha} + (\boldsymbol{\beta} - \hat{\boldsymbol{\beta}})^\top \boldsymbol{x}$. We have

$$\sum_{i\in[m_*]} \sum_{j\in\mathcal{T}_i} \mathbb{E}_{\boldsymbol{x}}[R_3(\boldsymbol{x}) a_j q_{ij} \boldsymbol{w}_i^{*\top}\boldsymbol{x}(\sigma'(\boldsymbol{w}_i^{*\top}\boldsymbol{x}) - \sigma'(\boldsymbol{w}_j^\top\boldsymbol{x}))]$$

$$\overset{(a)}{\geq} -O_*(\zeta/\lambda) \sum_{i\in[m_*]} \sum_{j\in\mathcal{T}_i} |a_j q_{ij}| \mathbb{E}_{\boldsymbol{x}}[|\boldsymbol{w}_i^{*\top}\boldsymbol{x}| \mathbb{1}_{\mathrm{sign}(\boldsymbol{w}_j^\top\boldsymbol{x})\neq\mathrm{sign}(\boldsymbol{w}_i^{*\top}\boldsymbol{x})}]$$

$$- \sum_{i\in[m_*]} \sum_{j\in\mathcal{T}_i} |a_j q_{ij}| \mathbb{E}_{\boldsymbol{x}}[|(\boldsymbol{\beta} - \hat{\boldsymbol{\beta}})^\top \boldsymbol{x}||\boldsymbol{w}_i^{*\top}\boldsymbol{x}| \mathbb{1}_{\mathrm{sign}(\boldsymbol{w}_j^\top\boldsymbol{x})\neq\mathrm{sign}(\boldsymbol{w}_i^{*\top}\boldsymbol{x})}]$$

$$\overset{(b)}{\geq} -O_*(\zeta/\lambda) \sum_{i\in[m_*]} \sum_{j\in\mathcal{T}_i} |a_j q_{ij}| O(\delta_j^2)$$

$$- O(\zeta^{1/2}) \sum_{i\in[m_*]} \sum_{j\in\mathcal{T}_i} |a_j q_{ij}| \delta_j \mathbb{E}_{\boldsymbol{x}}[\|\widetilde{\boldsymbol{x}}\|_2^2 \mathbb{1}_{\mathrm{sign}(\boldsymbol{w}_j^\top\widetilde{\boldsymbol{x}})\neq\mathrm{sign}(\boldsymbol{w}_i^{*\top}\widetilde{\boldsymbol{x}})}]$$

$$\overset{(c)}{\geq} -O_*(\zeta/\lambda) \sum_{i\in[m_*]} \sum_{j\in\mathcal{T}_i} |a_j q_{ij}| O(\delta_j^2)$$

$$\overset{(d)}{\geq} -O_*(\delta_{close}^2 \zeta/\lambda),$$

where (a) plugging in the expression of $R_3$ and using Lemma F.9 and Lemma F.11; (b) using Lemma H.3 and the fact that $\widetilde{\boldsymbol{x}}$ is a 3-dimensional Gaussian vector and $|\boldsymbol{w}_i^{*\top}\widetilde{\boldsymbol{x}}| \leq \delta_j \|\widetilde{\boldsymbol{x}}\|_2$ when $\mathrm{sign}(\boldsymbol{w}_i^{*\top}\widetilde{\boldsymbol{x}}) \neq \mathrm{sign}(\boldsymbol{w}_j^\top\widetilde{\boldsymbol{x}})$; (c) Lemma H.2; (d) choice of $q_{ij}$.

**Combine all bounds** Combine (I) (II) (III) we now get the last term of (12)

$$\sum_{i\in[m_*]} \sum_{j\in\mathcal{T}_i} \mathbb{E}_{\boldsymbol{x}}[R(\boldsymbol{x}) a_j q_{ij} \boldsymbol{w}_i^{*\top}\boldsymbol{x}(\sigma'(\boldsymbol{w}_j^\top\boldsymbol{x}) - \sigma'(\boldsymbol{w}_i^{*\top}\boldsymbol{x}))] \geq -O_*((\zeta/\lambda)^{3/4}\delta_{close}^2 + \tau\delta_{\mathrm{sign}}\delta_{close}^2 + \delta_{close}^2 \zeta\lambda^{-1}\delta_{\mathrm{sign}}^{-1})$$

From Lemma F.6 we can choose $\delta_{close} = O_*(\zeta^{1/3})$ and from Lemma F.16 we can choose $\delta_{\mathrm{sign}} = \Theta_*(\lambda/\zeta^{1/2})$. Also with $\tau = O(\zeta^{5/6}/\lambda)$, we finally get

$$\sum_{i\in[m_*]} \sum_{j\in\mathcal{T}_i} \mathbb{E}_{\boldsymbol{x}}[R(\boldsymbol{x}) a_j q_{ij} \boldsymbol{w}_i^{*\top}\boldsymbol{x}(\sigma'(\boldsymbol{w}_j^\top\boldsymbol{x}) - \sigma'(\boldsymbol{w}_i^{*\top}\boldsymbol{x}))] \geq \zeta/8,$$

as long as $\zeta = O(\lambda^{9/5}/\mathrm{poly}(r, m_*, \Delta, \|\boldsymbol{a}_*\|_1, a_{\min}))$ with small enough hidden constant.

Thus, we eventually get the lower bound of (12)

$$(\alpha + \alpha_*)\nabla_\alpha L_\lambda + \langle\nabla_{\boldsymbol{\beta}} L_\lambda, \boldsymbol{\beta} + \boldsymbol{\beta}_*\rangle + \sum_{i\in[m_*]} \sum_{j\in\mathcal{T}_i} \langle\nabla_{\boldsymbol{w}_i} L_\lambda, \boldsymbol{w}_j - q_{ij}\boldsymbol{w}_i^*\rangle \geq \zeta/4 - \zeta/8 = \zeta/8.$$

$\square$

## H.4 Technical Lemma

In this section, we collect several technical lemmas that are useful in the proof.

**Lemma H.1.** *Consider $\boldsymbol{\alpha}, \boldsymbol{\beta} \in \mathbb{R}^4$ with $\phi = \angle(\boldsymbol{\alpha}, \boldsymbol{\beta}) \in [0, \pi]$ and $\|\boldsymbol{\alpha}\|_2 = \|\boldsymbol{\beta}\|_2 = 1$ and $\boldsymbol{x} \sim N(\boldsymbol{0}, \boldsymbol{I})$. Then, for any $0 < \delta_1, \delta_2 \leq \phi$ we have*

$$\mathbb{E}_{\boldsymbol{x}}[\|\boldsymbol{x}\|_2^2 \mathbb{1}_{|\boldsymbol{\alpha}^\top\boldsymbol{x}|\leq\delta_1\|\boldsymbol{x}\|_2, |\boldsymbol{\beta}^\top\boldsymbol{x}|\leq\delta_2\|\boldsymbol{x}\|_2}] = O(\delta_1\delta_2/\sin\phi).$$

*Proof.* We first consider the case when at least one of $\delta_1, \delta_2 \geq c\phi$ for a fixed small enough constant. WLOG, suppose $\delta_2 \geq c\phi$. In this case, it suffices to show a bound $O(\delta_1)$. We have

$$\mathbb{E}_{\boldsymbol{x}}[\|\boldsymbol{x}\|_2^2 \mathbb{1}_{|\boldsymbol{\alpha}^\top\boldsymbol{x}|\leq\delta_1\|\boldsymbol{x}\|_2, |\boldsymbol{\beta}^\top\boldsymbol{x}|\leq\delta_2\|\boldsymbol{x}\|_2}] \leq \mathbb{E}_{\boldsymbol{x}}[\|\boldsymbol{x}\|_2^2 \mathbb{1}_{|\boldsymbol{\alpha}^\top\boldsymbol{x}|\leq\delta_1\|\boldsymbol{x}\|_2}] = O(\delta_1).$$

Then, we focus on the case when $\delta_1, \delta_2 \leq c\phi$ for a fixed small enough constant. WLOG, assume $\boldsymbol{\alpha} = (1,0,0,0)^\top$, $\boldsymbol{\beta} = (\cos\phi, \sin\phi, 0, 0)$ and $\phi \in [0, \pi/2]$. Then we have

$$
\begin{aligned}
&\mathbb{E}_{\boldsymbol{x}}[\|\boldsymbol{x}\|_2^2 \, \mathbb{1}_{|\boldsymbol{\alpha}^\top \boldsymbol{x}| \leq \delta_1 \|\boldsymbol{x}\|_2, |\boldsymbol{\beta}^\top \boldsymbol{x}| \leq \delta_2 \|\boldsymbol{x}\|_2}]\\
&=\frac{1}{(2\pi)^2} \int_0^\infty r^5 e^{-r^2/2} \, \mathrm{d}r\\
&\quad \int_{0\leq\theta_1\leq\pi, |\cos\theta_1|\leq\delta_1} \sin^2\theta_1 \int_{0\leq\theta_2\leq\pi, |\cos\theta_1\cos\phi+\sin\theta_1\cos\theta_2\sin\phi|\leq\delta_2} \sin\theta_2 \, \mathrm{d}\theta_2 \, \mathrm{d}\theta_1 \int_0^{2\pi} 1 \, \mathrm{d}\theta_3\\
&=O(1) \cdot \int_{0\leq\theta_1\leq\pi, |\cos\theta_1|\leq\delta_1} \sin^2\theta_1 \int_{0\leq\theta_2\leq\pi, \frac{-\delta_2-\cos\theta_1\cos\phi}{\sin\theta_1\sin\phi}\leq\cos\theta_2\leq\frac{\delta_2-\cos\theta_1\cos\phi}{\sin\theta_1\sin\phi}} \sin\theta_2 \, \mathrm{d}\theta_2 \, \mathrm{d}\theta_1\\
&=\int_{0\leq\theta_1\leq\pi, |\cos\theta_1|\leq\delta_1} \sin^2\theta_1 \cdot O\left(\frac{\delta_2}{\sin\theta_1\sin\phi}\right) \mathrm{d}\theta_1\\
&=O\left(\frac{\delta_1\delta_2}{\sin\phi}\right).
\end{aligned}
$$

$\square$

**Lemma H.2** (Lemma C.9 in Zhou et al. (2021)). *Consider* $\boldsymbol{\alpha}, \boldsymbol{\beta} \in \mathbb{R}^3$ *with* $\angle(\boldsymbol{\alpha}, \boldsymbol{\beta}) = \phi$ *and* $\boldsymbol{\alpha}^\top\boldsymbol{\beta} \geq 0$. *We have*

$$
\mathbb{E}_{\boldsymbol{x}}[\|\boldsymbol{x}\|^2 \, \mathbb{1}_{\mathrm{sign}(\boldsymbol{\alpha}^\top\boldsymbol{x})\neq\mathrm{sign}(\boldsymbol{\beta}^\top\boldsymbol{x})}] = O(\phi).
$$

**Lemma H.3.** *Consider* $\boldsymbol{\alpha}, \boldsymbol{\beta} \in \mathbb{R}^d$ *with* $\angle(\boldsymbol{\alpha}, \boldsymbol{\beta}) = \phi$, $\|\boldsymbol{\alpha}\|_2 = \|\boldsymbol{\beta}\|_2 = 1$ *and* $\boldsymbol{\alpha}^\top\boldsymbol{\beta} \geq 0$. *We have*

$$
\mathbb{E}_{\boldsymbol{x}}[|\boldsymbol{\alpha}^\top\boldsymbol{x}| \mathbb{1}_{\mathrm{sign}(\boldsymbol{\alpha}^\top\boldsymbol{x})\neq\mathrm{sign}(\boldsymbol{\beta}^\top\boldsymbol{x})}] = O(\phi^2).
$$

*Proof.* It suffices to consider $\boldsymbol{\alpha}, \boldsymbol{\beta}, \boldsymbol{x} \in \mathbb{R}^2$. WLOG, assume $\boldsymbol{\alpha} = (1,0)^\top$ and $\boldsymbol{\beta} = (\cos\phi, \sin\phi)^\top$ We have

$$
\begin{aligned}
\mathbb{E}_{\boldsymbol{x}}[|\boldsymbol{\alpha}^\top\boldsymbol{x}| \mathbb{1}_{\mathrm{sign}(\boldsymbol{\alpha}^\top\boldsymbol{x})\neq\mathrm{sign}(\boldsymbol{\beta}^\top\boldsymbol{x})}] &= \frac{1}{2\pi} \int_0^\infty r e^{-r^2/2} \, \mathrm{d}r \int_0^{2\pi} \cos\theta \, \mathbb{1}_{\mathrm{sign}(\cos\theta)\neq\mathrm{sign}(\cos(\theta-\phi))} \, \mathrm{d}\theta\\
&= O(\phi^2).
\end{aligned}
$$

$\square$

**Lemma H.4.** *Under Lemma 6, let*

$$
q_{ij} = \begin{cases} \frac{a_j a_i^*}{\sum_{j\in T_{i,+}(\delta_{close})} a_j^2} & , \text{if } j \in T_{i,+}(\delta_{close})\\ 0 & , \text{otherwise} \end{cases}
$$

*If* $\sum_{i\in[m_*]} \left|a_i^2 - \|\boldsymbol{w}_i\|_2^2\right| \leq a_{\min}/2$, *then* $\sum_{i\in[m_*]} \sum_{j\in\mathcal{T}_i} q_{ij}^2 = O(\|\boldsymbol{a}_*\|_1)$.

*Proof.* We have

$$
\sum_{i\in[m_*]} \sum_{j\in\mathcal{T}_i} q_{ij}^2 = \sum_{i\in[m_*]} \sum_{j\in\mathcal{T}_{i,+}(\delta_{close})} \frac{a_j^2 a_i^{*2}}{(\sum_{j\in T_{i,+}(\delta_{close})} a_j^2)^2} = \sum_{i\in[m_*]} \frac{a_i^{*2}}{\sum_{j\in T_{i,+}(\delta_{close})} a_j^2}.
$$

In the following, we aim to lower bound $\sum_{j\in T_{i,+}(\delta_{close})} a_j^2$. Given $\sum_{j\in T_{i,+}(\delta_{close})} |a_j^2 - \|\boldsymbol{w}_j\|_2^2| \leq |a_i^*|/2$, we have

$$
2 \sum_{j\in T_{i,+}(\delta_{close})} a_j^2 \geq \sum_{j\in T_{i,+}(\delta_{close})} a_j^2 + \|\boldsymbol{w}_j\|_2^2 - |a_i^*|/2 \geq 2 \sum_{j\in T_{i,+}(\delta_{close})} |a_j| \|\boldsymbol{w}_j\|_2 - |a_i^*|/2 \geq |a_i^*|/2,
$$

where the last inequality is due to Lemma F.6: $\sum_{j\in T_{i,+}(\delta_{close})} |a_j| \|\boldsymbol{w}_j\|_2 \geq |\sum_{j\in\mathcal{T}_i(\delta_{close})} a_j \|\boldsymbol{w}_j\|_2| \geq |a_i^*|/2$. Thus, we have $\sum_{i\in[m_*]} \sum_{j\in\mathcal{T}_i} q_{ij}^2 = O(\|\boldsymbol{a}_*\|_1)$. $\square$

# I  Proofs in Section G (non-degenerate dual certificate)

In this section, we give the omitted proofs in Section G. The proofs are mostly direct computations with the properties of Hermite polynomials in Claim A.1.

**Lemma G.1** (Non-degeneracy of kernel $K$). *For any $h > 0$, let $\ell \geq \Theta(\Delta^{-2}\log(m_*\ell/h\Delta))$, kernel $K_{\geq\ell}$ is non-degenerate in the sense that there exists $r = \Theta(\ell^{-1/2}), \rho_1 = \Theta(1), \rho_2 = \Theta(\ell)$ such that following hold:*

(i) $K(\boldsymbol{w}, \boldsymbol{u}) \leq 1 - \rho_1$ *for all* $\delta(\boldsymbol{w}, \boldsymbol{u}) := \angle(\boldsymbol{w}, \boldsymbol{u}) \geq r$.

(ii) $K^{(20)}(\boldsymbol{w}, \boldsymbol{u})[\boldsymbol{z}, \boldsymbol{z}] \leq -\rho_2 \|\boldsymbol{z}\|^2$ *for tangent vector $\boldsymbol{z}$ that $\boldsymbol{z}^\top\boldsymbol{w} = 0$ and $\delta(\boldsymbol{w}, \boldsymbol{u}) \leq r$.*

(iii) $\left\|K^{(ij)}(\boldsymbol{w}_1^*, \boldsymbol{w}_k^*)\right\|_{\boldsymbol{w}_i^*, \boldsymbol{w}_k^*} \leq h/m_*^2$ *for* $(i, j) \in \{0, 1\} \times \{0, 1, 2\}$

*Proof.* With the property of Hermite polynomials in Claim A.1, we have

$$K(\boldsymbol{w}, \boldsymbol{u}) = \mathbb{E}_{\boldsymbol{x}}[\overline{\sigma_{\geq\ell}}(\overline{\boldsymbol{w}}^\top\boldsymbol{x})\overline{\sigma_{\geq\ell}}(\overline{\boldsymbol{u}}^\top\boldsymbol{x})] = \frac{1}{Z_\sigma^2}\sum_{k\geq\ell}\hat{\sigma}_k^2\cos^k\theta,$$

$$K^{(10)}(\boldsymbol{w}, \boldsymbol{u}) = \frac{1}{Z_\sigma^2}\sum_{k\geq\ell}\hat{\sigma}_k^2 k\cos^{k-1}\theta\frac{1}{\|\boldsymbol{w}\|_2}(\boldsymbol{I} - \overline{\boldsymbol{w}}\overline{\boldsymbol{w}}^\top)\overline{\boldsymbol{u}},$$

$$K^{(11)}(\boldsymbol{w}, \boldsymbol{u}) = \frac{1}{Z_\sigma^2}\sum_{k\geq\ell}\hat{\sigma}_k^2 k(k-1)\cos^{k-2}\theta\frac{1}{\|\boldsymbol{w}\|_2\|\boldsymbol{u}\|_2}(\boldsymbol{I} - \overline{\boldsymbol{w}}\overline{\boldsymbol{w}}^\top)\overline{\boldsymbol{u}}\overline{\boldsymbol{w}}^\top(\boldsymbol{I} - \overline{\boldsymbol{u}}\overline{\boldsymbol{u}}^\top)$$

$$+ \frac{1}{Z_\sigma^2}\sum_{k\geq\ell}\hat{\sigma}_k^2 k\cos^{k-1}\theta\frac{1}{\|\boldsymbol{w}\|_2\|\boldsymbol{u}\|_2}(\boldsymbol{I} - \overline{\boldsymbol{w}}\overline{\boldsymbol{w}}^\top)(\boldsymbol{I} - \overline{\boldsymbol{u}}\overline{\boldsymbol{u}}^\top)$$

$$K^{(20)}(\boldsymbol{w}, \boldsymbol{u}) = \frac{1}{Z_\sigma^2}\sum_{k\geq\ell}\hat{\sigma}_k^2 k(k-1)\cos^{k-2}\theta\frac{1}{\|\boldsymbol{w}\|_2^2}(\boldsymbol{I} - \overline{\boldsymbol{w}}\overline{\boldsymbol{w}}^\top)\overline{\boldsymbol{u}}\overline{\boldsymbol{u}}^\top(\boldsymbol{I} - \overline{\boldsymbol{w}}\overline{\boldsymbol{w}}^\top)$$

$$- \frac{1}{Z_\sigma^2}\sum_{k\geq\ell}\hat{\sigma}_k^2 k\cos^{k-1}\theta\frac{1}{\|\boldsymbol{w}\|_2^2}\overline{\boldsymbol{w}}^\top\overline{\boldsymbol{u}}(\boldsymbol{I} - \overline{\boldsymbol{w}}\overline{\boldsymbol{w}}^\top)$$

$$K^{(21)}(\boldsymbol{w}, \boldsymbol{u})_i = \partial_{u_i}K^{(20)}(\boldsymbol{w}, \boldsymbol{u})$$

$$= \frac{1}{Z_\sigma^2}\sum_{k\geq\ell}\hat{\sigma}_k^2 k(k-1)(k-2)\cos^{k-3}\theta\frac{1}{\|\boldsymbol{w}\|_2^2\|\boldsymbol{u}\|_2}\boldsymbol{e}_i^\top(\boldsymbol{I} - \overline{\boldsymbol{u}}\overline{\boldsymbol{u}}^\top)\overline{\boldsymbol{w}}\cdot(\boldsymbol{I} - \overline{\boldsymbol{w}}\overline{\boldsymbol{w}}^\top)\overline{\boldsymbol{u}}\overline{\boldsymbol{u}}^\top(\boldsymbol{I} - \overline{\boldsymbol{w}}\overline{\boldsymbol{w}}^\top)$$

$$+ \frac{1}{Z_\sigma^2}\sum_{k\geq\ell}\hat{\sigma}_k^2 k(k-1)\cos^{k-2}\theta\frac{1}{\|\boldsymbol{w}\|_2^2\|\boldsymbol{u}\|_2}(\boldsymbol{I} - \overline{\boldsymbol{w}}\overline{\boldsymbol{w}}^\top)\left((\boldsymbol{I} - \overline{\boldsymbol{u}}\overline{\boldsymbol{u}}^\top)\boldsymbol{e}_i\overline{\boldsymbol{u}}^\top + \overline{\boldsymbol{u}}\boldsymbol{e}_i^\top(\boldsymbol{I} - \overline{\boldsymbol{u}}\overline{\boldsymbol{u}}^\top)\right)(\boldsymbol{I} - \overline{\boldsymbol{w}}\overline{\boldsymbol{w}}^\top)$$

$$- \frac{1}{Z_\sigma^2}\sum_{k\geq\ell}\hat{\sigma}_k^2 k(k-1)\cos^{k-2}\theta\frac{1}{\|\boldsymbol{w}\|_2^2\|\boldsymbol{u}\|_2}\boldsymbol{e}_i^\top(\boldsymbol{I} - \overline{\boldsymbol{u}}\overline{\boldsymbol{u}}^\top)\overline{\boldsymbol{w}}\cdot\overline{\boldsymbol{w}}^\top\overline{\boldsymbol{u}}(\boldsymbol{I} - \overline{\boldsymbol{w}}\overline{\boldsymbol{w}}^\top)$$

$$- \frac{1}{Z_\sigma^2}\sum_{k\geq\ell}\hat{\sigma}_k^2 k\cos^{k-1}\theta\frac{1}{\|\boldsymbol{w}\|_2^2}\overline{\boldsymbol{w}}^\top(\boldsymbol{I} - \overline{\boldsymbol{u}}\overline{\boldsymbol{u}}^\top)\boldsymbol{e}_i(\boldsymbol{I} - \overline{\boldsymbol{w}}\overline{\boldsymbol{w}}^\top),$$

$$\tag{15}$$

where $\theta = \arccos(\overline{\boldsymbol{w}}^\top\overline{\boldsymbol{u}})$.

**Part (i)**  Given that $r = \Theta(1/\sqrt{\ell})$ with a small enough hidden constant, we know for $\delta(\boldsymbol{w}, \boldsymbol{u}) \geq r$

$$K(\boldsymbol{w}, \boldsymbol{u}) = \frac{1}{Z_\sigma^2}\sum_{k\geq\ell}\hat{\sigma}_k^2\cos^k\theta \leq \frac{1}{Z_\sigma^2}\sum_{k\geq\ell}\hat{\sigma}_k^2\cdot(1 - r^2/5)^\ell = c < 1,$$

where $c$ is a constant less than 1. Thus, $\rho_1 = \Theta(1)$.

**Part (ii)** For tangent vector $z$ that $z^\top w = 0$, we have ($\|w\|_2 = \|u\|_2 = 1$, $\delta(w, u) \leq r$)

$$
\begin{aligned}
K^{(20)}(w, u)[z, z] =& \frac{1}{Z_\sigma^2} \sum_{k \geq \ell} \hat{\sigma}_k^2 k(k-1) \cos^{k-2} \theta \cdot (\overline{u}^\top z)^2 - \frac{1}{Z_\sigma^2} \sum_{k \geq \ell} \hat{\sigma}_k^2 k \cos^{k-1} \theta \cdot \overline{w}^\top \overline{u} \|z\|_2^2 \\
=& \frac{\|z\|_2^2}{Z_\sigma^2} \left( \sum_{k \geq \ell} \hat{\sigma}_k^2 k(k-1) \cos^{k-2} \theta \cdot (\overline{u}^\top \overline{z})^2 - \sum_{k \geq \ell} \hat{\sigma}_k^2 k \cos^{k-1} \theta \cdot \overline{w}^\top \overline{u} \right) \\
\leq& \frac{\|z\|_2^2}{Z_\sigma^2} \left( \sum_{k \geq \ell} \hat{\sigma}_k^2 k(k-1) \cos^{k-2} \theta \sin^2 \theta - \sum_{\ell \leq k \leq 2\ell} \hat{\sigma}_k^2 k \cos^k \theta \right).
\end{aligned}
$$

For the first term, we have

$$
\begin{aligned}
& \sum_{k \geq \ell} \hat{\sigma}_k^2 k(k-1) \cos^{k-2} \theta \sin^2 \theta \\
\leq& \sum_{k \geq 1/r^2} \hat{\sigma}_k^2 k(k-1) \cdot \Theta(1/k) + \sum_{\ell \leq k \leq 1/r^2} \Theta(k^{-1/2}) r^2 \\
\leq& \sum_{k \geq 1/r^2} \Theta(k^{-3/2}) + \Theta(r) = \Theta(r),
\end{aligned}
$$

where we use Lemma I.1 and $\hat{\sigma}_k^2 = \Theta(k^{-5/2})$ in Lemma A.1.

For the second term, we have

$$
\sum_{\ell \leq k \leq 2\ell} \hat{\sigma}_k^2 k \cos^k \theta \geq \Theta(\ell^{-1/2})(1-r^2)^{2\ell}.
$$

Given that $r = \Theta(1/\sqrt{\ell})$ with a small enough hidden constant, we know

$$
K^{(20)}(w, u)[z, z] \leq -\frac{\|z\|_2^2}{Z_\sigma^2} \Theta(\ell^{-1/2}) = -\Theta(\ell) \|z\|_2^2,
$$

since $Z_\sigma^2 = \Theta(\ell^{-3/2})$.

**Part (iii)** Recall that $\delta(w_i^*, w_j^*) \geq \Delta$ for $i \neq j$. It suffices to bound $\left\| K^{(ij)}(w, u) \right\|_2 \leq h/m_*^2$ for $\theta = \delta(w, u) \geq \Delta$. Given that $\ell \geq \Theta(\Delta^{-2} \log(m_* \ell/h\Delta))$ with large enough hidden constant, from (15) we have for $\|w\| = \|u\| = 1$

$$
\begin{aligned}
K(w, u) \leq& \frac{1}{Z_\sigma^2} \sum_{k \geq \ell} \hat{\sigma}_k^2 (1 - \Delta^2/5)^\ell \leq h/m_*^2, \\
\left\| K^{(10)}(w, u) \right\|_w \leq& \frac{1}{Z_\sigma^2} \sum_{k \geq \ell} \hat{\sigma}_k^2 k \cos^{k-1} \theta \sin \theta \leq \Theta(\ell)(1 - \Delta^2/5)^{\ell-1} \leq h/m_*^2, \\
\left\| K^{(11)}(w, u) \right\|_{w,u} =& \frac{1}{Z_\sigma^2} \sup_{\substack{z_1^\top w = z_2^\top u = 0, \\ \|z_1\|_2 = \|z_2\|_2 = 1}} \sum_{k \geq \ell} \hat{\sigma}_k^2 k(k-1) \cos^{k-2} \theta \overline{u}^\top z_1 \cdot \overline{w}^\top z_2 + \sum_{k \geq \ell} \hat{\sigma}_k^2 k \cos^{k-1} \theta z_1^\top z_2 \\
\leq& \frac{1}{Z_\sigma^2} \sum_{k \geq \ell} \hat{\sigma}_k^2 k(k-1) \cos^{k-2} \theta \sin^2 \theta + \frac{1}{Z_\sigma^2} \sum_{k \geq \ell} \hat{\sigma}_k^2 k \cos^{k-1} \theta \\
\leq& \Theta(\ell^{3/2}) \sum_{k \geq \ell} \Theta(k^{-1/2})(1 - \Delta^2/5)^{k-2} + \Theta(\ell)(1 - \Delta^2/5)^{\ell-1} \leq h/m_*^2,
\end{aligned}
$$

$$\left\|K^{(20)}(\boldsymbol{w},\boldsymbol{u})\right\|_{\boldsymbol{w}} = \frac{1}{Z_\sigma^2} \sup_{\substack{\boldsymbol{z}_1^\top \boldsymbol{w} = \boldsymbol{z}_2^\top \boldsymbol{w} = 0, \\ \|\boldsymbol{z}_1\|_2 = \|\boldsymbol{z}_2\|_2 = 1}} \sum_{k \geq \ell} \hat{\sigma}_k^2 k(k-1)\cos^{k-2}\theta \cdot \overline{\boldsymbol{u}}^\top \boldsymbol{z}_1 \cdot \overline{\boldsymbol{u}}^\top \boldsymbol{z}_2 - \sum_{k \geq \ell} \hat{\sigma}_k^2 k \cos^{k-1}\theta \cdot \overline{\boldsymbol{w}}^\top \overline{\boldsymbol{u}} \cdot \boldsymbol{z}_1^\top \boldsymbol{z}_2$$

$$\leq \frac{1}{Z_\sigma^2} \sum_{k \geq \ell} \hat{\sigma}_k^2 k(k-1)\cos^{k-2}\theta \sin^2\theta + \frac{1}{Z_\sigma^2} \sum_{k \geq \ell} \hat{\sigma}_k^2 k \cos^{k-1}\theta$$

$$\leq \Theta(\ell^{3/2}) \sum_{k \geq \ell} \Theta(k^{-1/2})(1 - \Delta^2/5)^{k-2} + \Theta(\ell)(1 - \Delta^2/5)^{\ell-1} \leq h/m_*^2,$$

$$\left\|K^{(21)}(\boldsymbol{w},\boldsymbol{u})\right\|_{\boldsymbol{w},\boldsymbol{u}}$$

$$= \sup_{\substack{\boldsymbol{z}_1^\top \boldsymbol{w} = \boldsymbol{z}_2^\top \boldsymbol{w} = \boldsymbol{q}^\top \boldsymbol{u} = 0, \\ \|\boldsymbol{z}_1\|_2 = \|\boldsymbol{z}_2\|_2 = \|\boldsymbol{q}\|_2 = 1}} \frac{1}{Z_\sigma^2} \sum_{k \geq \ell} \hat{\sigma}_k^2 k(k-1)(k-2)\cos^{k-3}\theta \sum_i q_i \boldsymbol{e}_i^\top (\boldsymbol{I} - \overline{\boldsymbol{u}\boldsymbol{u}}^\top)\overline{\boldsymbol{w}} \cdot \overline{\boldsymbol{u}}^\top \boldsymbol{z}_1 \cdot \overline{\boldsymbol{u}}^\top \boldsymbol{z}_2$$

$$+ \frac{1}{Z_\sigma^2} \sum_{k \geq \ell} \hat{\sigma}_k^2 k(k-1)\cos^{k-2}\theta \left( \sum_i q_i \boldsymbol{z}_1^\top (\boldsymbol{I} - \overline{\boldsymbol{u}\boldsymbol{u}}^\top)\boldsymbol{e}_i \cdot \overline{\boldsymbol{u}}^\top \boldsymbol{z}_2 + \sum_i q_i \boldsymbol{z}_2^\top (\boldsymbol{I} - \overline{\boldsymbol{u}\boldsymbol{u}}^\top)\boldsymbol{e}_i \cdot \overline{\boldsymbol{u}}^\top \boldsymbol{z}_1 \right)$$

$$- \frac{1}{Z_\sigma^2} \sum_{k \geq \ell} \hat{\sigma}_k^2 k(k-1)\cos^{k-2}\theta \sum_i q_i \boldsymbol{e}_i^\top (\boldsymbol{I} - \overline{\boldsymbol{u}\boldsymbol{u}}^\top)\overline{\boldsymbol{w}} \cdot \overline{\boldsymbol{w}}^\top \overline{\boldsymbol{u}} \cdot \boldsymbol{z}_1^\top \boldsymbol{z}_2$$

$$- \frac{1}{Z_\sigma^2} \sum_{k \geq \ell} \hat{\sigma}_k^2 k \cos^{k-1}\theta \sum_i q_i \overline{\boldsymbol{w}}^\top (\boldsymbol{I} - \overline{\boldsymbol{u}\boldsymbol{u}}^\top)\boldsymbol{e}_i \cdot \boldsymbol{z}_1^\top \boldsymbol{z}_2$$

$$\leq \frac{1}{Z_\sigma^2} \sum_{k \geq \ell} \hat{\sigma}_k^2 k(k-1)(k-2)\cos^{k-3}\theta \sin^3\theta + \frac{2}{Z_\sigma^2} \sum_{k \geq \ell} \hat{\sigma}_k^2 k(k-1)\cos^{k-2}\theta \sin\theta + \frac{1}{Z_\sigma^2} \sum_{k \geq \ell} \hat{\sigma}_k^2 k \cos^{k-1}\theta \sin\theta$$

$$\overset{(a)}{\leq} h/m_*^2,$$

where we use $\hat{\sigma}_k^2 = \Theta(k^{-5/2})$ in Lemma A.1 and (a) the last two terms bound similarly as in $K^{(20)}$ and first term $\frac{1}{Z_\sigma^2} \sum_{k \geq \ell} \hat{\sigma}_k^2 k(k-1)(k-2)\cos^{k-3}\theta \sin^3\theta \leq \Theta(\ell^{3/2}) \sum_{k \geq \ell} \Theta(k^{1/2})(1 - \Delta^2/5)^k \leq h/3m_*^2.$ □

**Lemma G.2** (Regularity conditions on kernel $K$). *Let* $B_{ij} := \sup_{\boldsymbol{w},\boldsymbol{u}} \left\|K^{(ij)}(\boldsymbol{w},\boldsymbol{u})\right\|_{\boldsymbol{w},\boldsymbol{u}}$ *and* $B_0 = B_{00} + B_{10} + 1$, $B_2 = B_{20} + B_{21} + 1$. *We have* $B_{00} = O(1)$, $B_{10} = O(\ell^{1/2})$, $B_{11} = O(\ell)$, $B_{20} = O(\ell)$, $B_{21} = O(\ell^{3/2})$, *and therefore* $B_0 = O(\ell^{1/2})$, $B_2 = O(\ell^{3/2})$.

*Proof.* We compute $B_{ij}$ one by one from (15) (see part (iii) proof in Lemma G.1). Using Lemma I.1 we have

$$B_{00} = \sup_{\boldsymbol{w},\boldsymbol{u}} \left| \frac{1}{Z_\sigma^2} \sum_{k \geq \ell} \hat{\sigma}_k^2 \cos^k\theta \right| \leq 1,$$

$$B_{10} \leq \frac{1}{Z_\sigma^2} \sum_{k \geq \ell} \hat{\sigma}_k^2 k \cos^{k-1}\theta \sin\theta \leq \Theta(\ell^{3/2}) \sum_{k \geq \ell} \Theta(k^{-5/2})k \frac{1}{\sqrt{k}} = O(\ell^{1/2}),$$

$$B_{11} \leq \frac{1}{Z_\sigma^2} \sum_{k \geq \ell} \hat{\sigma}_k^2 k(k-1)\cos^{k-2}\theta \sin^2\theta + \frac{1}{Z_\sigma^2} \sum_{k \geq \ell} \hat{\sigma}_k^2 k \cos^{k-1}\theta$$

$$\leq \Theta(\ell^{3/2}) \sum_{k \geq \ell} \Theta(k^{-5/2})k^2 \frac{1}{k} + \Theta(\ell^{3/2}) \sum_{k \geq \ell} \Theta(k^{-5/2})k = O(\ell),$$

$$B_{20} \leq \frac{1}{Z_\sigma^2} \sum_{k \geq \ell} \hat{\sigma}_k^2 k(k-1) \cos^{k-2}\theta \sin^2\theta + \frac{1}{Z_\sigma^2} \sum_{k \geq \ell} \hat{\sigma}_k^2 k \cos^k\theta$$

$$\leq \Theta(\ell^{3/2}) \sum_{k \geq \ell} \Theta(k^{-5/2}) k^2 \frac{1}{k} + \Theta(\ell^{3/2}) \sum_{k \geq \ell} \Theta(k^{-5/2}) k = O(\ell),$$

$$B_{21} \leq \frac{1}{Z_\sigma^2} \sum_{k \geq \ell} \hat{\sigma}_k^2 k(k-1)(k-2) \cos^{k-3}\theta \sin^3\theta + \frac{2}{Z_\sigma^2} \sum_{k \geq \ell} \hat{\sigma}_k^2 k(k-1) \cos^{k-1}\theta \sin\theta + \frac{1}{Z_\sigma^2} \sum_{k \geq \ell} \hat{\sigma}_k^2 k \cos^{k-1}\theta \sin\theta$$

$$\leq \Theta(\ell^{3/2}) \sum_{k \geq \ell} \Theta(k^{1/2})(1 - \theta^2/5)^{k-3}\theta^3 + \Theta(\ell^{3/2}) \sum_{k \geq \ell} \Theta(k^{-1/2})(1 - \theta^2/5)^{k-1}\theta$$

For first term above $\sum_{k \geq \ell} \Theta(k^{1/2})(1 - \theta^2/5)^{k-3}\theta^3$, using Lemma I.2 we have

$$\sum_{k \geq \ell} \Theta(k^{1/2})(1 - \theta^2/5)^{k-3}\theta^3 \leq \sum_{k \geq \ell} \Theta\left(\frac{1}{\sqrt{\ln(1/(1-\theta^2))}}\right)(1 - \theta^2/5)^{k/2-3}\theta^3$$

$$\leq \sum_{k \geq \ell} \Theta(\theta^2)(1 - \theta^2/5)^{k/2-3} = \Theta(\theta^2)\frac{(1 - \theta^2/5)^\ell}{\theta^2} = O(1).$$

For second term above $\sum_{k \geq \ell} \Theta(k^{-1/2})(1 - \theta^2/5)^{k-1}\theta$ we have

$$\sum_{k \geq \ell} \Theta(k^{-1/2})(1 - \theta^2/5)^{k-1}\theta \leq \Theta(\theta) \int_\ell^\infty x^{-1/2}(1 - \theta^2/5)^x \leq \Theta(\theta)\Theta\left(\frac{1}{\sqrt{\ln(1/(1-\theta^2))}}\right) = O(1).$$

Therefore, we have $B_{21} = O(\ell^{3/2})$. □

## I.1 Technical lemma

We collect few lemma here used in the proof. They mostly rely on direct calculations.

**Lemma I.1.** *For large enough integer $k$, we have*

$$\max |\cos^k \theta \sin\theta| \leq \Theta(1/\sqrt{k}),$$
$$\max |\cos^k \theta \sin^2\theta| \leq \Theta(1/k),$$
$$\max |\cos^k \theta \sin^3\theta| = \Theta(1/k^{3/2}).$$

*Proof.* We only compute the first one $\max |\cos^k \theta \sin\theta| = 1/\sqrt{k}$. Others are similar.

We compute the gradient of $f(\theta) = \cos^k \theta \sin\theta$ and get $f'(\theta) = \cos^{k-1}\theta(\cos^2\theta - k\sin^2\theta)$. We only need to consider $\theta \in [0, 2\pi]$. So the maximum is achieved either at boundary $\theta = 0, \pi$ or $f'(\theta) = 0$. Then one can verify that the bound is true. □

**Lemma I.2.** *For $\beta < 1$ and $k > 0$, we have $k^{1/2}\beta^{k/2} \leq \frac{1}{\sqrt{2\ln(2/\beta)}}$.*

*Proof.* Let $f(k) = k^{1/2}\beta^{k/2}$. We have $f'(k) = \frac{1}{2}k^{-1/2}\beta^{k/2} + k^{1/2}\beta^{k/2}\ln(\beta/2)$. Set $f'(k_0) = 0$ we have $k_0 = \frac{1}{2\ln(2/\beta)}$. It is easy to see $\max f(k) = f(k_0) \leq \frac{1}{\sqrt{2\ln(2/\beta)}}$. □

# J  Notes on Sample Complexity

The current paper focuses on the analysis on population loss, which is already highly non-trivial and requires new ideas that we developed in the paper. The finite-sample analysis is not our focus, so we omit it in the current paper.

For sample complexity, we believe the following strategy would work to get a polynomial sample complexity. We can break down the analysis into 2 parts: early-stage feature learning (Stage 1 and 2) and final-stage feature learning (Stage 3).

- Stage 1 and 2: This should follow the results in Damian et al. (2022). The most important step is to show the concentration of first-step gradient (Stage 1). As shown in Damian et al. (2022), using concentration tools we can get sample complexity $n = \Theta_*(d^2)$, where $n$ is the number of sample and $d$ is input dimension.

- Stage 3: In local convergence regime, all weights have norms bounded in $O_*(1)$ due to $\ell_2$ regularization we have. Thus, we can apply standard concentration tools to show the empirical gradients are close to population gradients given a large enough polynomial number of samples.

Achieving a tight sample complexity is an interesting and challenging open problem that is beyond the scope of current work.

