# OpenReview forum: "How does Gradient Descent Learn Features --- A Local Analysis for Regularized Two-Layer Neural Networks"
_NeurIPS.cc/2024/Conference — NeurIPS 2024 poster_

### Official Review · Reviewer_6HCx · 2024-06-27

**Soundness:** 3
**Presentation:** 3
**Contribution:** 2
**Rating:** 6
**Confidence:** 3

**Summary:**

The present manuscript concerns the study how gradient descent achieves feature learning in a certain class of two-layer neural networks.
The main idea of the manuscript, which builds heavily on previous line of works, is to consider how the population loss is minimized during the first steps of gradient descent. Using that after these steps the student network spans the feature space of the teacher network, the authors show that the successive steps of gradient descent are essential to fully recover the teacher network.

**Strengths:**

The results of the paper show that learning both layers of 2-layer neural networks can be lead to a perfect recovery of the teacher network. This is a stronger result than what was previously known in the literature, where the first layer of the network was trained only for few initial steps and then fixed, leading to a particular class of random feature models.

**Weaknesses:**

As far as I understand the paper concerns mainly the minimization of the population loss and there are no statements about the empirical loss and how sample complexity enters in the results that are presented.

**Questions:**

-

**Limitations:**

See weaknesses

---

> ### Author Rebuttal · Authors · 2024-08-05
>
> We would like to thank reviewers for the feedback. For the concern about sample complexity, please refer to the general response where we address the questions.

---

### Official Review · Reviewer_2mhB · 2024-07-12

**Soundness:** 2
**Presentation:** 2
**Contribution:** 3
**Rating:** 6
**Confidence:** 2

**Summary:**

This paper studies the learning properties of networks trained with gradient descent. More precisely, the authors focus on the late stages of the dynamics where the algorithm learns the ground truth directions. These findings extend the usual ones in the literature that are focused on the early stages of the dynamics. The main theoretical result shows through a local landscape analysis the presence of a strong notion of feature learning, i.e., matching the ground-truth teacher directions in a non-simplified setting where second-layer weights are allowed to have negative values.

**Strengths:**

The main strength of this submission is the nice theoretical contribution. The results are proved through a challenging local landscape analysis that significantly extends previous contributions.

**Weaknesses:**

This submission has no strong weaknesses. However, the presentation could be improved in some parts of the manuscript. I suggest in the following section some possible changes to enhance to quality of the presentation for non-expert readers.

**Questions:**

- I would emphasize more the local loss landscape analysis. As of now, it appears in a small paragraph at the end of Page 2, but introducing the challenges that the authors face for this type of analysis would help to grasp the quality of the contribution.
- The authors correctly highlight many related works that focus on feature learning in the early stages of gradient descent dynamics and how they surpass kernel methods/random features. I believe it would help the non-expert reader to mention [1,2] that showed an equivalent non-linear feature map for networks trained with 1 step. This contrasts the noisy linear equivalent map of random features emerging through Gaussian equivalence.
-  The nice theoretical characterization of this work looks at a stronger feature learning metric, in contrast with closely related works that focus only on weak recovery. Are there other works that dealt with this matter in the context of gradient descent learning? I believe it would be nice to mention more generally previous works that made this "feature learning distinction" even in different contexts. For example, see [3] for Bayes-optimal learning of two-layer networks (specialization phase transition).
- In equation (3) you preprocess the target network to remove its first Hermite coefficient. As correctly mentioned by the authors, this is reminiscent of the procedure done by Damian et al. (2022); however, it would be nice to describe what would happen if such a pre-processing could not be done. Are the strong feature learning capabilities of this network lost due to the presence of a non-vanishing first Hermite direction?
- Could the author be more precise on how they would extend the findings to a polynomial number of samples after Theorem 2? At that point, the hardness of the target function (e.g. information exponent) would matter?
- What do the authors mean when saying "complexity of $f_*$" on Page 4? It would be nice to mathematically formalize this concept.
- The key passage at the end of page 4 is a bit obscure to me and I would suggest rephrasing it more clearly, e.g., remind $\varepsilon_0$.
- What is $\bar{w}_i$ at page 5?
- What is meant for $\varepsilon_0$-net at page 5?
- A schematic drawing of the descent direction after the description on page 6 would help the reader to grasp intuitively the concepts.


[1] A theory of non-linear feature learning with one gradient step in two-layer neural networks. Moniri et al. ICML 2024
[2] Asymptotics of feature learning in two-layer networks after one gradient-step. Cui et al. ICML 2024
[3] The committee machine: Computational to statistical gaps in learning a two-layers neural network. Aubin et al. NeurIPS 2018.

**Limitations:**

The limitations are addressed in the submission.

---

> ### Author Rebuttal · Authors · 2024-08-05
>
> We would like to thank reviewer's efforts to provide detailed feedback. We will incorporate suggestions on the presentation of the paper and discuss more related works in the revision. Below we try to address reviewer's concerns.
>
> > In equation (3) you preprocess the target network to remove its first Hermite coefficient ...
>
> Removing the first Hermite direction is important for the current analysis and Damian et al. (2022). If there is a non-vanishing first Hermite direction, then it will become the dominate term in the first-step gradient. To be more specific, after first-step gradient update neurons $w_i^{(1)}\approx c_1 \beta +c_2 Hw_i^{(0)}$, where $c_1,c_2$ are scalars, $\beta$ is the first Hermite direction and matrix $H$ has full rank within the target subspace due to Assumption 3. When first Hermite direction $\beta\neq 0$, $\beta$ dominates so $w_i^{(1)}\approx c_1 \beta$. This makes all neurons collapse into a single direction.
>
> We would also like to note that this issue is likely a technical difficulty introduced by the setting we consider. In experiments, we can still achieve strong feature learning ability with proper choice of hyperparameters. Providing rigorous analysis beyond current setting is an interesting and challenging problem to consider in the future.
>
> > Could the author be more precise on how they would extend the findings to a polynomial number of samples after Theorem 2 ...
>
> Please refer to the general response where we address the questions regarding sample complexity.
>
>
> > What do the authors mean when saying "complexity of $f_*$" on Page 4? It would be nice to mathematically formalize this concept.
>
> Thanks for the suggestion. We will make it clear in the revision. Here by ``complexity of $f_*$", we mean it only depends on the quantities of $f_*$, such as teacher neurons' norm $|a_i^*|,\|\|w_i^*\|\|$, target subspace dimension $r$ and angle separation $\Delta$.
>
> > What is $\bar{w}_i$ at page 5?
>
> It is the normalized version of neuron $w_i$, i.e., $\bar{w}_i = w_i/\|\|w_i\|\|_2$.
>
> > What is meant for $\varepsilon_0$-net at page 5?
>
> We mean an $\epsilon$-net with $\epsilon=\varepsilon_0$. That is, neurons cover the whole target subspace well in the sense that for every direction $v$ in the target subspace, there exists a neuron $w$ such that the angle between them is small $\angle(w,v)\le \varepsilon_0$.
>
>
> > Presentation of paper and discussion of prior work.
>
> We will improve the presentation of paper based on the suggestions and discuss more related works in the revision.

---

> > ### Comment · Reviewer_2mhB · 2024-08-10
> > **Thank you for the response**
> >
> > I sincerely thank the authors for their rebuttal, I have no further concerns to discuss and I believe the proposed changes will improve the submission. After carefully reading the response along with other reviewers’ comments I would like to keep my score as in the original review.

---

### Official Review · Reviewer_WTVj · 2024-07-14

**Soundness:** 3
**Presentation:** 2
**Contribution:** 2
**Rating:** 6
**Confidence:** 4

**Summary:**

The present paper studies feature learning in the end phase of training. The authors show that when the loss is small, gradient steps capture relevant directions.

**Strengths:**

- The problem of feature learning studied in the paper is important.
- From a technical point of view, the analysis of phase 3 of the algorithm that shows that the local landscape is benign is interesting and can be used in other problems.

**Weaknesses:**

Comments:


1. In the abstract, the authors write "We show that once the loss is below a certain threshold, gradient descent with a carefully regularized objective will capture ground-truth directions". Where is this threshold? Looking at theorem 2, its hard to connect the description in the abstract to the statement proved here.


2. The relation between assumption 3 and information exponent should be made explicit. Based on my understanding, the information exponent of the teacher function is always one (because it is assumed that it has a linear part).

3. What happens if information exponent is larger than 1?

4. Why is the head a and the back layer W have the same regularization parameter \lambda? Also, in line 128, it seems that you are not analyzing the problem at \lambda = 0, but you study the problem in the ridgeless case where \lambda \to 0.

5. The analysis is for gradient descent ran on expected loss (Eq. 2). This problem is not very realistic and a finite sample analysis should be performed. What is the sample complexity here?  I don't think a concentration type argument is possible here; at least in the realistic setting where the dimenion of the covariates and the number of samples are roughly in the same order.

6. The algorithm analyzed in this paper is not close to typical training methods used in practice. What is the role of phase 2? Why are we normalizing in this particular way?

7. In the discussion below Theorem 2, the authors write " In these works, neural networks only learn the target subspace and do random features within it". Is this correct? Specifically what result are the authors pointing towards? Can the authors be more formal here? This discussion is very vague.

8. Instead of the lengthly discussion on the construction of the dual certificate, I think the authors should have discussed in more detail the general take-aways of this result.

9. Missing citations and discussion of prior work. The following (highly relevant) paper have not been discussed. Papers [1] and [2] are missing in the discussion of feature learning in the early phase of training. The authors should also discuss other approaches to analyze feature learning [3], [4], etc.




[1] B Moniri, D Lee, H Hassani, E Dobriban. A Theory of Non-Linear Feature Learning with One Gradient Step in Two-Layer Neural Networks.

[2] H Cui, L Pesce, Y Dandi, F Krzakala, YM Lu, L Zdeborová, B Loureiro, Asymptotics of feature learning in two-layer networks after one gradient-step.

[3] A Radhakrishnan, D Beaglehole, P Pandit, M Belkin, Mechanism for feature learning in neural networks and backpropagation-free machine learning models.

[4] D Beaglehole, I Mitliagkas, A Agarwala, Gradient descent induces alignment between weights and the empirical NTK for deep non-linear networks.

**Questions:**

please see weaknesses

**Limitations:**

please see weaknesses

---

> ### Author Rebuttal · Authors · 2024-08-05
>
> We would like to thank reviewer's efforts to provide detailed feedback. We will try to address reviewer's concerns below and incorporate the suggestions in the revision.
>
> >1. In the abstract, ...
>
> Theorem 2 is a combination of early-stage feature learning (Stage 1 and 2) and final-stage feature learning (Stage 3). This sentence refers to the local convergence result (Stage 3), and the threshold is $\varepsilon_0$ in Theorem 2. We show at the end of Stage 2 (Lemma 4) the loss/optimality gap is below $\varepsilon_0$ so that we enter the local convergence regime in Stage 3 (Lemma 5). We agree that this should have been made clearer and we will address this issue in the revision.
>
> > 2&3. The relation between assumption 3 and information exponent should be made explicit ...
>
> Thanks for the suggestion. We will make it more explicit in the revision. The information exponent of teacher network is indeed 1. Assumption 3 we made here is the same as in Damian et al. (2022), and we use the same preprocessing procedure to remove the linear part. After this preprocessing step, the information exponent becomes 2 due to Assumption 3.
>
> Our results hold when information exponent is 2 after preprocessing. If it's larger, the current analysis does not work, and it would require a generalization of Damian et al. (2022) to make it work (Dandi et al. (2023) explored a generalized version of this problem and left it as a conjecture).
>
> Neural networks can learn representations with gradient descent, Damian et al., 2022
>
> How Two-Layer Neural Networks Learn, One (Giant) Step at a Time, Dandi et al., 2023
>
> >4. Why is the head a and the back layer W ...
> - Regularization on $a$ and $W$:
>
> This is the same as using weight decay on both layers. At the minimum we have $\|\|w_i\|\|_2^2+|a_i|^2 = 2\|\|w_i\|\|_2|a_i|$, so $\ell_2$ regularization $\sum_i\|\|w_i\|\|_2^2+|a_i|^2$ becomes $2\sum_i\|\|w_i\|\|_2|a_i|$ that is effectively $\ell_1$ regularization over the norm $\|\|w_i\|\|_2|a_i|$ of neurons. Such a regularization favors sparse solution and helps us to recover the ground-truth directions and remove irrelevant directions.
>
> - Analyze the problem of $\lambda=0$ or $\lambda\to0$:
>
> Our goal is to minimize the loss with $\lambda=0$. However, directly analyzing the unregularized case is challenging so we choose to analyze the case where $\lambda$ is positive but gradually decreasing to 0 ($\lambda\to 0$). In this way, the solution we get will eventually converge to the minima with $\lambda=0$. In Theorem 2 we show that the unregularized loss ($\lambda=0$) is small at the end of training.
>
> >5. The analysis is for gradient descent ran on expected loss ...
>
> Please refer to the general response where we address the questions regarding sample complexity.
>
> >6. The algorithm analyzed in this paper ...
>
> - Non-standard algorithm:
>
> Recent feature learning literature often focuses on early-stage learning, using layer-wise training (training 1st-layer weights $W$ first, then 2nd-layer weights $a$), which is also not standard in practice. As discussed in the paper, analyzing the entire training dynamics of standard gradient descent, especially the middle stage (Stage 2), is technically challenging and remains an open problem. We hope our results contribute to understanding standard training methods.
>
> - Role of Stage 2:
>
> In our algorithm, the Stage 1 and 2 essentially follow the similar layer-wise training procedure that are common in early-stage feature learning literature. The role of Stage 2 is to do a regression on top of the learned feature after Stage 1. This makes the loss small so that we enter the local convergence regime in Stage 3.
>
> - Regularization and balancing:
>
> We assume `normalizing' mentioned by reviewer refers to the regularization or the norm balance. We'd be happy to further clarify if this does not answer the original question.
>
> The regularization is essentially the same as $\ell_2$ regularization or weight decay used in practice, as $\|\|w_i\|\|_2^2 + |a_i|^2 \ge 2\|\|w_i\|\|_2|a_i|$, with equality at the minimum of $\ell_2$ regularized loss. The reason of using weight decay is further discussed in Section 5: it induces an effective $\ell_1$ regularization over the neurons and helps to reduce norm cancellation between neurons.
>
> The norm balancing step is just a preparation step for Stage 3 for technical convenience to enter the local convergence regime.
>
> >7. In the discussion below Theorem 2, ...
>
> We will make it clear in the revision. To be more specific, we could look at Damian et al. (2022) (although the discussion also applies to other works that rely on (large) first-step gradient, e.g., Ba et al. (2022), Abbe et al. (2022)). It is shown that after the first-step gradient update neurons $w_i^{(1)}\approx c Hw_i^{(0)}$, where $c$ is a scalar and $w_i^{(0)}$ is neuron $w_i$ at random initialization. Given the assumption that matrix $H$ has full rank within the target subspace, $w_i^{(1)}$ is essentially sampled randomly from target subspace. This suggests neural networks in fact only learn the target subspace and do random feature within it.
>
> >8. Instead of the lengthly discussion on the construction of the dual certificate, ....
>
> Thanks for the suggestion. We do believe the construction of the dual certificate is the core technical contribution. However We will clarify general take-aways more clearly in the revision. Our main finding is that training both layers of 2-layer networks to convergence can achieve 0 loss and recover ground-truth directions. This finding highlights a strong form of feature learning that complements the early-stage feature learning literature, where the first-layer weights are typically fixed after one or a few steps of gradient descent.
>
> >9. Missing citations and discussion of prior work...
>
> Thanks for the references. We will add discussions about these works in the revision.

---

> > ### Comment · Reviewer_WTVj · 2024-08-10
> >
> > I thank the author for their response. This resolves most of my concerns regarding the paper.
> >
> > This paper mostly focuses on minimizing population loss instead of the training loss. However, the analysis of even the population loss is already very challenging. Thus, I will increase my score to 6.

---

### Author Rebuttal · Authors · 2024-08-05

We appreciate the reviewers' detailed feedback and their efforts in providing constructive comments. One common question raised by reviewers is about sample complexity. We try to address it below.

First, we would like to emphasize that even the analysis on population loss is highly non-trivial and requires new ideas that we developed in the paper. The finite-sample analysis is not our focus, so we omit it in the current paper.

For sample complexity, we believe the following strategy would work to get a polynomial sample complexity. We can break down the analysis into 2 parts: early-stage feature learning (Stage 1 and 2) and final-stage feature learning (Stage 3).

- Stage 1 and 2: This should follow the results in Damian et al. (2022). The most important step is to show the concentration of first-step gradient (Stage 1). As shown in Damian et al. (2022), using concentration tools we can get sample complexity $n = \Theta(d^2)$, where $n$ is the number of sample and $d$ is input dimension.

- Stage 3: In local convergence regime, all weights have norms bounded in $O(1)$ due to $\ell_2$ regularization we have. Thus, we can apply standard concentration tools to show the empirical gradients are close to population gradients given a large enough polynomial number of samples.


In this way, we could get a polynomial sample complexity result.

Below we address more detailed questions on sample complexity:

- Better sample complexity

    It is indeed not clear how to get a better sample complexity bound like the one reviewer asked for ($n=\Theta(d)$). A simple parameter counting approach would suggest likely $\Theta(dr)$ samples are needed. As discussed in Damian et al. (2022), they showed at least $\Theta(d)$ sample is required in this setting and only provided a sample complexity depending on $\Theta(d^2)$. Moreover, if we focus on the first-step gradient, Dandi et al. (2023) showed that with only $n=\Theta(d)$ samples only 1 direction could be learned.  In contrast, $n = \Theta(d^2)$ is essential for neurons to learn multiple relevant directions of the target with a single gradient step. This suggests more than $\Theta(d)$ samples might be necessary to learn multi-index models like the 2-layer nets we consider in the paper. Improving sample complexity, especially in early-stage feature learning (e.g., Abbe et al. (2023), Damian et al. (2024), Arnaboldi et al. (2024)), is an active and interesting research topic, which we leave for future work.

- Impact of information exponent

    The information exponent indeed would matter for sample complexity. Due to Assumption 3, the information exponent after preprocessing step is 2 in our setting. In the outline above, $d^2$ sample complexity appears in Stage 1 and 2. This is because in fact we need an accurate estimation of degree-2 Hermite polynomial (matrix $H$), as discussed in Damian et al. (2022). When the target function is harder (information exponent is larger), we expect the sample complexity would increase accordingly.




Neural networks can learn representations with gradient descent, Damian et al., 2022

Sgd learning on neural networks: leap complexity and saddle-to-saddle dynamics, Abbe et al., 2023

How Two-Layer Neural Networks Learn, One (Giant) Step at a Time, Dandi et al., 2023

Computational-Statistical Gaps in Gaussian Single-Index Models, Damian et al., 2024

Repetita Iuvant: Data Repetition Allows SGD to Learn High-Dimensional Multi-Index Functions, Arnaboldi et al., 2024

---

### Decision · Program_Chairs · 2024-09-25

**Decision:**

Accept (poster)

**Comment:**

This paper provides a convergence analysis of gradient descent to learn a two layer neural network in a teacher-student setting. Under a bit strong assumption, it is shown that the dynamics decreases the loss to 0 and it can be divided into three stages. The computational complexity is polynomial.

One main drawback of this paper is that the dynamics is analyzed only for the population risk minimization. A typical approach in this direction is to show the sample complexity as well as its polynomial time computational complexity and discuss its optimality (referring an SQ or CSQ lower bound). This drawback is somehow resolved in the author rebuttal but it requires substantial revision to include the finite sample analysis.
On the other hand, the analysis itself is not trivial. The authors provided an interesting contribution even though the loss is of population.

As a summary, I think the result in this paper is informative in the literature and worth publication in NeurIPS. Hence, I recommend acceptance of this paper.